

# Phylogenetic relationships of the genus *Mischonyx* Bertkau, 1880, with taxonomic changes and three new species description (Opiliones: Gonyleptidae)

Caio Gueratto[1], Alípio Benedetti[1,2] and Ricardo Pinto-da-Rocha[1]

[1] Departamento de Zoologia/Instituto de Biociências, Universidade de São Paulo, São Paulo, Brazil
[2] Centro Universitário Metodista Izabela Hendrix, Belo Horizonte, Minas Gerais, Brazil

## ABSTRACT

The type species of *Mischonyx* Bertkau 1880, *Mischonyx squalidus*, was described based on a juvenile. The holotype is lost. Based on a revision of publications, the genus includes 12 species, all in Brazil. The objectives of this research are: to propose a phylogenetic hypothesis for *Mischonyx* based on Total Evidence (TE); propose taxonomic changes based on the phylogeny; and analyze the phylogenetic hypothesis biogeographically. Using the exemplar approach to taxon selection, we studied 54 specimens, 15 outgroups and 39 ingroup taxa using seven molecular markers (28S, 12S and 16S ribosomal genes, citochrome oxidase subunit I gene, carbamoyl-phosphate synthetase gene, internal transcribed spacer subunit 2 and histone H3 gene), totaling 3,742 bp, and 128 morphological characters. We analyzed the dataset under three optimality criteria: Maximum likelihood (ML), Maximum parsimony (MP) and Bayesian. We discuss the transformation of character states throughout the phylogeny, the different phylogenetic hypotheses using different datasets and the congruence of evidence between the clades obtained by the phylogenetic analysis and the biogeographical hypothesis for the Atlantic Forest areas of endemism. We estimate that *Mischonyx* clade diverged 50.53 Mya, and inside the genus there are two major clades. One of them cointains species from Paraná, Santa Catarina, South of São Paulo and Serra do Mar Areas of Endemism and the other has species from Espinhaço, Bocaina, South coast of Rio de Janeiro and Serra dos Órgãos Areas of Endemism. The first split inside these two clades occurred at 48.94 and 44.80 Mya, respectively. We describe three new species from Brazil: *Mischonyx minimus* **sp. nov.** (type locality: Petrópolis, Rio de Janeiro), *Mischonyx intervalensis* **sp. nov.** (type locality: Ribeirão Grande, São Paulo) and *Mischonyx tinguaensis* **sp. nov** (type locality: Nova Iguaçu, Rio de Janeiro). The genus *Urodiabunus* Mello-Leitão, 1935 is considered a junior synonym of *Mischonyx*. *Weyhia spinifrons* Mello-Leitão, 1923; *Weyhia clavifemur* Mello-Leitão, 1927 and *Geraeocormobius reitzi* Vasconcelos, 2005 were transferred to *Mischonyx*. *Mischonyx cuspidatus* (*Roewer, 1913*) is a junior synonym of *M. squalidus* Bertkau, 1880. In the results of the phylogenetic analyses, *Gonyleptes antiquus* Mello-Leitão, 1934 (former *Mischonyx antiquus*) does not belong in *Mischonyx* and its original combination is re-established. As it is now defined, *Mischonyx* comprises 17 species, with seven new combinations.

Corresponding author
Caio Gueratto,
caio.gueratto@gmail.com

## INTRODUCTION

Laniatores is the most diverse suborder within Opiliones. There are more than 4,200 species in the group (*Kury, 2020*), of which at least 2,400 are from the Neotropical region (*Kury, 2003*). The evolution and phylogenetic relationships of most families and genera within the suborder have been poorly studied.

Modern taxonomists base their classifications on cladistics hypotheses (*e.g.*, *Bragagnolo & Pinto-da-Rocha, 2009*; *DaSilva & Gnaspini, 2010*; *Pinto-da-Rocha, 2002*; *Pinto-da-Rocha & Bragagnolo, 2010*) using a number of markers, including molecular data (*e.g.*, *Bragagnolo et al., 2015*; *Pinto-da-Rocha et al., 2014*). This also applies to the taxonomy of Laniatores to a certain extent, but despite recent progress, the classification system devised by Carl F. Roewer (1881–1963) still prevails in this group. Roewer based his nomenclature and groups on a few arbitrary characters. As a result, he created a lot of monotypic genera and placed closely-related species in distinct clades (*Pinto-da-Rocha et al., 2012*).

Gonyleptidae *Sundevall, 1833* is one of the families within Laniatores that includes many monotypic genera and artificial groups. According to *Kury (1990)*, there are many species in the family that have been cited only once, suggesting that there may be many synonyms to be established. Recent research on Gonyleptidae subfamilies using phylogenetic systematics has found evidence supporting several groups (*Benedetti & Pinto-da-Rocha, 2019*; *Bragagnolo & Pinto-da-Rocha, 2012*; *DaSilva & Gnaspini, 2010*; *DaSilva & Pinto-da-Rocha, 2010*; *Pinto-da-Rocha & Bragagnolo, 2010*). In addition, with the use of molecular data in phylogenetic inference, *Pinto-da-Rocha et al. (2014)*, *Benedetti (2017)* and *Benavides, Pinto-da-Rocha & Giribet (2021)*, proposed new relationships among most Gonyleptidae subfamilies. One subfamily, however, Gonyleptinae *Sundevall, 1833* (39 genera, 140 species in total), remains to be analyzed under a phylogenetic framework (*Kury, 2003*). The diagnosis of the subfamily, based on the number of areas on the dorsal scutum and the absence of certain features that characterize other subfamilies (*Pinto-da-Rocha et al., 2014*), suggests that Gonyleptinae is a polyphyletic clade and it is possible that several genera will need to be transferred out of it.

### Mischonyx background

*Bertkau (1880)* described *Mischonyx squalidus*, type species of the genus by monotypy, from Copacabana, Rio de Janeiro, Brazil. *Rower (1923)* pointed out that the holotype was a juvenile, evidenced by the incomplete tarsal segmentation. After *Bertkau (1880)*, the genus remained monotypic until *Kury (2003)*, who synonymized other genera (cited below) within *Mischonyx*.

In the first half of the 20th century, Carl Roewer and Candido Mello-Leitão described genera of interest for this research, namely, *Ilhaia Roewer, 1913*, *Weyhia Roewer, 1913*, *Xundarava Mello-Leitão, 1927a*, *Eduardoius Mello-Leitão, 1931a*, *Geraecormobiella*

*Mello-Leitão, 1931b* and *Giltaya Mello-Leitão, 1932*. In addition, Mello-Leitão described and transferred species into these genera and recognized *Weyhia* as a synonym of *Geraeocormobius* (*Mello-Leitão, 1940*).

In the second half of the 20th century, B. Soares and H. Soares synonymized *Ilhaia* with *Eduardoius* (*Soares, 1943*), *Geraecormobiella* with *Geraeocormobius* Holmberg, 1887 (*Soares, 1945c*) and *Ilhaia* with *Xundarava* (*Soares & Soares, 1987*). Along with that, the authors synonymized some species of these genera and described more species.

*Kury (2003)* synonymized *Ilhaia* and *Giltaya* with the almost forgotten genus *Mischonyx*. Besides that, he transferred *G. antiquus* (then in *Paragonyleptes*) to *Mischonyx*. Since the holotype of *Mischonyx squalidus* is lost, Kury based his conclusions on Roewer's drawings and description. In his catalog, Kury considers *Mischonyx* as including 11 species.

Finally, in *Vasconcelos (2004, 2005a)* the two last *Mischonyx* species were described: *Mischonyx kaisara*, from the coast of the state of São Paulo, and *Mischonyx poeta*, from the northern portion of the state of Rio de Janeiro. He also described *Gearaeocormobius reitzi Vasconcelos, 2005b*. Besides these publications, there is one unpublished M.Sc. dissertation on the taxonomy of *Mischonyx* taxonomy (*Vasconcelos, 2003*).

The last published research containing taxonomical remarks on the genus, *Pinto-da-Rocha et al. (2012)*, considered 12 valid species within *Mischonyx*: *M. anomalus* (*Mello-Leitão, 1936*); *M. antiquus* (*Mello-Leitão, 1934*); *M. cuspidatus* (*Roewer, 1913*); *M. fidelis* (*Mello-Leitão, 1931a*); *M. insulanus* (*Soares, 1972*); *M. intermedius* (*Mello-Leitão, 1935c*); *M. kaisara Vasconcelos, 2004*; *M. poeta Vasconcelos, 2005a*; *M. processigerus* (*Soares & Soares, 1970*); *M. scaber* (*Kirby, 1819*); *M. squalidus Bertkau, 1880* and *M. sulinus* (*Soares & Soares, 1947*).

The biology of *Mischonyx cuspidatus* has been extensively studied, including the chemical composition of the odoriferous glands (*Rocha et al., 2013*), defensive behavior (*Dias & Willemart, 2013*; *Dias et al., 2014*; *Willemart & Pellegatti-Franco, 2006*), odor sensitivity (*Dias, 2017*) and synanthropic behavior (*Mestre & Pinto-da-Rocha, 2004*). Although there has been a lot of discussion on the taxonomy of *Mischonyx*, no phylogenetic hypothesis has yet been proposed for the genus.

The main goal of this work is to propose a phylogenetic hypothesis for *Mischonyx*, based on total evidence combining sequences from seven genes and morphological characters that include the external morphology and genitalia. In addition, we propose taxonomical changes, describe new species and make remarks on biogeography based on the phylogenetic hypothesis.

## MATERIALS AND METHODS

### Species distribution and areas of endemism

To build an updated map of the geographical distribution of *Mischonyx* species, we used DIVA-GIS to plot the geographical coordinates of the specimens available in the collection of Museu de Zoologia da Universidade de São Paulo (MZSP) and the Arachnology Lab (IB-USP) tissue collection. We also included the type localities and records extracted from *Kury (2003)*.

**Table 1 Sequenced genes per taxon with their respective identification vouchers and GenBank access number (Outgroup only).**

| Sequence ID | ITS | 28S | COI | 16S | 12S | CAD | H3 |
|---|---|---|---|---|---|---|---|
| *Ampheres leucopheus* 0377 | MT957104 | MT990789 | MT992270 | MW000844 | MW000802 | MW017372 | MW017447 |
| *Deltaspidium asperum* 2201 | MT957119 | MT990804 | MT992285 | MW000859 | MW000818 | MW017385 | MW017418 |
| *Deltaspidium orguense* 0520 | MT957106 | MT990791 | MT992272 | MW000858 | MW000804 | MW017374 | MW017454 |
| *Deltaspidium tenue* | | MT990783 | MT992264 | MW000857 | MW000796 | MW017370 | MW017436 |
| *Gonyleptes antiquus* 3707 | MT957132 | MT990822 | MT992301 | MW000847 | MW000834 | MW017397 | MW017416 |
| *Gonyleptes antiquus* 3708 | MT957133 | MT990823 | MT992302 | MW000848 | MW000835 | MW017398 | MW017417 |
| *Gonyleptes horridus* 0103 | MT957100 | MT990784 | MT992265 | MW000841 | MW000797 | | MW017448 |
| *Heliella singularis* 1837 | MT957113 | MT990798 | MT992279 | MW000839 | MW000812 | | MW017412 |
| *Multumbo dimorphicus* 0069 | MT957096 | MT990778 | MT992259 | MW000865 | MW000791 | | MW017455 |
| *Multumbo terrenus* 2136 | MT957117 | MT990802 | MT992283 | MW000864 | MW000816 | MW017383 | MW017425 |
| *Piassagera brieni* 0141 | | MT990787 | MT992268 | MW000842 | MW000800 | | MW017409 |
| *Promitobates ornatus* 0054 | | MT990776 | MT992257 | MW000837 | MW000789 | | MW017406 |
| *Pseudotroglus telluris* 2118 | MT957115 | MT990800 | MT992281 | MW000843 | MW000814 | MW017381 | MW017411 |
| *Roeweria virescens* 0081 | | MT990780 | MT992261 | MW000838 | MW000793 | | MW017407 |
| *Sodreana sodreana* 0056 | MT957095 | MT990777 | MT992258 | MW000852 | MW000790 | MW017366 | MW017410 |

**Note:**
Each code represents the GenBank access number for each gene sequence. Blank cells represent individuals that we could not acquire sequences.

The nomenclature used for the areas of endemism of the Atlantic Rainforest and their delimitation follows *DaSilva, Pinto-da-Rocha & Morrone (2017)*.

## Type specimens and ingroup selection

We analyzed (see Table 1) at least one type specimen from each valid *Mischonyx* species listed in *Kury (2003)*, except the holotype of *Mischonyx squalidus*, which has been lost. Type specimens were compared with the harvestmen tissue collection of the Arachnology Lab (Instituto de Biociências - Universidade de São Paulo). Additionally, we collected fresh specimens for DNA extraction. Individuals that resembled *Mischonyx* species but did not match described species were also included in the analysis. The ingroup used in the phylogenetic analysis is listed in Table 2.

## Outgroup selection

Besides the ingroup specimens mentioned above, we included in our matrix specimens from different gonyleptid subfamilies, as follows: Caelopyginae *Sørensen, 1884*, Gonyleptinae, Hernandariinae *Sørensen, 1884*, Mitobatinae *Simon, 1879*, Pachylinae *Sørensen, 1884*, Progonyleptoidellinae *Soares & Soares, 1985*, Sodreaninae *Soares & Soares, 1985*. Following the exemplar approach to taxon selection, we included up to two species from of these subfamilies. The species used as outgroups are showin in Table 1.

## Molecular data acquisition

Specimens for the molecular analysis were kept at 92–98% ethanol and at −20 °C. Our lab has a database with gene sequences originated from different projects. We used sequences

**Table 2 Sequenced genes per taxon with their respective identification vouchers and GenBank access number (Ingroup only).**

| Sequence ID | ITS | 28S | COI | 16S | 12S | CAD | H3 |
|---|---|---|---|---|---|---|---|
| *Mischonyx anomalus* 0122 | MT957102 | MT990786 | MT992267 | MW000854 | MW000799 | | MW017452 |
| *Mischonyx anomalus* 0693 | MT957108 | MT990793 | MT992274 | MW000853 | MW000807 | MW017376 | MW017423 |
| *Mischonyx anomalus* 1638 | MT957112 | MT990797 | MT992278 | MW000840 | MW000811 | MW017379 | MW017421 |
| *Mischonyx anomalus* 2953 | MT957122 | MT990808 | MT992289 | MW000856 | MW000821 | MW017388 | MW017424 |
| *Mischonyx clavifemur* 0079 | MT957097 | MT990779 | MT992260 | MW000862 | MW000792 | MW017367 | MW017449 |
| *Mischonyx clavifemur* 0845 | MT957109 | MT990794 | MT992275 | MW000863 | MW000808 | | MW017422 |
| *Mischonyx fidelis* 4115A | MT957135 | MT990825 | MT992304 | MW000872 | | MW017400 | MW017441 |
| *Mischonyx fidelis* 4115B | MT957136 | MT990826 | MT992305 | MW000867 | | MW017401 | MW017442 |
| *Mischonyx insulanus* 1455 | MT957111 | MT990796 | MT992277 | MW000869 | MW000810 | MW017378 | |
| *Mischonyx insulanus* 3066 | MT957123 | MT990811 | MT992290 | MW000855 | | MW017389 | MW017408 |
| *Mischonyx intermedius* 4116A | MT957137 | MT990827 | MT992306 | MW000850 | MW000831 | MW017402 | MW017426 |
| *Mischonyx intermedius* 4116B | MT957138 | MT990809 | MT992307 | MW000849 | MW000832 | MW017403 | MW017427 |
| *Mischonyx intermedius* 4117A | MT957139 | MT990810 | MT992308 | MW000851 | MW000833 | MW017404 | MW017428 |
| *Mischonyx intervalensis* **sp. nov.** 0099 | MT957099 | MT990782 | MT992263 | MW000845 | MW000795 | MW017369 | MW017451 |
| *Mischonyx intervalensis* **sp. nov.** 3709 | MT957134 | MT990824 | MT992303 | MW000846 | MW000836 | MW017399 | MW017420 |
| *Mischonyx kaisara* 0143 | MT957103 | MT990788 | MT992269 | | MW000801 | | MW017414 |
| *Mischonyx kaisara* 1374 | MT957110 | MT990795 | MT992276 | MW000868 | MW000809 | MW017377 | MW017405 |
| *Mischonyx kaisara* 2345 | MT957120 | MT990805 | MT992286 | MW000866 | MW000819 | MW017386 | MW017415 |
| *Mischonyx kaisara* 3575 | MT957124 | MT990814 | MT992293 | MW000860 | MW000824 | | MW017413 |
| *Mischonyx minimus* **sp. nov.** 3649 | MT957128 | MT990818 | MT992297 | MW000879 | MW000828 | MW017393 | MW017443 |
| *Mischonyx parvus* 3621A | MT957125 | MT990815 | MT992294 | MW000875 | MW000825 | MW017390 | MW017437 |
| *Mischonyx parvus* 3621B | MT957126 | MT990816 | MT992295 | MW000877 | MW000826 | MW017391 | MW017438 |
| *Mischonyx parvus* 3651A | MT957131 | MT990821 | MT992300 | MW000876 | MW000806 | MW017396 | MW017439 |
| *Mischonyx poeta* 3650A | MT957129 | MT990819 | MT992298 | MW000880 | MW000829 | MW017394 | MW017445 |
| *Mischonyx poeta* 3650B | MT957130 | MT990820 | MT992299 | MW000881 | MW000830 | MW017395 | MW017446 |
| *Mischonyx processigerus* 0463 | MT957105 | MT990790 | MT992271 | MW000870 | MW000803 | MW017373 | MW017450 |
| *Mischonyx processigerus* 3648 | MT957127 | MT990817 | MT992296 | MW000871 | MW000827 | MW017392 | MW017444 |
| *Mischonyx reitzi* 0672 | MT957107 | MT990792 | MT992273 | MW000861 | MW000805 | MW017375 | MW017419 |
| *Mischonyx spinifrons* 0111 | MT957101 | MT990785 | MT992266 | MW000884 | MW000798 | MW017371 | MW017431 |
| *Mischonyx spinifrons* 2120 | MT957116 | MT990801 | MT992282 | MW000885 | MW000815 | MW017382 | MW017432 |
| *Mischonyx spinifrons* 2151 | MT957118 | MT990803 | MT992284 | MW000886 | MW000817 | MW017384 | MW017430 |
| *Mischonyx spinifrons* 2809 | MT957121 | MT990807 | MT992288 | MW000882 | | MW017387 | MW017433 |
| *Mischonyx spinifrons* 3363 | | MT990812 | MT992291 | MW000887 | MW000822 | | MW017434 |
| *Mischonyx spinifrons* 3375 | | MT990813 | MT992292 | MW000883 | MW000823 | | MW017435 |
| *Mischonyx squalidus* 0085 | MT957098 | MT990781 | MT992262 | MW000873 | MW000794 | MW017368 | MW017453 |
| *Mischonyx squalidus* 2026 | MT957114 | MT990799 | MT992280 | MW000874 | MW000813 | MW017380 | MW017440 |
| *Mischonyx tinguaensis* **sp. nov.** 2361 | | MT990806 | MT992287 | MW000878 | MW000820 | | MW017429 |

**Note:**
Each code represents the GenBank access number for each gene sequence. Blank cells represent individuals that we could not acquire sequences.

from that source and sequenced the DNA from additional species using muscular tissue from coxa IV (*Pinto-da-Rocha et al., 2014*). Alternatively, when the individual to be sequenced was small, we used tissues from the chelicerae and pedipalps. We used the kit

Agencourt® DNAdvance System (EUA; Beckman Coulter, Brea, California, USA) for extractions and modified the protocols according to *Pinto-da-Rocha et al. (2014)*.

From the extracted DNA, we amplified seven molecular *loci*: the ribosomal nuclear gene 28S rRNA; the ribosomal mitochondrial genes 12S rRNA and 16S rRNA; the nuclear sequences of the internal transcribed spacer subunit 2 (ITS2), carbamoylphosphate synthetase 2 gene (CAD) and the histone H3 gene (H3); and the mitochondrial cytochrome oxidase subunit I gene (COI). For polymerase chain reactions (PCRs), we used Thermo-fisher Taq kit, following the concentration present in *Pinto-da-Rocha et al. (2014)*.

The primers used to amplify the genes were:

– 28S rRNA: overlap of two primer sets: 28SRDIAF–28SRD4B (*Arango & Wheeler, 2007* and *Edgecombe & Giribet, 2006*, respectively) and 28SD3AP–28SB (*Reyda & Olson, 2003* and *De Ley et al., 1999*, respectively);

– 16S rRNA: 16SpotFN–16SBR (*Pinto-da-Rocha et al., 2014* and *Palumbi, 1996*, respectively);

– 12S rRNA: 12SAIN–12SOP2RN (*Pinto-da-Rocha et al., 2014*);

– COI: dgLCO1490–dgHCO2198 (*Meyer, 2003*). Alternatively, LCO1490–HCO2198 (*Folmer et al., 1994*) and LCO1490–HCOout (*Folmer et al., 1994* and *Prendini, Weygoldt & Wheeler, 2005*, respectively);

– H3: H3AF–H3AR (*Colgan et al., 1998*). Alternatively, H3AF_edit (5′-GCVMGVAAGTCYACVGGMGG-3′) – H3AR_edit (5′-ATGGTSACTCTCTTGGCGTGR-3′), made at the Molecular Systematics Laboratory of IBUSP;

– ITS2: 5.8SF–CAS28Sb1d (*Ji, Zhang & He, 2003*);

– CAD: op_cad_F1 – op_cad_R1 (*Peres et al., 2018*).

– We conducted PCR reactions in an Eppendorf Mastercycler® gradient thermal cycler and the cycles and temperature used in this work are the same as in *Pinto-da-Rocha et al. (2014)*. Afterwards, we inspected the PCR products using agarose gel electrophoresis (2% agarose), purified the products using Agencourt Ampure XP (Beckman Coulter, Brea, CA, USA) and quantified the products using a Thermo Scientific NanoDrop spectrophotometer. In order to prepare the products for sequencing, we used the BigDye® Terminator v3.1 Cycle Sequencing Kit (Applied Biosystems, Waltham, MA, USA). The precipitation was with sodium acetate and the sequencing process was in an ABI PRISM® 3100 Genetic Analyser/HITACHI (Applied Biosystems, Waltham, MA, USA).

We assembled the contiguous sequences using Consed/PhredPhrap package (*Ewing & Green, 1998*; *Ewing et al., 1998*; *Gordon, Abajian & Green, 1998*; *Gordon, Desmarais & Green, 2001*). We queried the contigs against the online NCBI BLAST database to check for contamination from other external sources. We aligned the sequences using MAFFT (*Katoh et al., 2002*), visualized, and edited the results in Aliview (*Larsson, 2014*). We searched for stop codons in the coding genes (COI, CAD and H3) in Aliview. We trimmed the coding genes sequences to match the first base of the sequences with the

first codon position. All sequences are at GenBank and their respective access codes are in Tables 1 and 2.

## Morphological data acquisition, terminology and new species drawings

We coded the external morphological characters after analyzing the type material and other individuals of the species when available under a Zeiss Stemi DV4 stereomicroscope. Analyzis of the male genitalia characters was conducted under a Scanning Electron Microscopy (SEM). We followed the protocol of *Pinto-da-Rocha (1997)* to dissect and prepare the genitalia for Scanning Electron Microscope (Zeiss DSM940, from Instituto de Biociências, Universidade de São Paulo) and built the character matrix using Mesquite 3.51 (*Maddison & Maddison, 2017*). We coded most characters as binary to avoid redundancy and tried to ensure that all characters were independent from each other (*Strong & Lipscomb, 1999*). Nonetheless, to avoid building non-comparable characters, in some cases, we used multistate characters and treated them as unordered. The character descriptions follow *Sereno (2007)*. The complete character matrix is available online, at MorphoBank (http://morphobank.org/permalink/?P3599).

The general terminology follows *DaSilva & Gnaspini (2010)*. Granules refer to minute elevations, concentrated on a particular region or article. Tubercles are elevations that are clearly distinguishable from granules by their height and width and can have blunt or acuminated apex. Spines are acuminated elevations present on the ocularium. Apophyses, which have different shapes, are the armatures present on coxa IV, free tergites, anterior and posterior margins. The terminology for the shape of the dorsal scutum follows *Kury & Medrano (2016)*. The terminology for the penial macrosetae follows *Kury & Villareal (2015)*.

We used a stereomicroscope coupled with a *camara lucida* to make our drawings. After that, we digitalized them and made corrections on the background using Adobe Photoshop Lightroom 6.0®.

## Nomenclatural acts and collecting license

The electronic version of this article in Portable Document Format (PDF) will represent a publication according to the International Commission on Zoological Nomenclature (ICZN), and hence the new names contained in the electronic version are effectively published under that Code from the electronic edition alone. This published work and the nomenclatural acts it contains have been registered in ZooBank, the online registration system for the ICZN. The ZooBank LSIDs (Life Science Identifiers) can be resolved and the associated information can be viewed using any standard web browser by appending the LSID to the prefix http://zoobank.org/. The LSIDs for this publication are: urn:lsid: zoobank.org:act:A6F34641-1AF1-4BE2-A16A-4A4497ECA1FC; urn:lsid:zoobank.org: act:3DDE0A87-E9F6-4504-9C54-6DC37D202A0E; urn:lsid:zoobank.org:act:5FA4CC13-EC27-4E3A-AB19-81A97FE74177. The online version of this work is archived and available from the following digital repositories: PeerJ, PubMed Central and CLOCKSS.

Field expeditions and collections were approved by Ministério do Meio Ambiente (MMA), Instituto Chico Mendes de Conservação da Biodiversidade (ICMBio), Sistema de Autorização e Informação em Biodiversidade (SISBIO) (License number: 57281-2).

## Molecular dating

First, we used only the COI to estimate how long ago *Mishconyx* diverged from its ancestor. We did this because there are more Gonyleptidae sequences of this gene than any other on GenBank. Only one sequence from each species was included, totaling 122 terminal sequences. To set the priors for the BEAST 2.5 analysis (*Bouckaert et al., 2019*), we employed the program BEAUti. We used the Beast Model Test to set the site model, a lognormal relaxed clock with substitution rate of 0.005 (according to *Bragagnolo et al., 2015* and *Peres et al., 2019*) with Yule tree and constrained the root using a normal distribution. In this initial analysis three clades were dated: Gonyleptidae, with $T_{MRCA}$ 140 ± 40 Mya, based on *Sharma & Giribet (2011)*; Sodreaninae *Kury, 2003* clade (*sensu Peres et al., 2019*), with $T_{MRCA}$ 31.5 ± 10 Mya, based on *Peres et al. (2019)*; *Promitobates Rower, 1913*, with $T_{MRCA}$ 58.5 ± 3.9 Mya, based on *Bragagnolo et al. (2015)*. We then ran two independent analyses, with 10 million generations each, sampling trees every 10,000 generations. Both analyses were verified in TRACER 1.7 (*Rambaut et al., 2018*) and checked for EES > 200. The results were combined in LOGCOMBINER 2.5.

Next, we applied the $T_{MRCA}$ estimated for *Mischonyx* to calibrate the multilocus species tree using *BEAST, with the seven genes cited above and the terminals from Table 2, also using BEAST 2.5. We pruned the dataset to one sequence per haplotype per species, used all the priors from the first step and performed two independent analyses with 100 million generations, sampling trees each 5,000 generations. The output from the analyses was checked using Tracer 1.7 and combined trees using LOGCOMBINER 2.5. The maximum clade credibility was annotated and the first 10% was discarded, using TREEANNOTATOR 2.5. The final tree was analyzed using FigTree 1.4.4 (*Rambaut, 2010*).

## Phylogenetic inferences

Three separate analyses were carried as follows: (1) morphological data alone, (2) molecular data alone; and (3) combined molecular and morphological matrixes (Total Evidence Analysis). Each matrix was analyzed using Maximum parsimony (MP) and maximum likelihood (ML). In all analyses, we used *Promitobates ornatus Mello-Leitão, 1922* to root our trees because its is consistent with *Pinto-da-Rocha et al. (2012)* phylogeny, in which this species is the furthest from *Mischonyx* clade, when compared to the other species used as outgroups in our research.

## Bayesian inference

In the morphological analysis (B1), we activated the morph-models package on BEAUti 2.5 and imported the matrix, with the option "add MK morphological data" while importing. The Lewis MK was chosen as the substitution model, and the relaxed log normal clock and fossilized birth and death model were chosen as tree priors.

For analysis using strictly molecular data (B2), the trees for all genes were linked. The Beast Model Test was selected for calculations of the best model for each gene, estimating the mutation rate. The relaxed log normal clock, with the estimates of clock rate for each gene, followed *Bragagnolo et al. (2015)* and *Peres et al. (2019)*. The selected tree model was the Birth and Death model.

The same parameters used for the molecular data analysis were used in the total evidence (TE) (B3). We chose Fossilized Birth and Death Model as the tree prior, with 0.05 as the starting value for the tree diversification rate, with estimation of *Rho* parameter. To estimate the morphological and molecular clock rates we chose the LogNormal distribution.

All Bayesian analyses were carried out on BEAST 2.5, performing two independent analyses, with 100 million generations each, sampling trees every 10,000 generations. We checked the output from the analyses, using Tracer 1.7, checked for EES > 200 and combined trees using LOGCOMBINER 2.5. The maximum clade credibility was annotated and the first 10% was discarded, using TREEANNOTATOR 2.5. The final tree was analyzed using FigTree 1.4.4 (*Rambaut, 2010*).

*Maximum likelihood.* For morphological analysis (ML1), we inserted the dataset as input in the IQ-TREE version 1.6.10 (*Nguyen et al., 2015*), using the best model found by the program, which uses BIC (*Bayesian information criterion*) (*Schwarz, 1978*) to analyze which model is the best for that specific dataset. The analysis displayed by the program is the same described for the molecular data below. To analyze character changes, we inserted the phylogeny output from IQ-TREE on YBIRÁ (*Machado, 2015*).

The DNA sequences were aligned in MAFFT and analyzed with Aliview. The FASTA file contained all the sequences concatenated using SequenceMatrix 1.8 (*Vaidya, Lohman & Meier, 2011*). The analysis was carried out in IQ-TREE version 1.6.10 (*Nguyen et al., 2015*). All the partitions coming from the seven different genes present in the concatenated FASTA file (and the morphological dataset for TE) were first analyzed on IQ-TREE through the partition model (*Chernomor, von Haeseler & Minh, 2016*), using the "-spp" command. The program selected the best substitution model for each gene partition under the BIC (*Schwarz, 1978*), using the program ModelFinder (*Kalyaanamoorthy et al., 2017*), through the command "-m TESTNEWMERGE". Maximum Likelihood analysis was based on 10,000 search iterations, using the command "-s -n 10000". Confidence was measured using bootstrap analysis based on 1,000 iterations of ultrafast bootstrap using the command "-bb 1000" (*Minh, Nguyen & von Haesler, 2013*). The output was analyzed using FigTree 1.4.4 (*Rambaut, 2010*). We used the parsimony method to analyze character changes because, as pointed by *Cheng & Kuntner (2014)*, the aim is to "understand the evolutionary changes of characters rather than the probability of particular ancestral states on the phylogeny".

*Maximum parsimony.* The morphological analysis (MP1) was carried out using TNT (*Goloboff, Farris & Nixon, 2008*). The search was heuristic with TBR branch-swapping (10,000 replicates) while retaining 100 trees per replicate. The command "collapse branches after search" was used to eliminate non-supported nodes, and searches using Ratchet (*Nixon, 1999*) and Tree Fusing (*Goloboff, 1999*). The characters were treated as
unordered and unweighted. To analyze character changes throughout the phylogeny, we used Winclada 1.61.

The molecular (MP2) and TE (MP3) analyses were implemented using the program POY 5.1.1 (*Varón, Vinh & Wheeler, 2010*), which searches using direct optimization (hereafter DO) of unaligned sequences (*Wheeler, 1996*), a strategy referred as Dynamic Homology (*Wheeler, 2001a*, *2001b*). This strategy differs from the traditional static homology search in that the former integrates both alignment and tree searches, while the last treats them as two separated searches. DO is able to test dynamically, in a static matrix, the hypotheses of homology among unaligned nucleotides, optimizing these sequences directly on the available trees and, concomitantly, converting the transformation series of pre-aligned sequences (*Kluge & Grant, 2006*; *Grant & Kluge, 2009*; *Sánchez-Pacheco et al., 2017*).

An exploratory DO analysis was carried out five times, specifying search time (from two to ten hours, totaling 30 hours of search), to check which one yielded the lowest tree scores as outputs and, consequently, the optimal search time for DO ("max_time" parameter). The best tree scores for our dataset were obtained with a maximum search time of 2 h. After that the dataset was analyzed treating H3, COI and CAD sequences as pre-aligned, because they are coding genes, and 28S, 12S, 16S and ITS to be aligned using dynamic homology methods ("transform" command in POY). The program performed five rounds of searches using the "max_time" (with "search" command). In POY each "search" round implements Tree Bisection and Reconnection (TBR), Wagner tree building, Subtree Pruning and Regrafting (SPR), Branch Swapping (RAS+swapping, as in *Goloboff, 1999*), Tree fusing (*Goloboff, 1999*) and Parsimony Ratchet (*Nixon, 1999*). We used the final trees from this previous analysis in an exact iterative pass (IP) analysis (*Wheeler, 2003*). Costs for all the previous optimal trees were calculated and POY generated the implied alignment of this final analysis (*Wheeler, 2003*). TNT 1.5 (*Goloboff & Catalano, 2016*) was used to calculate Bootstrap values and Bremer support, with "hold" command of 10,000,000 trees, "mult" command of 1,000 replicates, holding 10 trees per replicate. Finally, we analyzed the character changes over the optimal tree using parsimony on YBIRÁ (*Machado, 2015*).

## RESULTS

### Molecular data and maximum likelihood models

In total, 54 individuals of *Mischonyx* species were sequenced in this work, encompassing almost all species with two exceptions: *Urodiabunus arlei* and *Mischonyx scaber*. The following fragments were sequenced: 28S (972 bp), 16S (386 bp), 12S (408 bp), CAD (639 bp), COI (570 bp), H3 (309 bp) and ITS (456 bp), totaling 3742 bp for all sequences. Collectivelly, we were able to sequence 88% of the fragments of the 54 exemplar specimens. In this analysis we only included terminal taxa for which we were able to obtain at least five out of the seven sequenced fragments (see Table 2).

The best evolutionary model found under BIC for morphological data was MK+FQ +G4. For 12S rRNA, 16S rRNA, 28S rRNA, CAD, COI, H3 and ITS2, the best models are,

respectively, TIM3+F+I+G4, TMP2u+F+I+G4, TN+F+I, JTDDCMut+G4, mtMAM+I
+G4, DCMut and TIM2+F+I+G4.

## Morphological data

The morphological matrix totals 128 characters, some of which were taken from the
literature and are distributed as follows: 45 characters from the dorsal scutum, 44
characters from the appendages, 6 characters from free tergites, 27 characters from the
male genitalia and two characters from the general habitus.

## List of Morphological Characters and States

1. Dorsal scutum, shape (males) (*Kury & Medrano, 2016*): 0, Gamma P; 1, Gamma R; 2,
   Gamma; 3, Gamma T; 4, Non-Gamma;
2. Dorsal scutum, shape (females) (*Kury & Medrano, 2016*): 0, Alpha; 1, Gamma; 2,
   Gamma T; 3, Gamma P; 4, Non-Gamma;
3. Pedipalp, length: 0, Short (shorter than the dorsal scutum); 1, Long (longer than the
   dorsal scutum);
4. Pedipalp, tibia and tarsus, thickness: 0, Same thickness of femur; 1, Clearly more
   expanded than femur;
5. Dorsal scutum, anterior margin, lateral tubercles (*Mendes, 2011*): 0, Absence; 1,
   Presence;
6. Dorsal scutum, anterior margin, lateral tubercles, number: 0, Three on each lateral; 1,
   Two on each lateral; 2, Four or more on each lateral;
7. Dorsal scutum, anterior margin, lateral tubercles, size: 0, All tubercles with the same
   size; 1, One of the tubercles clearly more developed than the others;
8. Dorsal scutum, frontal hump, elevation: 0, Low (smaller than the ocularium height,
   without considering the median armature); 1, Elevated (bigger than the ocularium
   height, without considering the median armature) (Figs. 1–9);
9. Dorsal scutum, frontal hump, tubercles: 0, Absent; 1, Present;
10. Dorsal scutum, frontal hump, tubercles, number: 0, One (single armature); 1, Two (one
    pair) (Fig. 4C); 2, Four (2 pairs);
11. Dorsal scutum, number of areas: 0, Three; 1, Four;
12. Dorsal scutum, ocularium, median armature: 0, Absent; 1, Present;
13. Dorsal scutum, ocularium, median armature, number: 0, One; 1, Two (one pair)
    (Figs. 1–9); 2, Three pairs;
14. Dorsal scutum, ocularium, median armature, size: 0, Tubercle (smaller than the
    ocularium height) (Fig. 1D); 1, Spine (longer than the ocularium height) (Fig. 4C);
15. Dorsal scutum, ocularium, median armature, merge: 0, Not merged (Figs. 1–9); 1, Apex
    merged;
16. Dorsal scutum, ocularium, anterior granule: 0, Absent (Fig. 7D); 1, Present (Fig. 1C);

17. Dorsal scutum, ocularium, posterior granulation: 0, Absent (Fig. 2D); 1, Present (Fig. 3C);

18. Dorsal scutum, prosoma, lateral granulation: 0, Absent 1, Present (Fig. 1A);

19. Dorsal scutum, prosoma, posterior armature: 0, Absent; 1, Present;

20. Dorsal scutum, prosoma, posterior armature, number: 0, Pair of tubercles (Figs. 1–9); 1, Several tubercles;

21. Dorsal scutum, mid-bulge, lateral margin, armature: 0, Absent; 1, Present;

22. Dorsal scutum, mid-bulge, lateral margin, armature distribution: 0, Present in the whole extension (Fig. 2B); 1, Present on the posterior half only (Fig. 3B);

23. Dorsal scutum, mid-bulge, lateral margin, armature, size: 0, Large tubercles (Fig. 9A); 1, Small tubercles (Fig. 2C);

24. Dorsal scutum, mid-bulge, lateral margin, armature, shape: 0, Rounded (Figs. 1–9); 1, Pointed;

25. Dorsal scutum, mid-bulge, lateral margin, armature, color (in ethanol): 0, Clearer than the rest of the body (Fig. 9A); 1, Darker than the rest of the body (Fig. 7A); 2, Same color of the rest of the body (Fig. 1B);

26. Dorsal scutum, mid-bulge, lateral margin, posterior armature, merge: 0, Merged, forming large tubercles (Fig. 9A); 1, Not merged (Fig. 2B);

27. Dorsal scutum, area I, longitudinal groove: 0, Absent; 1, Present;

28. Dorsal scutum, area I, paired median armature: 0, Absent; 1, Present;

29. Dorsal scutum, area I, paired median armature, size: 0, Small tubercles (Fig. 2B); 1, Conspicuous tubercles (Fig. 1B);

30. Dorsal scutum, area I, paired median armature, color (in ethanol): 0, Clearer than the rest of the body (Fig. 1B); 1, Darker than the rest of the body (Fig. 1A); 2, Same color of the rest of the body;

31. Dorsal scutum, area I, paired median armature, length in comparison to median armatures of area III: 0, Larger than the median armatures from area III (Fig. 1B); 1, Smaller than the median armatures from area III (Fig. 1A); 2, Same size of the median armatures from area III;

32. Dorsal scutum, area II, paired median armature: 0, Absent; 1, Present;

33. Dorsal scutum, area II, lateral tubercle: 0, Absent (Fig. 6B); 1, Present (Fig. 3A);

34. Dorsal scutum, area II, paired median armature, color (in ethanol): 0, Paler than the rest of the body (Fig. 5A); 1, Darker than the rest of the body (Fig. 4A); 2, Same color of the rest of the body;

35. Dorsal scutum, area II, paired median armature, size in comparison to median armatures of area III: 0, Larger than the median armatures from area III (Fig. 5A); 1, Smaller than the median armatures from area III (Fig. 4A); 2, Same size of the median armatures from area III;

36. Dorsal scutum, area III, armature: 0, Absent; 1, Present;

37. Dorsal scutum, area III, median armature, number: 0, One pair; 1, Single;

38. Dorsal scutum, area III, paired median armature, color (in ethanol): 0, Paler than the rest of the body (Fig. 6A); 1, Darker than the rest of the body (Fig. 5B); 2, Same color of the rest of the body;

39. Dorsal scutum, area III, paired median armature, form: 0, Rounded; 1, Elliptic (Fig. 5B); 2, Sharp (Fig. 1D);

40. Dorsal scutum, area III, elliptic paired median armature: 0, Slightly compressed laterally (Fig. 5B); 1, Strongly compressed laterally (Fig. 9A);

41. Dorsal scutum, area III, lateral tubercles: 0, Absent; 1, Present (Fig. 9A);

42. Dorsal scutum, area III, lateral armature, size: 0, Small tubercles (Fig. 3A); 1, Well-developed tubercles (Fig. 9A);

43. Dorsal scutum, area III, lateral armature, color (in ethanol): 0, Clearer than the rest of the body (Fig. 6B); 1, Darker than the rest of the body (Fig. 9A); 2, Same color of the rest of the body (Fig. 1B);

44. Dorsal scutum, area III, lateral armature, form: 0, Rounded (Fig. 3A); 1, Elliptic (Fig. 5B);

45. Dorsal scutum, posterior margin, armature: 0, Absent; 1, Present;

46. Dorsal scutum, posterior margin, armature, size: 0, Small tubercles (Fig. 1A); 1, Presence of central tubercle more developed or apophysis (Fig. 9B); 2, All tubercles well-developed;

47. Dorsal scutum, granulation, density (*DaSilva & Pinto-da-Rocha, 2010*): 0, Low (scattered granules, some regions of dorsal scute smooth); 1, Median (granules scattered throughout dorsal scute); 2, High;

48. Free tergite I, armature: 0, Absent; 1, Present;

49. Free tergite I, armature, size: 0, Small tubercles (Fig. 1A); 1, Presence of central tubercle more developed or apophysis (Fig. 9B); 2, All tubercles well-developed;

50. Free tergite II, armature: 0, Absent; 1, Present;

51. Free tergite II, armature, size: 0, Small tubercles (Fig. 1A); 1, Presence of central tubercle more developed or apophysis (Fig. 6B); 2, All tubercles well-developed;

52. Free tergite III, armature: 0, Absent; 1, Present;

53. Free tergite III, armature, size: 0, Small tubercles (Fig. 1A); 1, Presence of central tubercle more developed or apophysis (Fig. 6B); 2, All tubercles well-developed;

54. Leg II, basitarsus, segmentation, number: 0, Six; 1, Seven; 2, Eight; 3, Nine; 4, more than nine;

55. Leg III, trochanter, armature: 0, Absent; 1, Present;

56. Leg III, trochanter, armature, type: 0, Trochanter with many tubercles; 1, Trochanter with a prolateral basal apophysis;

57. Leg IV, coxa, apical width of males in ventral view (compared to coxa III) (modified from *Benedetti & Pinto-da-Rocha, 2019*): 0, Coxae III and IV with the same width; 1, Coxa IV 2 times larger than coxa III; 2, Coxa IV 4 times larger than coxa III;

58. Leg IV, coxa, apical prolateral apophysis on males: 0, Absent; 1, Present;

59. Leg IV, coxa, apical prolateral apophysis, length (compared to trochanter IV) (modified from *Benedetti & Pinto-da-Rocha, 2019*): 0, Shorter than trochanter IV (Fig. 3B); 1, Similar size of trochanter IV (Fig. 3A); 2, Longer than trochanter IV; 3, Much smaller than trochanter IV (as a tubercle);

60. Leg IV, coxa, apical prolateral apophysis, basal tubercle: 0, Absent; 1, Present (Fig. 2B);

61. Leg IV, coxa, apical prolateral apophysis, secondary subdistal lobe (*Benedetti & Pinto-da-Rocha, 2019*): 0, Absent; 1, Present (Fig. 4A);

62. Leg IV, coxa, apical prolateral apophysis, direction in dorsal view (*Benedetti & Pinto-da-Rocha, 2019*): 0, Slightly inclined relative to the axis of the base of coxa IV (Fig. 4A); 1, Transversal; 2, Oblique (Fig. 3B);

63. Leg IV, coxa, apical prolateral apophysis, apex width (modified from *Benedetti & Pinto-da-Rocha, 2019*): 0, Base more than 4 times larger than the apex (Fig. 2B); 1, Base 2 times larger than the apex (Fig. 9B); 2, Base as large as the apex;

64. Leg IV, coxa, apical prolateral apophysis, thickness: 0, Robust (Fig. 5B); 1, Sharp (Fig. 5A);

65. Leg IV, coxa, apical prolateral apophysis in females (*Benedetti & Pinto-da-Rocha, 2019*): 0, Absent; 1, Smaller than the male;

66. Leg IV, coxa, apical retrolateral apophysis in males (*Benedetti & Pinto-da-Rocha, 2019*): 0, Absent; 1, Present (Fig. 8B);

67. Leg IV, coxa, apical retrolateral apophysis, size (*Benedetti & Pinto-da-Rocha, 2019*): 0, Tubercle; 1, Apophysis;

68. Leg IV, coxa, apical retrolateral apophysis, number of branches: 0, One; 1, Two;

69. Leg IV, trochanter, prolateral armature in males: 0, Absent; 1, Present;

70. Leg IV, trochanter, retrolateral apical armature: 0, Absent; 1, Present;

71. Leg IV, trochanter, retrolateral apical armature, size: 0, Tubercle; 1, Apophysis (Fig. 1B);

72. Leg IV, trochanter, retrolateral armature, number: 0, One (Fig. 4B); 1, Two (Fig. 7B); 2, Three (forming a line);

73. Leg IV, femur, thickness: 0, Short and robust (Fig. 5B); 1, Long and thin (Fig. 3B);

74. Leg IV, femur, prolateral curvature: 0, Straight (not curved) (Fig. 3B); 1, Curved (Fig. 6B);

75. Leg IV, femur, retrolateral basal apophysis: 0, Absent; 1, Present (Fig. 2D);

76. Leg IV, femur, dorso-basal apophysis (DBA) (*Benedetti & Pinto-da-Rocha, 2019*): 0, Absent; 1, Present (Fig. 2D);

77. Leg IV, femur, dorso-basal apophysis, size: 0, Small (Fig. 8D); 1, large (longer than larger) (Fig. 2D); 2, Very small (Tubercle) (Fig. 1D);

78. Leg IV, femur, dorso-basal apophysis, apex direction: 0, Apex anteriorly directed (Fig. 9B); 1, Apex dorsally directed (Fig. 5D); 2, Apex retrolaterally directed (Fig. 6B); 3, Apex prolaterally directed;

79. Leg IV, femur, dorso-basal apophysis, apex width: 0, Base more than 4 times wider than apex (Fig. 2D); 1, Base 2 times wider than apex (Fig. 9B); 2, Base as wide as apex (Fig. 8D);

80. Leg IV, femur, dorso-basal apophysis, shape: 0, Digitiform (Fig. 6C); 1, Falciform (Fig. 7D); 2, Blunt; 3, Branched (Fig. 9B); 4, Conic (Fig. 2D);

81. Leg IV, femur, branched dorso-basal apophysis, larger branch: 0, Retrolateral (Fig. 6B); 1, Dorsal (Fig. 4C);

82. Leg IV, femur, prolateral row of tubercles in males: 0, Absent; 1, Present;

83. Leg IV, femur, prolateral row of tubercles, development: 0, Equally developed (Fig. 9A); 1, Median larger (Fig. 6B); 2, Apical larger (Fig. 6A);

84. Leg IV, femur, prolateral row of tubercles, single apical apophysis: 0, Absent; 1, Present (Fig. 3B);

85. Leg IV, femur, dorsal row of tubercles: 0, Absent (dorsally smooth) (Fig. 3D); 1, Present (Fig. 2C);

86. Leg IV, femur, dorsal row of tubercles, apophysis after DBA: 0, Absent (Fig. 2D); 1, Present (Fig. 2C);

87. Leg IV, femur, dorsal row of tubercles, apophysis after DBA, number: 0, One (Fig. 5D); 1, Two (Fig. 4C); 2, Three–Six (Fig. 2C); 3, More than six;

88. Leg IV, femur, row of tubercles between the dorsal and retrolateral lines: 0, Absent; 1, Present;

89. Leg IV, femur, retrolateral row of tubercles: 0, Absent; 1, Present;

90. Leg IV, femur, retrolateral row of tubercles, position of the larger apophysis: 0, Basal third; 1, Medial third (Fig. 9A); 2, Apical Third (Fig. 5B);

91. Leg IV, femur, retrolateral row of tubercles, number of apophysis on the basal half: 0, Absence of apophysis on the basal half) (Fig. 3B); 1, One (Fig. 3A); 2, Two (Fig. 4A); 3, Three–Six (Fig. 6A); 4, More than 6;

92. Leg IV, femur, retrolateral row of tubercles, median apophysis: 0, Absent (Fig. 6B); 1, Present (Fig. 4A);

93. Leg IV, femur, retrolateral row of tubercles, number of apophysis on the apical half: 0, Absence of apophysis on the apical half; 1, One (Fig. 7D); 2, Two (Fig. 1A); 3, Three–Six (Fig. 3B); 4, More than 6;

94. Leg IV, femur, retrolateral row of tubercles, more developed apical tubercle: 0, Absent; 1, Present (Fig. 1B);

95. General body color (in ethanol): 0, Brownish; 1, Black; 2, Yellowish; 3, Reddish;

96. Body totally or partially covered with debris (*DaSilva & Pinto-da-Rocha, 2010*): 0, Absent; 1, Present;

97. Penis, ventral plate, form in lateral view: 0, Globose (Fig. 10E); 1, Thin (Fig. 11E);

98. Penis, ventral plate, form in dorsal view: 0, Longer than larger (thin) (Fig. 11D); 1, Larger than longer (developed lateral expansions) (Fig. 11A);

99. Penis, ventral plate, ventral side, T1 microsetae: 0, Absent; 1, Present;

100. Penis, ventral plate, ventral side, T1 microsetae, distribution: 0, Sparse or present in some regions (Fig. 10F); 1, Presence in the whole extension (Fig. 12C);

101. Penis, ventral plate, ventral side, medio-apical excavation: 0, Absent; 1, Present;

102. Penis, ventral plate, ventral side, degree of the medio-apical excavation: 0, Slightly excavated (Fig. 10C and 10I); 1, Very excavated (Fig. 12F);

103. Penis, ventral plate, apical cleft (*Kury, 1992*): 0, Absent; 1, Present;

104. Penis, ventral plate, apical cleft, depth: 0, Shallow (in dorsal view, reaches at most the line of the first MS C) (Fig. 11D); 1, Deep (in dorsal view it is more basal than the MS C) (Fig. 13G);

105. Penis, ventral plate, apical cleft, format: 0, Edges slightly sloped (Fig. 13A); 1, Edges very sloped (Fig. 13G);

106. Penis, ventral plate, Macrosetae C (MS C), number: 0, Two; 1, Three (Fig. 11D); 2, Four;

107. Penis, ventral plate, Macrosetae C (MS C), shape: 0, Straight; 1, Helicoidal (Fig. 10G); 2, Curved (Fig. 11D);

108. Penis, ventral plate, Macrosetae C (MS C), position: 0, Distal (Fig. 13A); 1, Sub-distal (Fig. 11D);

109. Penis, ventral plate, Macrosetae A (MS A), number: 0, Two (Fig. 12D); 1, Three (Fig. 14G); 2, Four (Fig. 13A);

110. Penis, ventral plate, Macrosetae A (MS A), position on the ventral plate: 0, Linear in dorso-ventral direction (Fig. 10A); 1, Triangle shaped (Fig. 12D); 2, Parable shaped (Fig. 10H); 3, Linear in baso-apical direction;

111. Penis, ventral plate, Macrosetae B (MS B), size: 0, Small (clearly smaller than the MS A) (Fig. 14B); 1, Large (same size of the MS A) (Fig. 14G);

112. Penis, ventral plate, Macrosetae D (MS D): 0, Absent (Fig. 12E); 1, Present (Fig. 13H);

113. Penis, ventral plate, Macrosetae D (MS D), number: 0, One (Fig. 13H); 1, Two; 2, Three;

114. Penis, ventral plate, Macrosetae D (MS D), size: 0, Small (Fig. 13H); 1, Large (Fig. 14B);

115. Penis, ventral plate, Macrosetae D (MS D), position in lateral view: 0, Ventral to the MS C (Fig. 10A); 1, Dorsal to the MS C;

116. Penis, ventral plate, Macrosetae E (MS E): 0, Absent; 1, Present;

117. Penis, ventral plate, Macrosetae E (MS E), number: 0, One; 1, Two;

118. Penis, ventral plate, Macrosetae E (MS E), position of the most basal MS E: 0, Ventral and aligned to the MS C (Fig. 11B); 1, Ventral and medial to the MS C (Fig. 13H);

119. Penis, ventral plate, well-developed lateral lobes (modified from *Kury, 1992*): 0, Absent (Fig. 11D); 1, Present (Fig. 11A);

120. Penis, ventral plate, lateral lobes, position: 0, Medial (Fig. 11A); 1, Basal (Fig. 11D);

121. Penis, ventral process: 0, Absent; 1, Present;

122. Penis, ventral process, flabellum: 0, Absent; 1, Present;

123. Penis, ventral process, flabellum, shape: 0, As long as large (Fig. 12A); 1, Longer than wide (thin) (Fig. 12D);

124. Penis, ventral process, flabellum, lateral parts: 0, Serrated (Fig. 10A); 1, Smooth (Fig. 13G);

125. Penis, ventral process, flabellum, apex: 0, Without a longer central terminal; 1, With a longer central terminal (Fig. 10H);

126. Penis, stylus, apex, microsetae: 0, Absent (Fig. 13D); 1, Present (Fig. 12B);

127. Penis, stylus, apex, format: 0, Inclined relative to the penis axis; 1, Straight;

128. Penis, stylus, apex, keel: 0, Absent; 1, Present.

## Geographical distribution and areas of endemism

The geographical distribution of all *Mischonyx* species is depicted in Figs. 15, S1 and S2. All species occur only in Brazil, from the state of Santa Catarina to the state of Espirito Santo, throughout the Atlantic Forest and in two in Cerrado areas (*e.g.*, Minas Gerais and Mato Grosso do Sul). The species that occur in Cerrado areas are *M. intermedius* and *M. squalidus*. In general, all species are restricted to a narrow range, with the exception of *M. anomalus*, which occurs in the entire state of Paraná, and *M. squalidus*, which is widespread where other species of the genus occur and also in areas where no other species are present, for example the state of Espírito Santo. *M. squalidus* is synanthropic (*Mestre & Pinto-da-Rocha, 2004*) and can be found in degraded areas where *Pinus* is grown, in pasture areas and even in cities.

Regarding the Areas of Endemism (AoE) proposed by *DaSilva, Pinto-da-Rocha & Morrone (2017)*, most species are endemic/restricted to one AoE. The only exception is *M. squalidus. Mischonyx reitzi* **comb. nov.** and *M. clavifemur* **comb. nov.** are restricted to SC AoE; *M. anomalus* is restricted to PR AoE; *M. intervalensis* **sp. nov.** is restricted to SSP; *M. insulanus* and *M. kaisara* are restricted to SMSP; *M. processigerus* is restricted to Boc; *M. fidelis, M. scaber* and *M. parvus* **comb. nov.** are restricted to LSRJ; *M. arlei* **comb. nov.**, *M. spinifrons* **comb. nov.**, *M. minimus* **sp. nov.**, *M. tinguaensis* **sp. nov.** and *M. poeta* are restricted to Org. Clearly, the AoE with more endemic species is Org. The locality of each species, plotted on the map of Figs. 15, S1 and S2, are in different colors, each representing one different AoE.

## Phylogenetic analyses

### Morphological analyses

In all analyses using morphological data alone, under the maximum likelihood (hereon ML1, Fig. S3), under Bayesian (hereon B1, Fig. S4) and under maximum parsimony (heron MP1, Fig. S5) criteria, the first lineage branching off inside the *Mischonyx* clade is composed of *M. arlei* **comb. nov.**, *M. minimus* **sp. nov.** and *M. intermedius*, followed by the divergence of *G. antiquus* (former *Mischonyx antiquus*, before this work). The only difference is that, in B1, *Multumbo* species are in a clade with *M. intermedius, M. minimus* and *M. arlei*. Moreover, all analyses recover the clade formed by *M. anomalus, M. clavifemur* **comb. nov.** and *M. reitzi* **comb. nov.**, consistent with the results of the

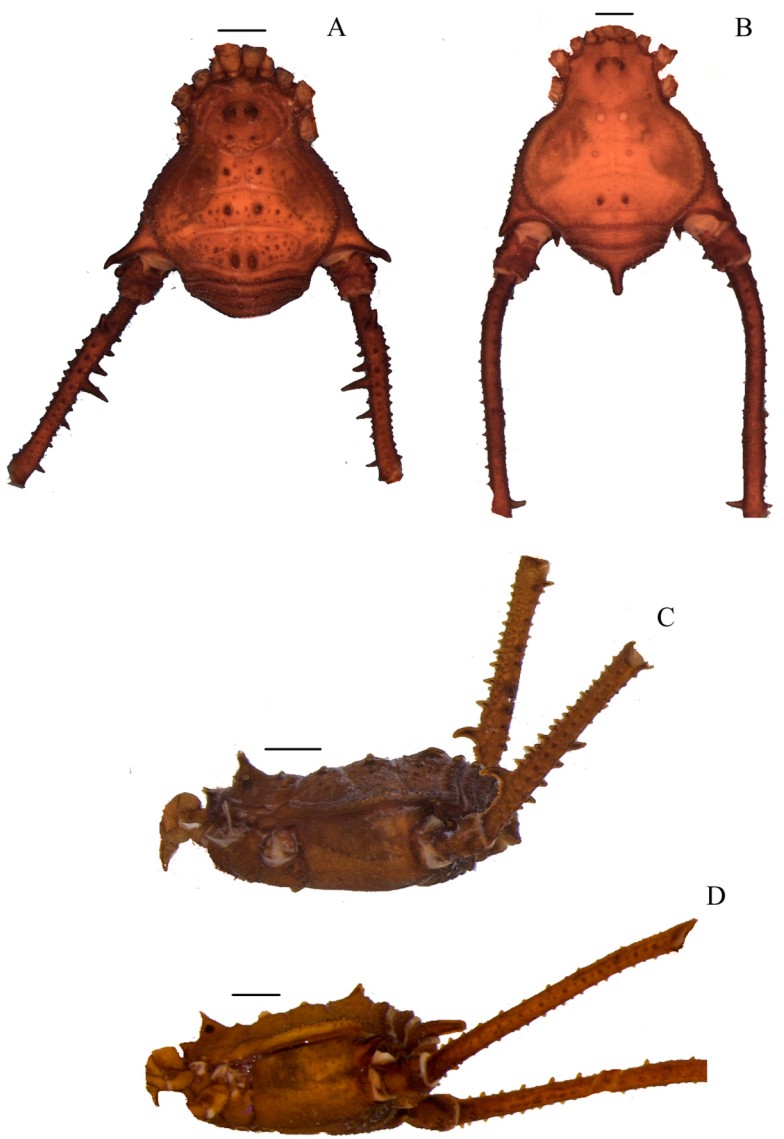

**Figure 1** *Mischonyx anomalus* and *Mischonyx arlei* holotypes. (A & C) *Mischonyx anomalus*, dorsal and lateral views, respectively. (B & D) *Mischonyx arlei*, dorsal and lateral views, respectively. Scale bars: 1 mm.                                                                      

molecular and TE analyses (Figs. 16–18 and S6–S12). B1 is the only analysis that places both *Multumbo* species within the *Mischonyx* clade. The results of ML1 and MP1 agree with our TE results (see below). All analyses were weakly supported by Bootstrap. The bootstrap values obtained for the *Mischonyx* clade is 25 in ML1 and 7 in MP1. All internal branches inside the genus have values below 50 in both analyses (Figs. S3 and S5). In B1, the posterior probability of the *Mischonyx* clade was 0.872 and most nodes inside the genus have posterior probabilities lower than 0.6.

## Molecular analyses

In all analyses using molecular data alone, *Mischonyx* is monophyletic if *G. antiquus* (former *Mischonyx antiquus*) is removed from the genus: under maximum likelihood

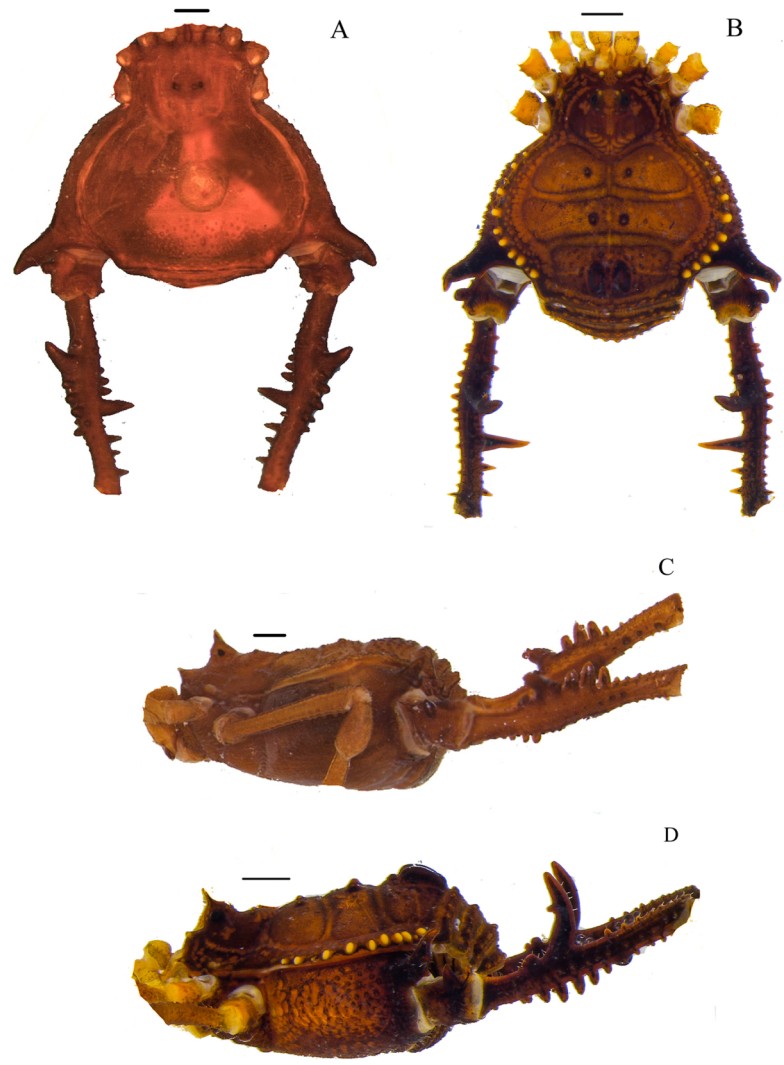

**Figure 2** ***Mischonyx clavifemur* holotype and *Mischonyx fidelis* (4115A).** (A & C) *Mischonyx clavifemur*, dorsal and lateral views, respectively. (B & D) *Mischonyx fidelis*, dorsal and lateral views, respectively. Scale bars: 1 mm.

(hereon ML2, Fig. S6), Bayesian (hereon B2, Fig. S7) and maximum parsimony (hereon MP2, Fig. S8). However, in MP2, there is a clade formed by *Deltaspidium* and *Multumbo* species, which is inside the clade that holds all the other *Mischonyx* species. These other genera are inside the clade with species from SMSP, SSP, PR and SC AoE. This group is sister to another clade with the remaining species of *Mischonyx*, which are from Boc, Esp, LSRJ and Org AoE.

ML2 and B2 differ from MP2 in that the species of *Deltaspidium* and *Multumbo* are recovered inside the clade with all *Mischonyx* species. The only difference between ML2 and B2 is the position of *M. poeta*. While in ML2 this lineage is basal in the clade with species from Org and LSRJ, in B1 it is sister to *M. spinifrons* **comb. nov.** Besides this difference, the main relationships inside *Mischonyx* are the same as found in MP2: a clade

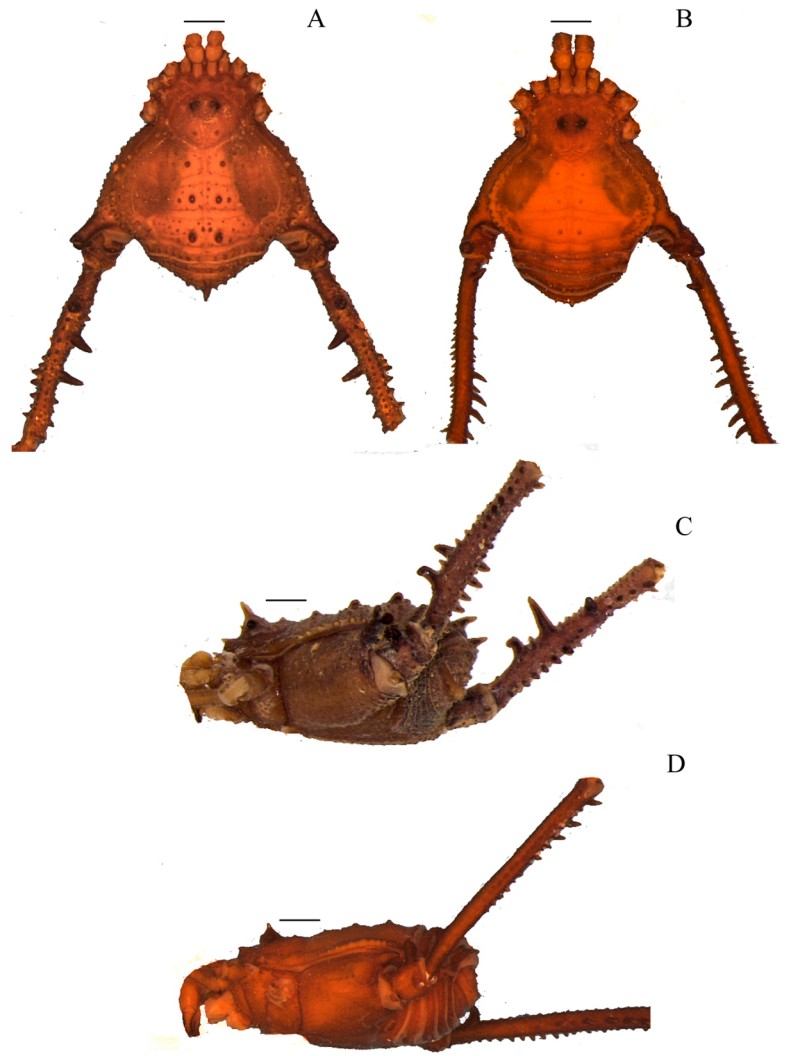

**Figure 3** ***Mischonyx insulanus*** and ***Mischonyx intermedius*** **holotypes.** (A & C) *Mischonyx insulanus*, dorsal and lateral views, respectively. (B & D) *Mischonyx intermedius*, dorsal and lateral views, respectively. Scale bars: 1 mm.

with species from SMSP, SSP, PR and SC AoE sister to the lineage with species from Boc, Esp, LSRJ and Org AoE.

The support values were high in all three analyses: Bootstrap (in ML2 and MP2), Bremer (in MP2) and posterior probability (in B2). The bootstrap value for the *Mischonyx* clade in ML2 was 92 and in MP2 it was 100. In MP2, the node with the lowest bootstrap value is the one holding *Deltaspidium*, *Multumbo* and some *Mischonyx* species (cited above). In ML2, the lowest value inside the *Mischonyx* clade is 67 (Figs. S6 and S8). All posterior probabilities inside the genus are higher than 0.9, except for two nodes, which have values above 0.6.

### Molecular dating
The Bayesian analysis (henceforward abbreviated as BM, fig. 16) generally corroborates the topologies obtained from the other molecular analyses, except for the position of *M. poeta*.

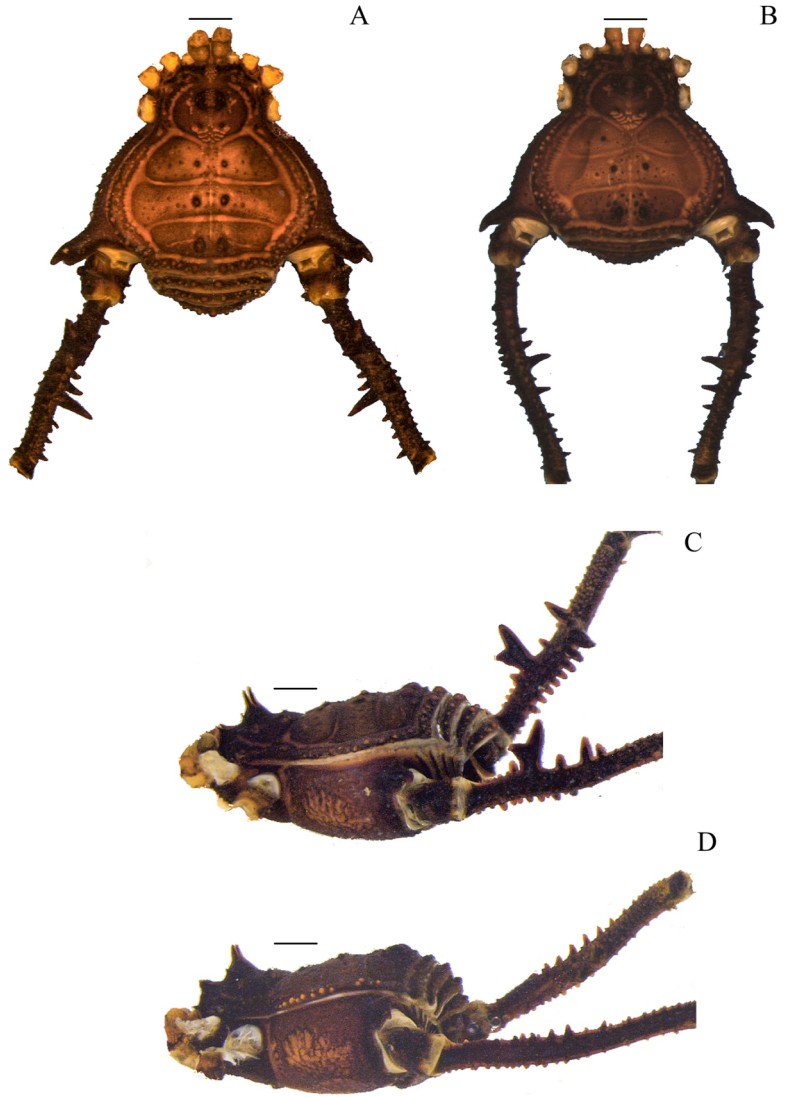

**Figure 4 *Mischonyx intervalensis* sp. nov. holotype and *Mischonyx kaisara*.** (A & C) *Mischonyx intervalensis* **sp. nov.**, dorsal and lateral views, respectively. (B & D) *Mischonyx kaisara*, dorsal and lateral views, respectively. Scale bars: 1 mm.               

While in the results of BM this species is sister to *M. spinifrons* **comb. nov.**, in ML2 it is sister to a larger clade that includes *M. spinifrons* **comb. nov.**, *M. fidelis*, *M. parvus* **comb. nov.** and *M. squalidus*. The more inclusive clades have the same composition and same relationships in BM and ML2: one clade including the species from LSRJ, Boc, Org and SEsp AoE and another with species from SMSP, SSP, PR and SC AoE. The main divergence time of the *Mischonyx* clade occurred at 50.53 Mya (95% HPD = 44.07–57.12), when the two speciose clades split. The first split time inside these two clades is very similar: 48.94 Mya (95% HPD = 39.65–54.60), for the one holding species from SMSP, SSP, PR and SC AoE and 44.80 (95% HPD = 35.57–52.32) for the other clade. Within the former clade, the lineage containing species in the SSP, PR and SC areas of endemism formed approximately at 28 Mya. The main divergence time after the divergence of

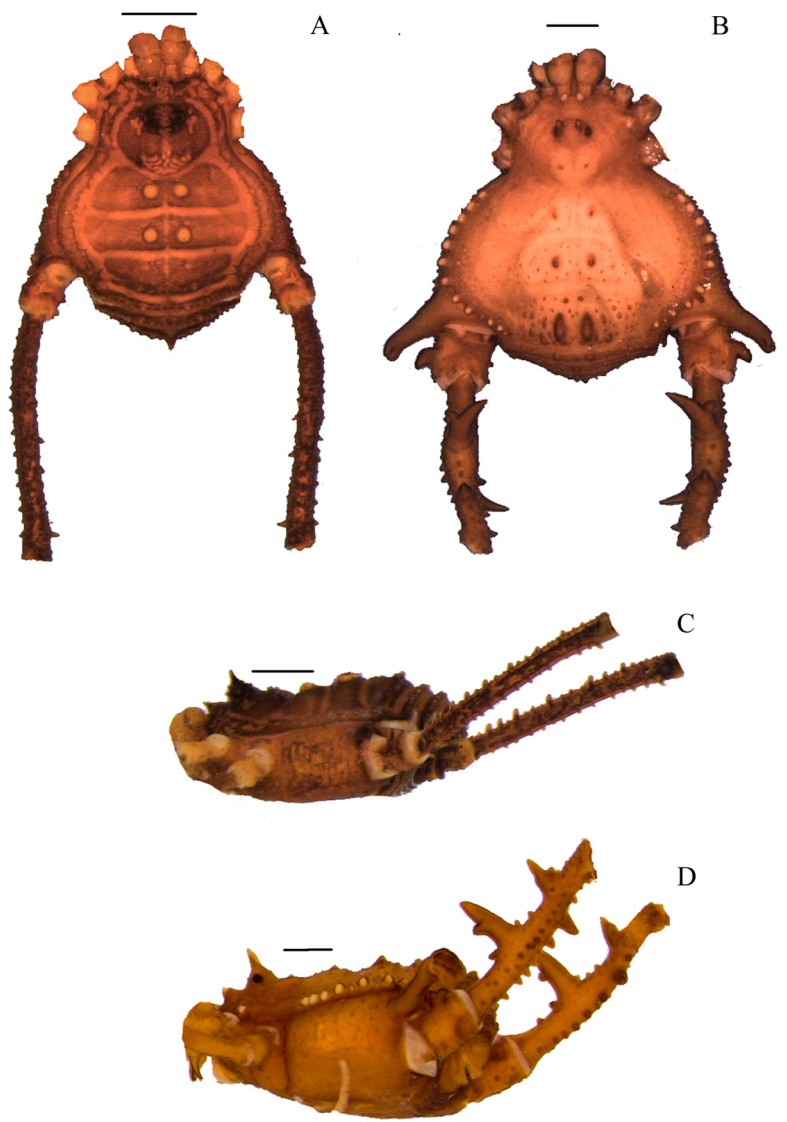

**Figure 5 *Mischonyx minimus* sp. nov. and *Mischonyx parvus* holotypes.** (A & C) *Mischonyx minimus* **sp. nov.**, dorsal and lateral views, respectively. (B & D) *Mischonyx parvus*, dorsal and lateral views, respectively. Scale bars: 1 mm.

*M. intermedius* from the remaining species of the clade occurred at 34.24 Mya (95% HPD = 27.07–41.38).

### Total evidence analyses

All TE analyses, under maximum likelihood (hereon ML3, Fig. 17), Bayesian (hereon B3, Figs. 18B, S9 and S10) and maximum parsimony (heron MP3, Fig. 18A, S11 and S12), yielded very similar results. *G. antiquus* (former *Mischonyx antiquus*) is placed outside *Mischonyx*. Inside *Mischonyx*, there are two major clades. One contains species of SMSP AoE, as sister to the clade containing species from SSP, PR and SC AoE. The other, with a clade holding *M. intermedius* as sister to *M. arlei* **comb. nov.** and *M. minimus* **sp. nov.** and this lineage as sister to the clade which contains species from Boc, LSRJ and Org AoE.

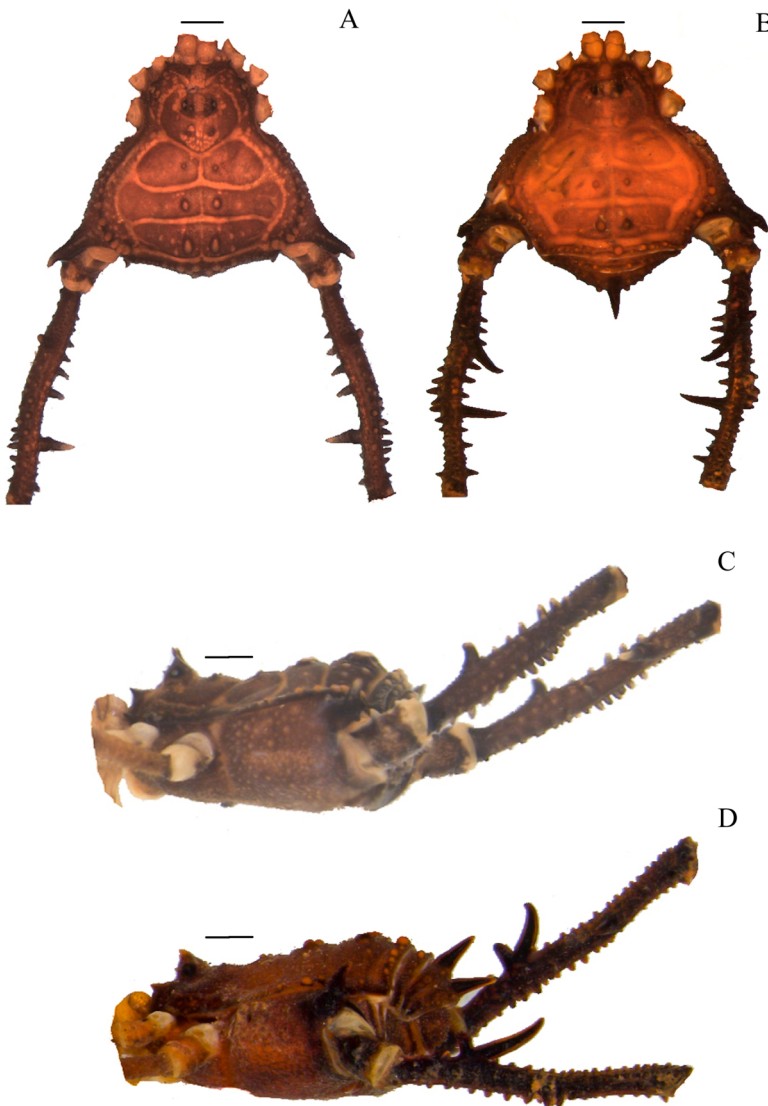

**Figure 6 *Mischonyx poeta* and *Mischonyx processigerus* paratypes.** (A & C) *Mischonyx poeta*, dorsal and lateral views, respectively. (B & D) *Mischonyx processigerus*, dorsal and lateral views, respectively. Scale bars: 1 mm.

Inside this last clade, there are some differences among the analyses. In MP3, the species from LSRJ + *M. squalidus* form a clade sister to species from Org (excepting *M. arlei* **comb. nov.** and *M. minimus* **sp. nov.** which have already diverged). In ML3, two species from Org (*M. poeta* and *M. scaber*) branches off in a clade, followed by *M. spinifrons* **comb. nov.**, which is sister to the lineage containing the species from LSRJ + *M. squalidus*. In B3, there are two clades with these species: *M. poeta* + *M. spinifrons* **comb. nov.** and the other with LSRJ species, *M. scaber* and *M. tinguaensis* **sp. nov.**. MP3 and ML3 have bootstrap values over 50 for inner branches inside *Mischonyx*. Bootstrap values for *Mischonyx* node are 89 in ML3 and 81 in MP3. Bremer support in MP3 for *Mischonyx* clade is 4 (Fig. 18A). Posterior probabilities inside the genus are higher than 0.6 and *Mischonyx* clade posterior probability is 0.971 (Fig. 18B).

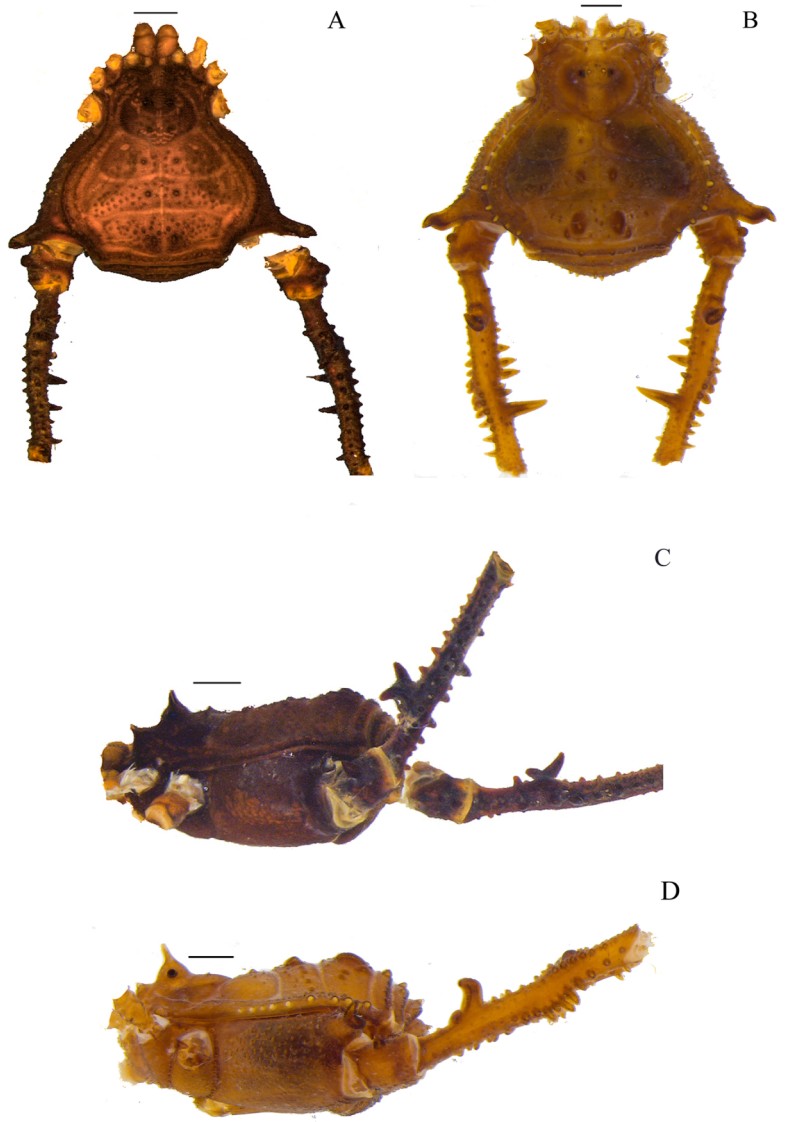

**Figure 7 *Mischonyx reitzi* (0672) and *Mischonyx scaber*.** (A & C) *Mischonyx reitzi*, dorsal and lateral views, respectively. (B & D) *Mischonyx scaber*, dorsal and lateral views, respectively. Scale bars: 1 mm.

Henceforward, we are going to consider ML3 as our working phylogeny to present the further results regarding character state changes and to discuss relationships and character evolution.

### Character changes through ML3

In ML3, *Mischonyx* is supported by the following character changes: Lateral tubercles on anterior margin of dorsal scutum subequal in size (#7-0), elliptic tubercles on area III (#39-1), absence of prolateral apophysis in females (#65-0), femur prolaterally curved (#74-1), three to six apophyses on apical half of retrolateral row on femur IV (#93-3) and general body color brown (#95-0). The clade with species from SMSP, SSP, PR and SC AoE is supported by the presence of median apophysis on retrolateral row of femur IV (#92-1).

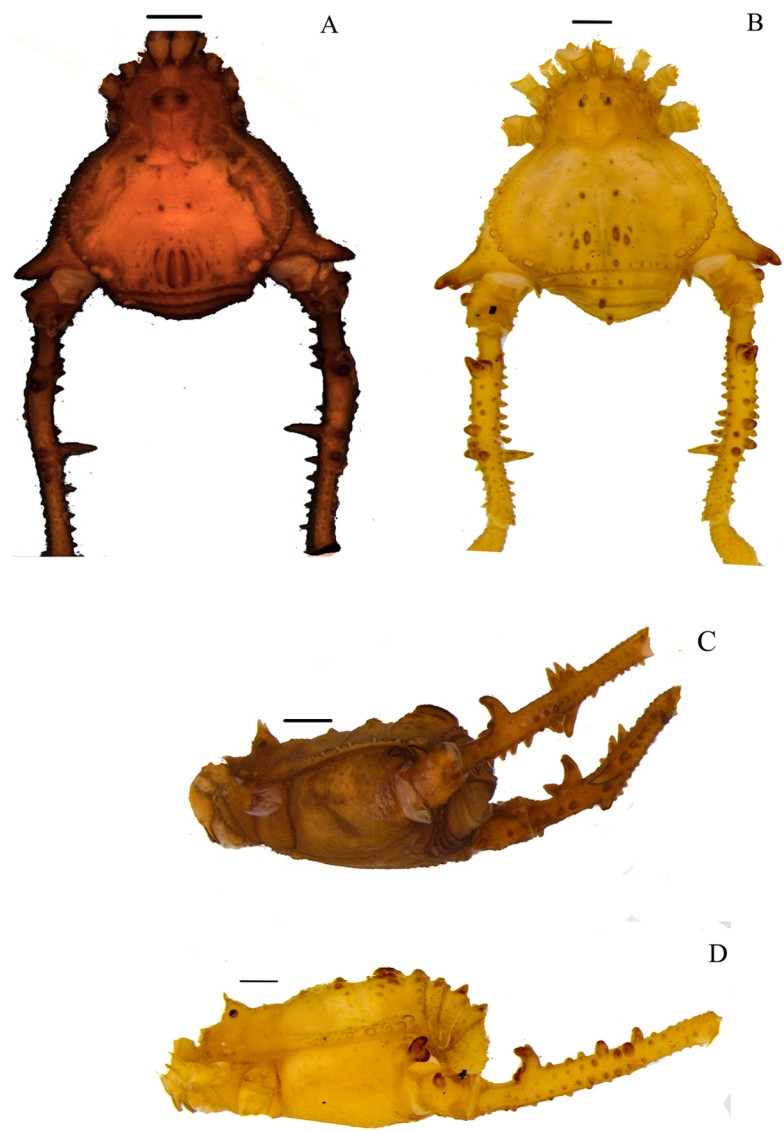

**Figure 8** *Mischonyx spinifrons* (*M. bresslaui* **paratype)** and *Mischonyx squalidus* (*M. cuspidatus* **holotype).** (A & C) *Mischonyx spinifrons*, dorsal and lateral views, respectively. (B & D) *Mischonyx squalidus*, dorsal and lateral views, respectively. Scale bars: 1 mm.

Inside this clade, the lineage with species from SMSP is supported by basitarsus II with nine segments (#54-3) and falciform DBA (#80-1). The clade containing species from SSP, PR and SC is supported by median armature on ocularium longer than the high (#14-1), small tubercles on free tergite II (#51-0), thin ventral plate (#98-0) and MSA forming a parable (#110-2). The group with species from PR and SC is supported by (#25-1), (#47-2), retrolateral apophysis on trochanter IV (#71-1), two apophyses on apical half on retrolateral row of femur IV (#93-2) and ventral plate thin in lateral view (#97-1).

The other lineage inside the clade, with species from Boc, Esp, LSRJ, and Org, is supported by flabellum as long as large (#123-0). Inside this clade, the lineage formed by *M. arlei* **comb. nov.**, *M. intermedius* and *M. minimus* **sp. nov.** is supported by median

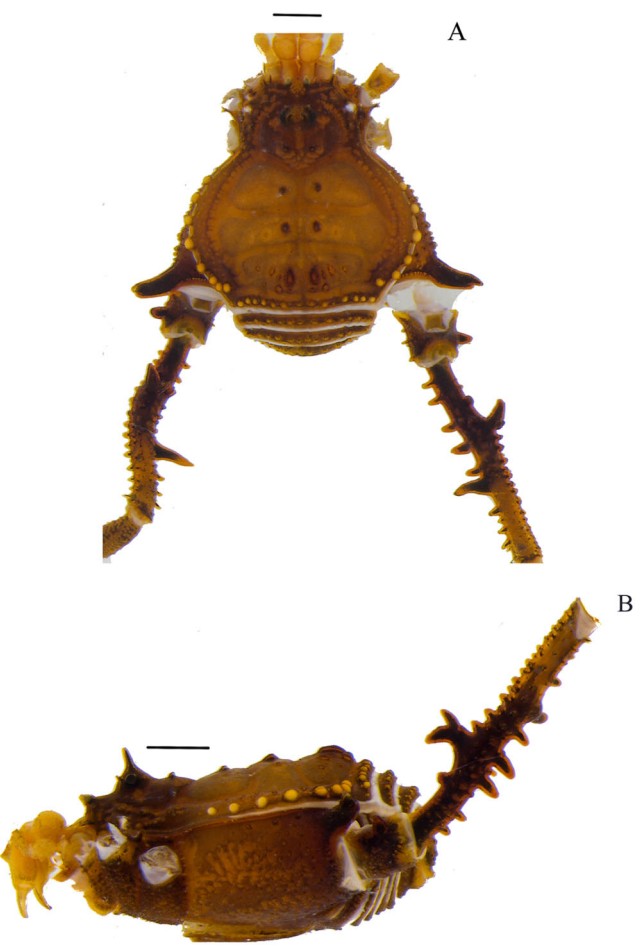

**Figure 9 *Mischonyx tinguaensis* sp. nov. holotype.** (A) Dorsal view. (B) Lateral. Scale bars: 1 mm.

armature on area I larger than median armatures on area III (#31-0), median armature on area II larger than median armatures on area III (#35-0), low density of granulation on dorsal scutum (#47-0), prolateral apophysis on coxa IV shorter than trochanter IV (#59-0), prolateral apophysis on coxa IV oblique in dorsal view (#62-2) and ventral plate thin (#98-0). The clade containing *M. arlei* **comb. nov.** and *M. minimus* **sp. nov.** is supported by large tubercles on lateral margin of dorsal scutum (#23-0), median armature on area III of the same color as the rest of the body (#38-2), femur straight (#74-0), absence of retrolateral basal apophysis on femur IV (#75-0). The less inclusive clade holding species from Boc, LSRJ and remaining species from Org AoE is supported by rounded lateral armatures on area III (#44-1), branched DBA (#80-3) and without apophysis after DBA (#86-0). Inside this last group, the lineage with species from LSRJ and remaining Org species is supported by small tubercles on free tergite II (#51-0) and sparse T1 microsetae on ventral side of ventral plate (#100-0). The clade with *M. scaber*, *M. poeta*, *M. spinifrons* **comb. nov.** and species from LSRJ is supported by basal tubercle on prolateral apophysis on coxa IV (#60-1), absence of a more developed apical tubercle on retrolateral row on femur IV (#94-0) and ventral plate thin in dorsal view (#98-0). The lineage holding

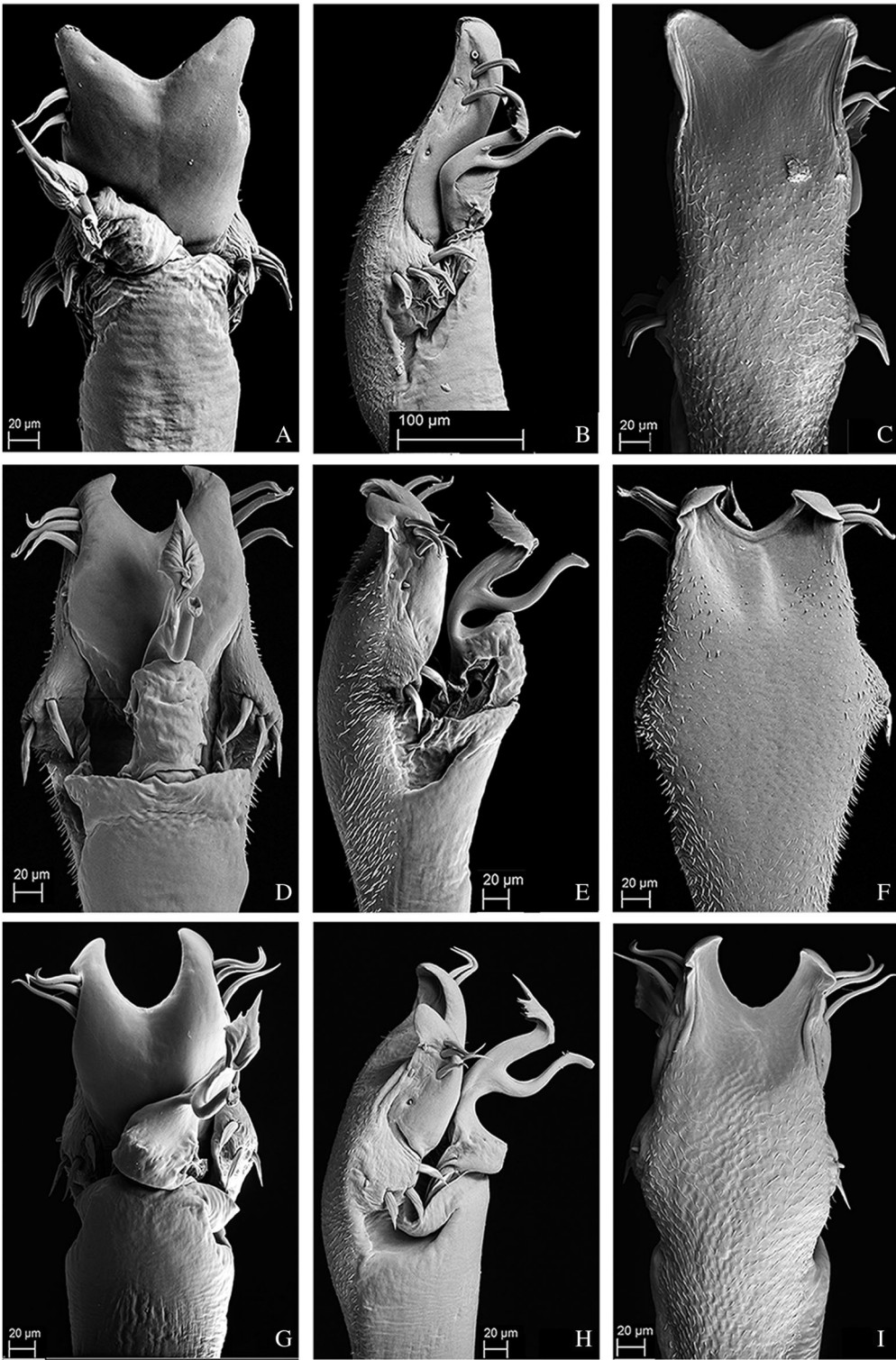

**Figure 10 Penis of *Mischonyx fidelis*, *M. insulanus* and *M. intermedius*.** (A–C) Dorsal, right lateral and ventral views, respectively, of the penis of *Mischonyx fidelis*. (D–F) Dorsal, right lateral and ventral views, respectively, of *Mischonyx insulanus*. (G–I) Dorsal, right lateral and ventral views, respectively, of the penis of *Mischonyx intermedius*.           

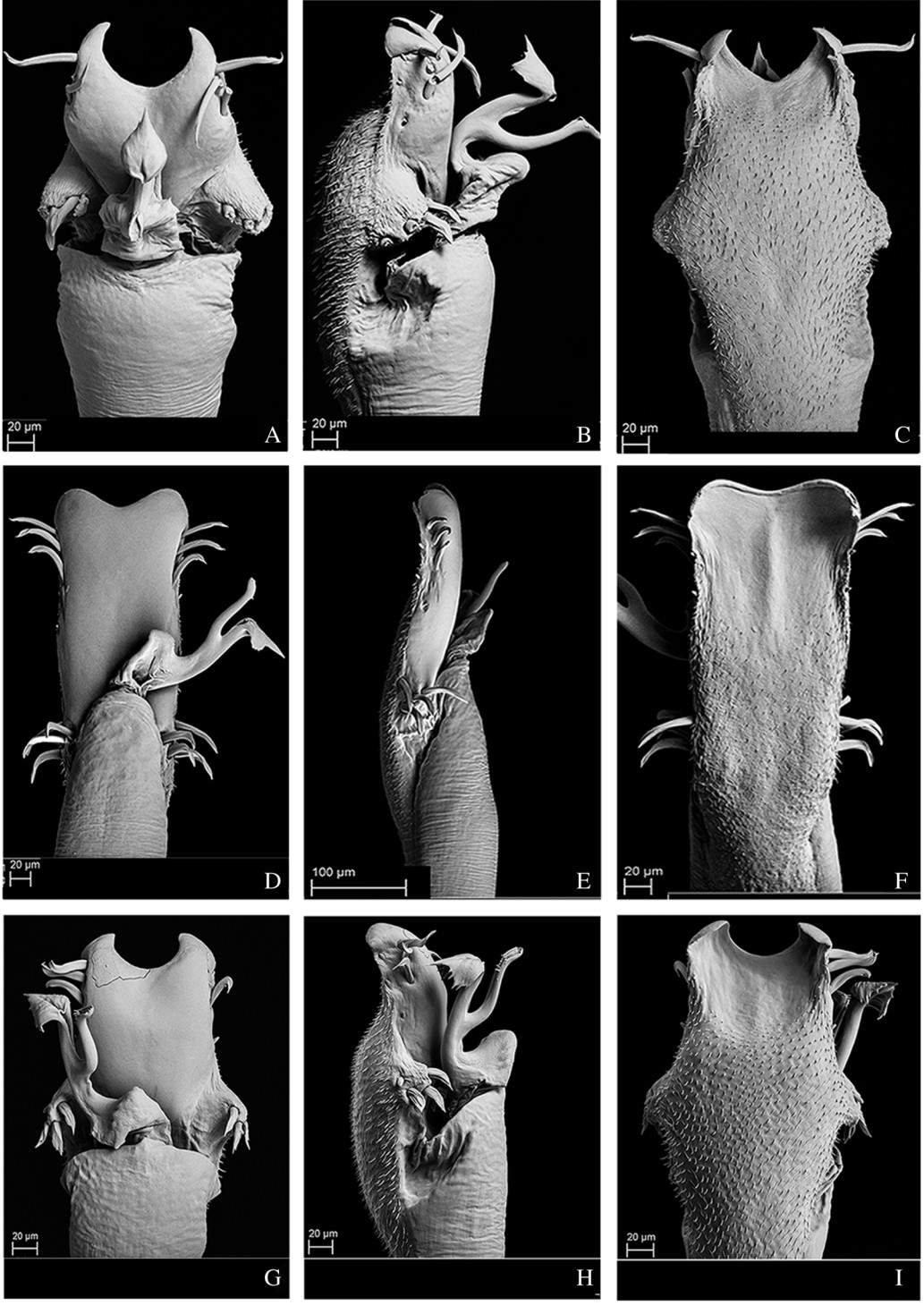

**Figure 11 Penis of *Mischonyx kaisara*, *M. parvus* and *M. poeta*.** (A–C) Dorsal, right lateral and ventral views, respectively, of the penis of *Mischonyx kaisara*. (D–F) Dorsal, right lateral and ventral views, respectively, of the penis of *Mischonyx parvus*. (G–I). Dorsal, right lateral and ventral views, respectively, of *Mischonyx poeta*.

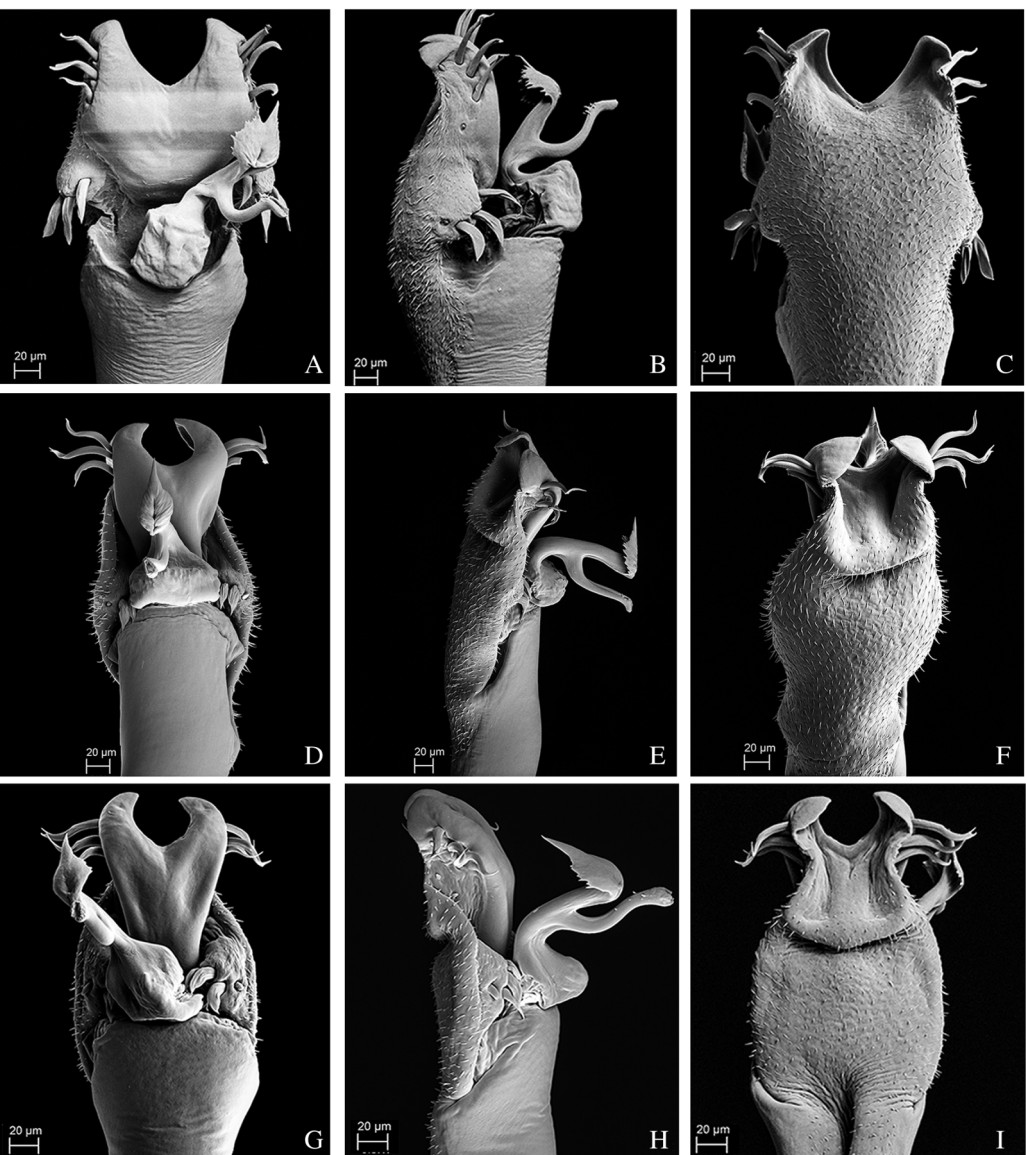

**Figure 12 Penis of *Mischonyx processigerus*, *M. spinifrons* and *M. squalidus*.** (A–C) Dorsal, right lateral and ventral views, respectively, of the penis of *Mischonyx processigerus*. (D–F) Dorsal, right lateral and ventral views, respectively, of the penis of *Mischonyx spinifrons*. (G–I) Dorsal, right lateral and ventral views, respectively, of the penis of *Mischonyx squalidus*.

*M. scaber* and *M. poeta* is supported by absence of secondary distal lobe on prolateral apophysis of coxa IV (#61-0), without retrolateral basal apophysis on femur IV (#75-0), small DBA (#77-0) and one apophysis on apical half of retrolateral row of femur IV (#93-1). The group with *M. spinifrons* **comb. nov.** and species from LSRJ is supported by ventral plate thin on lateral view (#97-1), weakly developed lateral lobes on ventral plate (#119-0) and flabellum longer than wide (#123-1). The clade with species from LSRJ is supported by DBA with base four times wider than apex (#79-0) and lateral parts of

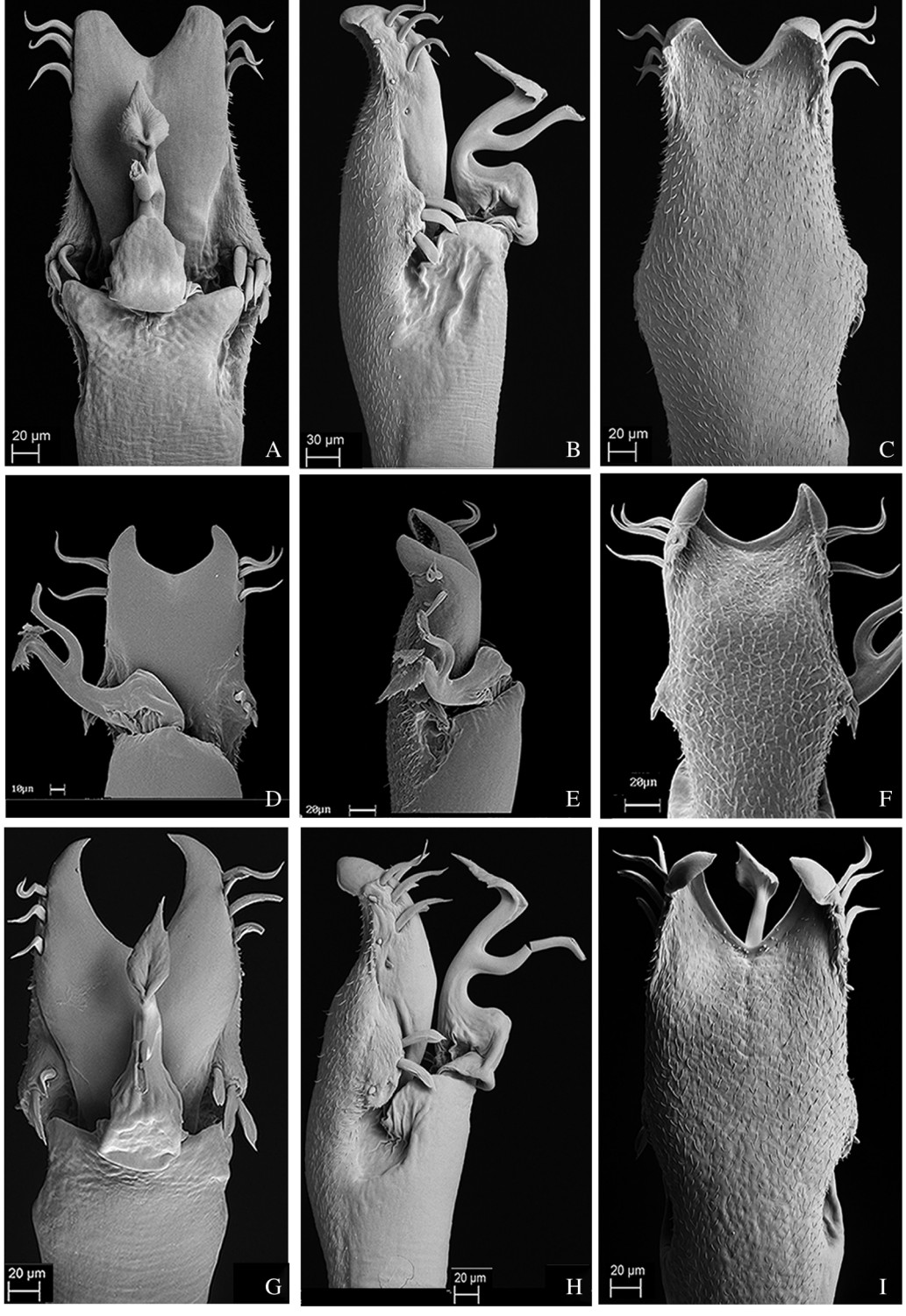

**Figure 13** **Penis of _Mischonyx anomalus, M. arlei_ and _M. clavifemur_.** (A–C) Dorsal, right lateral and ventral views, respectively, of _Mischonyx anomalus_. (D–F) Dorsal, right lateral and ventral views, respectively, of the penis of _Mischonyx arlei_. (G–I) Dorsal, right lateral and ventral views, respectively, of the penis of _Mischonyx clavifemur_.

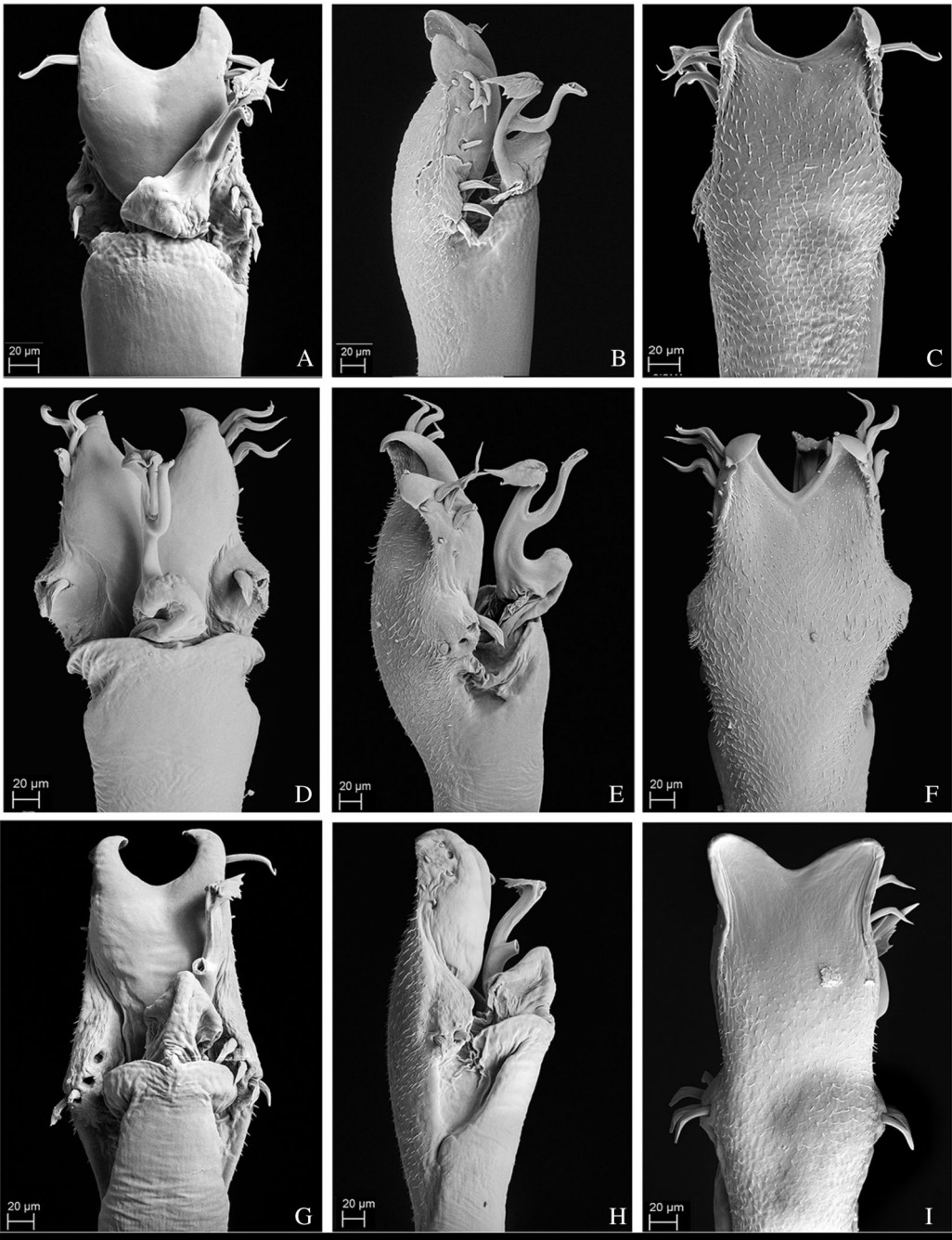

**Figure 14 Male genitalia of the new species.** (A–C) Dorsal, right lateral and ventral views, respectively, of the penis of *Mischonyx minimus* **sp. nov.** paratype (3649). (D–F) Dorsal, right lateral and ventral views, respectively, of the penis of *Mischonyx tinguaensis* **sp. nov.** paratype (2361). (G–I) Dorsal, right lateral and ventral views, respectively, of the penis of *Mischonyx intervalensis* **sp. nov.** paratype (0099).

flabellum smooth (#124-1). Finally, the clade holding *M. squalidus* and *M. parvus* **comb. nov.** is supported by presence of lateral tubercles on area II (#33-1), free tergite II with more developed central tubercle/ apophysis (#51-1), free tergite III with more

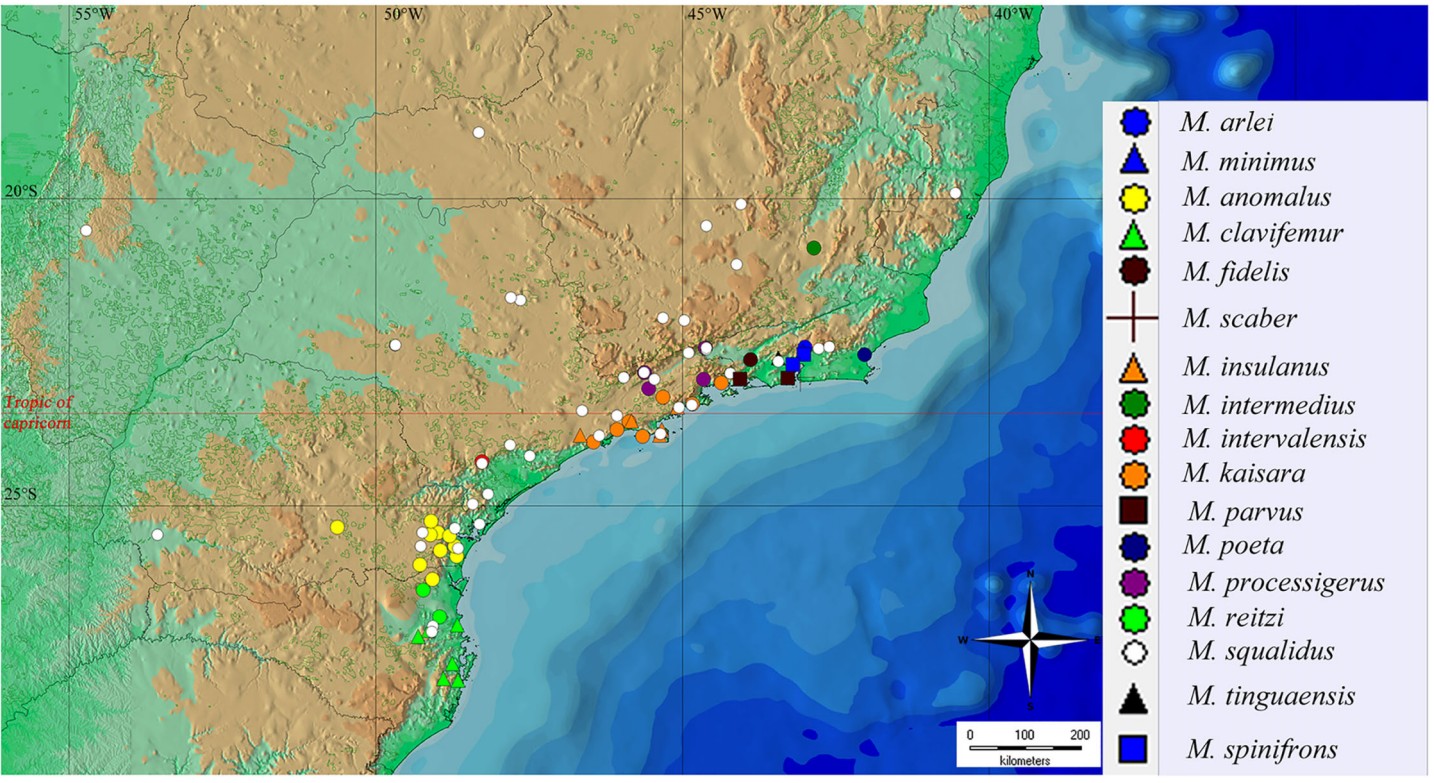

**Figure 15 General geographical distribution of *Mischonyx* species.** The red line represents the Tropic of Capricorn and the black grid represents the full meridians and parallels.

developed central tubercle/apophysis (#53-1) and absence of retrolateral basal apophysis on femur IV (#75-0).

## Taxonomic changes

### *Mischonyx: new combinations and diagnosis*

Before this publication, *Mischonyx* included the following 13 species, listed in *Kury (2003)* and *Pinto-da-Rocha et al. (2012)*: *M. anomalus* (*Mello-Leitão, 1936*); *M. antiquus* (*Mello-Leitão, 1934*); *M. cuspidatus* (*Roewer, 1913*); *M. fidelis* (*Mello-Leitão, 1931a*); *M. insulanus* (*Soares, 1972*); *M. intermedius* (*Mello-Leitão, 1935c*); *M. kaisara Vasconcelos, 2004*; *Mischonyx meridionalis* (*Mello-Leitão, 1927a*); *M. poeta Vasconcelos, 2005a*; *M. processigerus* (*Soares & Soares, 1970*); *M. scaber* (*Kirby, 1819*); *M. squalidus Bertkau, 1880* and *M. sulinus* (*Soares & Soares, 1947*).

Based on the ML3 hypothesis, we propose new combinations, composition and diagnosis for this genus:

## Mischonyx *Bertkau, 1880*

*Mischonyx Bertkau, 1880*: 106 (type species: *Mischonyx squalidus Bertkau, 1880*, by monotypy); *Mello-Leitão, 1935a*: 22; *Soares & Soares, 1949*: 221; *Kury, 2003*: 132; *Vasconcelos, 2004*: 129; 2005: 229; *Pinto-da-Rocha et al., 2012*: 51.

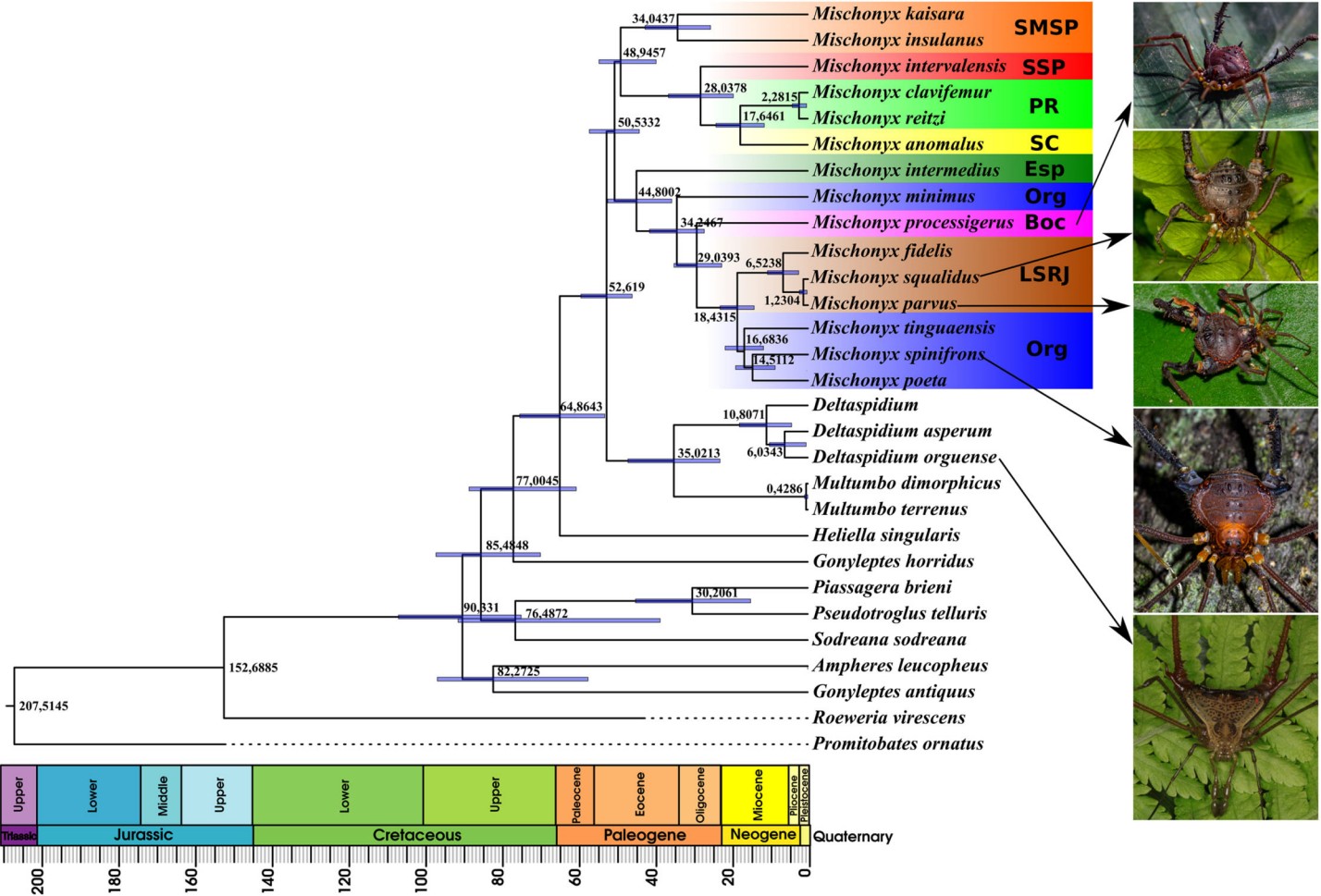

**Figure 16 Bayesian molecular dating (BM).** The values near the nodes are their respective node ages and the bars on each node are the 95% HPD values of each one. The colored clades are according to their location, respective to each Area of Endemism. Light green: PR; yellow: SC; Red: SSP; orange: SMSP; blue: Org; dark green: Esp; purple: Boc; brown: LSRJ and *M. squalidus*. Images on the right are individuals of the species indicated with arrows.

*Ilhaia Roewer, 1913*: 221; (type species *Ilhaia cuspidata Roewer, 1913*, by monotypy). Junior subjective synonym of *Mischonyx, Bertkau, 1880*: by *Kury, 2003*. In the present paper considered as a junior objective synonym of *Mischonyx*.

*Jlhaia* (misspelling): *Roewer, 1930*: 362.

*Eugonyleptes Roewer, 1913*: 219 (type species *Gonyleptes scaber Kirby, 1819*, by monotypy). Junior subjective synonym of *Mischonyx Bertkau, 1880*: by *Pinto-da-Rocha et al., 2012*.

*Xundarava Mello-Leitão, 1927b*: 19 (type species *Xundarava holacantha* Mello-Leitão, 1927, by original designation). Junior subjective synonym of *Mischonyx Bertkau, 1880*: by *Kury, 2003*.

*Gonazula Roewer, 1930*: 417 (type species *Gonazula gibbosa Roewer, 1930*, by monotypy). Junior subjective synonym of *Mischonx Bertkau, 1880*: by *Pinto-da-Rocha et al., 2012*.

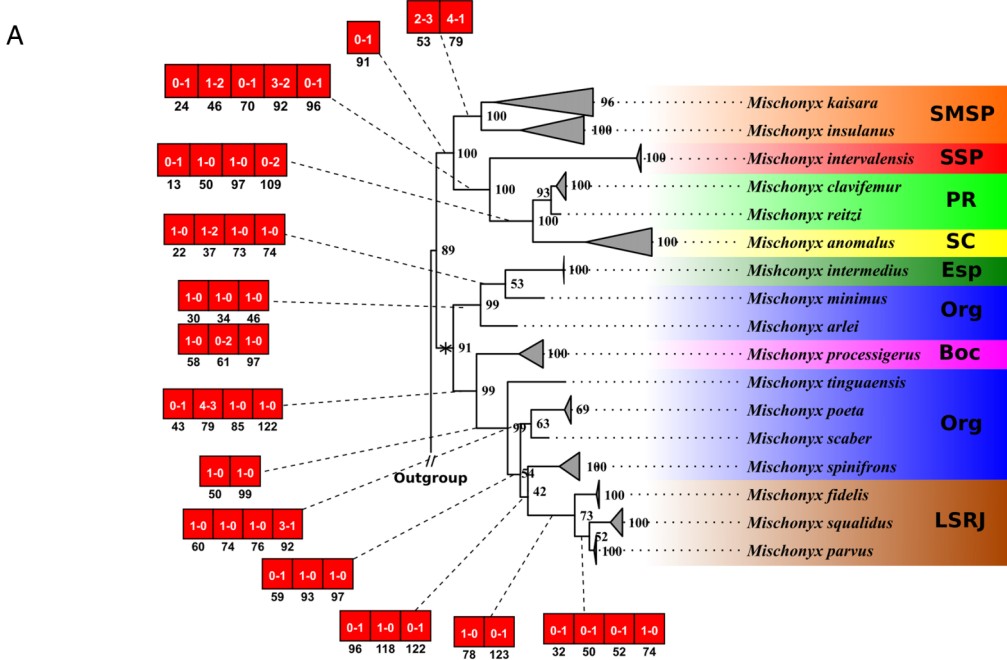

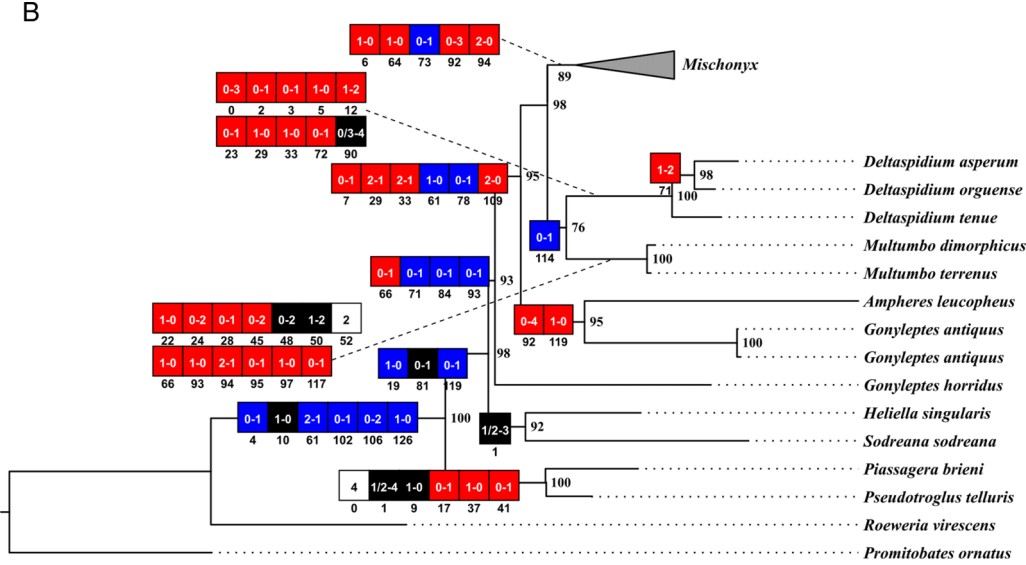

**Figure 17 Total Evidence Maximum Likelyhood hypothesis topology (ML3).** (A) *Mischonyx* clade; (B) external group. The numbers near the nodes are their respective bootstrap values. Unambiguous synapomorphies are represented in the squares. The numbers below each square is the character and the numbers inside the squares are plesiomorphic-apomorphic character-states. Red square = unique, homoplastic; blue square = non-unique, homoplastic; white = ambiguous; black = unique, non-homoplastic. The color of the clades are according to their location, respective to each Area of Endemism. Light green: SC; yellow: PR; Red: SSP; orange: SMSP; blue: Org; dark green: Esp; purple: Boc; brown: LSRJ and *M. squalidus*.                                     

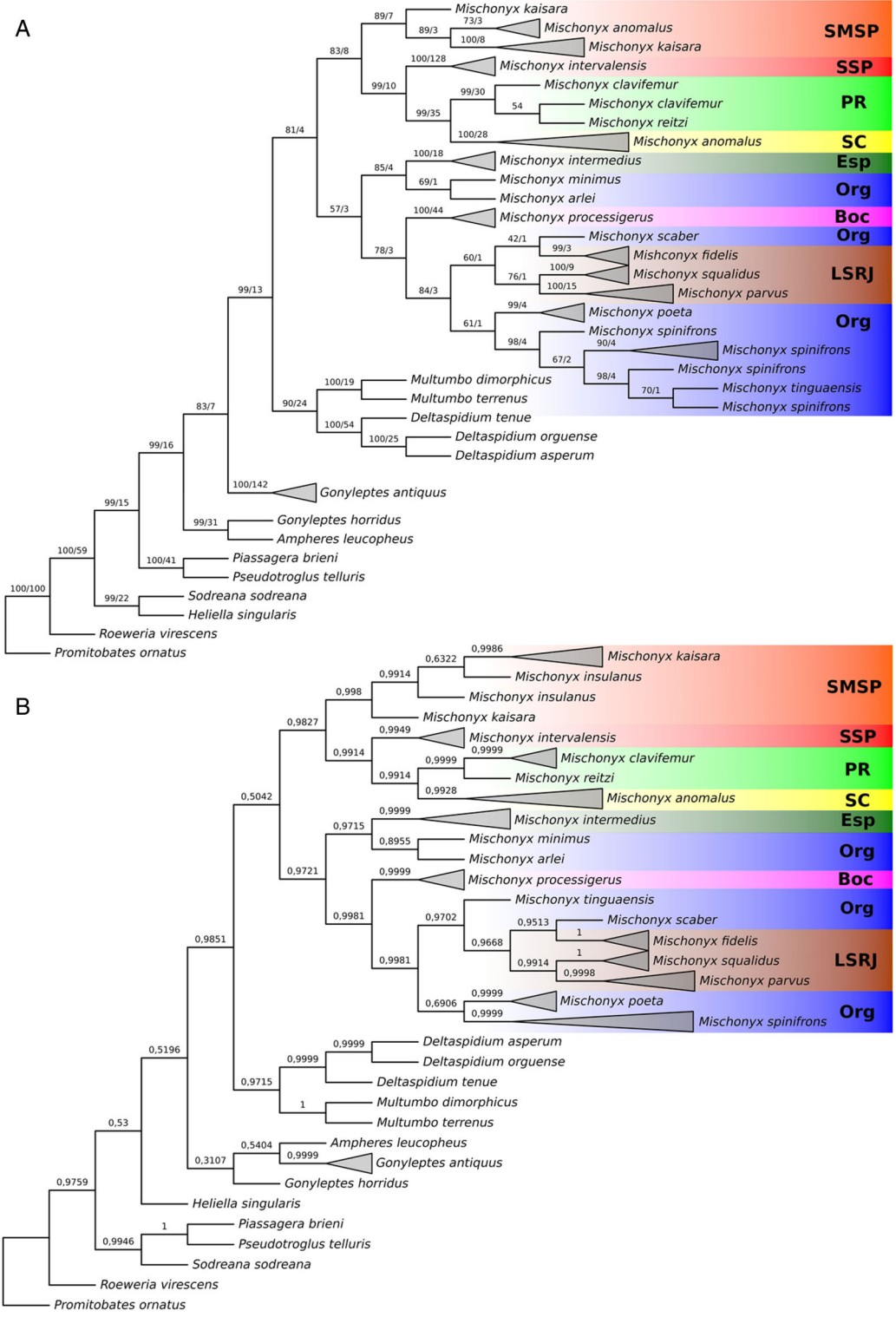

**Figure 18 Total Evidence Parsimony (MP3) and Bayesian (B3) phylogenetic hypothesis.** (A) MP3, values on each node are their respective bootstrap/Bremer values; (B) B3, values on each node are their respective posterior probability. The color of the clades are according to their location, respective to each Area of Endemism. Light green: SC; yellow: PR; Red: SSP; orange: SMSP; blue: Org; dark green: Esp; purple: Boc; brown: LSRJ and *M. squalidus*.

*Eduardoius Mello-Leitão, 1931a*: 94 (type species *Eduardoius fidelis Mello-Leitão, 1931a*, by original designation). Junior subjective synonym of *Mischonyx, Bertkau, 1880*: by *Kury, 2003*. *Cryptomeloleptes Mello-Leitão, 1931b*: 137 (type species *Criptomeloleptes spinosus Mello-Leitão, 1931b*, by original designation). Junior subjective synonym of *Mischonyx, Bertkau, 1880*: by *Kury, 2003*.

*Geraecormobiella Mello-Leitão, 1931b*: 127; *Soares, 1945c*: in a footnote [= *Geraeocormobius* Holmberg, 1887] (type species *Geraecormobiella convexa Mello-Leitão, 1931b*, by original designation). **Syn.nov.**

*Ariaeus Sørensen, 1932*: 281; *Vasconcelos, 2005b*: 2 [= *Geraeocormobius* Holmberg, 1887] (type species *Ariaeus tuberculatus Sørensen, 1932*, by monotypy). **Syn.nov.**

*Giltaya Mello-Leitão, 1932*: 466 (type species *Giltaya solitaria Mello-Leitão, 1932*, by original designation). Junior subjective synonym of *Mischonyx, Bertkau, 1880*: by *Kury, 2003*. *Bunoleptes Mello-Leitão, 1935c*: 398. (type species *Bunoleptes armatus Mello-Leitão, 1935c*, by original designation). Junior subjective synonym of *Mischonyx, Bertkau, 1880*: by *Kury, 2003*. *Arleius Mello-Leitão, 1935a*: 22 (type species *Arleius incisus Mello-Leitão, 1935a*, by original designation). Junior subjective synonym of *Mischonyx, Bertkau, 1880*: by *Kury, 2003*.

*Urodiabunus Mello-Leitão, 1935c*: 396; *1935b*: 104; *Soares & Soares, 1949*: 219. (type species *Urodiabunus arlei Mello-Leitão, 1935c*, by original designation). **Syn.nov.** *Penygorna Mello-Leitão, 1936*: 30 (type species *Penygorna infuscata Mello-Leitão, 1936*, by original designation). Junior subjective synonym of *Mischonyx, Bertkau, 1880*: by *Kury, 2003*.

Composition: *Mischonyx. anomalus* (*Mello-Leitão, 1936*); *Mischonyx arlei* (*Mello-Leitão, 1935b*) **comb. nov.**, *Mischonyx clavifemur* (*Mello-Leitão, 1927a*) **comb. nov.**; *Mischonyx fidelis* (*Mello-Leitão, 1931a*); *Mischonyx insulanus* (*Soares, 1972*); *Mischonyx intermedius* (*Mello-Leitão, 1935b*); *Mischonyx intervalensis* **sp. nov.**; *Mischonyx kaisara Vasconcelos, 2004*; *Mischonyx minimus* **sp. nov.**; *Mischonyx parvus* (*Roewer, 1917*) **comb. nov.**; *Mischonyx poeta Vasconcelos, 2005a*; *Mischonyx processigerus* (*Soares & Soares, 1970*); *Mischonyx reitzi* (*Vasconcelos, 2005b*) **comb. nov.**; *Mischonyx scaber* (*Kirby, 1819*); *Mischonyx spinifrons* (*Mello-Leitão, 1923*) **comb. nov.**; *Mischonyx squalidus Bertkau, 1880*; *Mischonyx tinguaensis* **sp. nov.**.

Taxonomic remarks: we transferred *Geraeocormobius reitzi Vasconcelos, 2005b*, *Urodiabunus arlei Mello-Leitão, 1935c*, *Weyhia clavifemur* Mello-Leitão, 1927, *Weyhia spinifrons Mello-Leitão, 1923* and *Weyhia parva Roewer, 1917* to *Mischonyx* based on molecular and morphological evidence. The other new combinations are also based on the morphological analysis of the types, with one exception, *M. squalidus*. Since we were not able to study the holotype of this species, the new synonym had to be based on original figures and description from Bertkau. *Vasconcelos (2003)*, in his master's dissertation, and *Benedetti (2017)*, in his PhD thesis, had already proposed most of these combinations. However, they did not publish their conclusions. According to the *ICZN (1999)*, nomenclatural acts in theses or dissertations are not valid if they are not officially published.

Besides that, based on our phylogenetic analysis, we re-establish the original combination of *Gonleptes antiquus Mello-Leitão, 1934*, removing the species from *Mischonyx*. This species was considered a member of *Mischonyx* by *Kury (2003)* and

*Pinto-da-Rocha et al. (2012)*. Now it returns to the genus in which it had been originally described. Consequently, we remove the genus *Anoploleptes Piza, 1940* from subjective the junior synonym list of *Mischonyx*, as established by *Kury (2003)*, since *Anoploleptes dubium* (type species of *Anoploleptes*) is a junior synonym of *Gonyleptes antiquus* (see *Soares, 1943*). Therefore, *Anoploleptes* is a junior synonym of *Gonyleptes* as established by *Soares (1943)*.

As pointed out by *Acosta, Kury & Juárez (2007)* "the correct (original) spelling of the generic name is *Geraeocormobius*".

**Diagnosis.** Small Gonyleptinae (3–6 mm of dorsal scutum length). Dorsal scutum outline γP in males, with coda involved by the mid-bulge, which is very distinct.

Females have dorsal scutum outline α, with coda long and clearly separated from mid-bulge. Anterior margin with lateral armature, normally two or three tubercles on each side. Frontal hump high and narrow, with a pair of median tubercles (except in *M. processigerus*, which has two pairs). Lateral margin of prosoma with several granules, posterior to ozopore. Ocularium narrow and not very high, armed with median spines or tubercles. Some species have small tubercles anterior or posterior to the eye (or both). Posterior margin of prosoma with a pair of tubercles. Dorsal scutum with three areas. Mesotergal area I is divided by a longitudinal groove. Areas I and II armed with median tubercles (which are large and whitish in *M. arlei* **comb. nov.** and *M. minimus* **sp. nov.**). Area III with a pair of median elliptic tubercles (except in *M. arlei* **comb. nov.** and *M. minimus* **sp. nov.**), which can vary in size and lateral compression. Some species have other elliptic tubercles besides the median ones (*e.g.*, *M. spinifrons* **comb. nov.**). Lateral margin of dorsal scutum (mid-bulge) with rounded tubercles, which are fused in some species (*e.g.*, *M. spinifrons* **comb. nov.**). Distitarsi of all legs with three segments. Basitarsus of leg I with three or four segments. Basitarsus II with four – eight segments. Basitarsi III and IV with four or five segments. Ventral surface of coxae I generally with more developed tubercles than the ones on the other coxa. Coxa IV with apical prolateral apophysis, generally robust, in some speciemens with ventral process and a basal tubercle. Trochanter IV short and robust, with a blunt prolateral apophysis and at least one retrolateral armature. Femur IV with DBA, which can be small (as in *M. arlei* **comb. nov.** and *M. minimus* **sp. nov.**), or large (most species). DBA can be branched or not and varies in shape and size in every species. Retrolateral row of tubercles generally with some large apophysis. Penis with ventral plate trapezoidal with an apical parabolic groove; three pairs of MS A and one pair of MS B on lateral projections; three pairs of helicoidal MS C, two pairs of reduced MS E, one pair of MS D, venter of ventral plate with microsetae type T1 covering its whole extension or the basal half. Glans with ventral process, *flabellum* can be serrated or smooth. Stylus with microsetae, inclined in relation to axis of penis and with ventral groove.

### Species new combinations

Besides the combinations and synonyms present in *Kury (2003)* and *Pinto-da-Rocha et al. (2012)*, the following new combinations are here proposed:

***Mischonyx. anomalus*** (*Mello-Leitão, 1936*) (Figs. 1A, 1C, 13A–13C)

*Xundarava anomala* *Mello-Leitão, 1936*: 13, fig. 10; *Soares, 1945b*: 192; *1945c*: 366; *Soares, 1945d*: 210; *Soares & Soares, 1949*: 220 (Male and female syntypes, Brazil, Paraná, Antonina; MNRJ 42282).

*Ilhaia anomala*: *Soares & Soares, 1987*: 7.

*Mischonyx anomalus*: *Kury, 2003*: 133; *Pinto-da-Rocha et al., 2012*: 52.

*Ilhaia sulina* *Soares & Soares, 1947*: 215 (Male lectotype and female paralectotype; Brazil Paraná, Florestal; MHNCI 3618 and MHNCI 3619, respectively). **Syn. nov.**

*Mischonyx sulinus*: *Kury, 2003*: 134; *Pinto-da-Rocha et al., 2012*: 52.

**Diagnosis.** *Mischonyx anomalus* resembles *M. clavifemur* **comb. nov.** in the following: prolateral apophysis of coxa IV with apex directed posteriorly; prolateral apophysis of trochanter IV small when compared to other species; retrolateral row of femur IV with median apophysis larger than the other armatures of this row; ventral plate of penis with MS A forming a baso-apical, reduced MS B, MS E slightly medial when compared to the MS C, ventral side entirely covered with microsetae, lateral lobes basal. It differs from *M. clavifemur* **comb. nov.** in the following: reduced size (4–4.5 mm of dorsal scutum length) (5–6 mm in *M. clavifemur* **comb. nov.**); Dorsal scutum narrower than in *M. clavifemur* **comb. nov.**; Mesotergal Area III with a pair of large median tubercles (reduced in *M. clavifemur* **comb. nov.**); retrolateral side of trochanter IV with a row of small tubercles (two tubercles in *M. clavifemur* **comb. nov.**, with the apical more developed than the other); ventral plate longer than wider (as wide as long in *M. clavifemur* **comb. nov.**) dorsal row of femur IV with small tubercles only after DBA (three large tubercles after DBA in *M. clavifemur* **comb. nov.**) apical groove reaching the line of the second MS C (reaching deeper than the MS C in *M. clavifemur* **comb. nov.**).

*Mischonyx arlei* **(Mello-Leitão, 1935) comb. nov.** (Figs. 1B, 1D, 13D–13F)

*Urodiabunus arlei* *Mello-Leitão, 1935c*: 397, fig. 22 (1 Male 1 female syntypes; Brazil, Rio de Janeiro, Petrópolis; MNRJ 42476).

**Diagnosis.** *Mischonyx arlei* **comb. nov.** resembles *M. minimus* **sp. nov.** by the following combinations of characters: mesotergal area I with a pair of well-developed median tubercles, which are paler (whitish) than the rest of the dark brown body; median armatures on mesotergal area III are spines; lateral margin of dorsal scutum with several small tubercles; Free tergite II with a well-developed median apophysis; prolateral apophysis on coxa IV small and pointing posteriorly; retrolateral side of trochanter IV with two armatures; femur IV with several small apophyses on dorsal and retrolateral row of tubercles; femur IV with a well-developed terminal tubercle on pro and retrolateral rows of tubercles; ventral plate with three subdistal MS C on each side; MS B smaller than MS A; *flabellum* with serrated ends. It differs from *M. minimus* **sp. nov.** in the following: size (7–8 mm) (3–3.5 mm in *M. minimus* **sp. nov.**); mesotergal area II with median tubercles small and darker than the rest of the body (median tubercles whitish and as large as the median tubercles on mesotergal area I in *M. minimus* **sp. nov**); basitarsus II with seven segments (four in *M. minimus* **sp. nov**); leg IV curved in dorsal view (straight in *M. minimus* **sp. nov**); MS D reduced (well-developed in *M. minimus* **sp. nov**).

***Mischonyx clavifemur*** **(Mello-Leitão, 1927) comb. nov.** (Figs. 2A, 2C, 13G–13I)

*Weyhia clavifemur Mello-Leitão, 1927a*: 416; *Roewer, 1930*: 356; *Mello-Leitão, 1932*: 286, fig. 177 (Male holotype; Brazil, Santa Catarina, Blumenau; MNRJ 1496).

*Geraeocormobius clavifemur*: Mello-Leitão, 1940: 22; *Soares, 1945c*: 354; *Soares & Soares, 1949*: 169; *Vasconcelos, 2005b*: 3, figs. 1–9; *Pinto-da-Rocha et al., 2014*: 12, 16.

*Ilhaia meridionalis Mello-Leitão, 1927a*: 417 (female holotype; Brazil, Santa Catarina, Blumenau; MNRJ 1474); *Vasconcelos, 2005b*: 3. Synonymy established by *Vasconcelos, 2005b*.

*Jlhaia meridionalis* (misspelling): *Roewer, 1930*: 363.

*Mischonyx meridionalis*: *Kury, 2003*: 133–34.

*Ariaeus tuberculatus Sørensen, 1932*: 282 (female holotype; Brazil, Santa Catarina, Blumenau; BMNH); *Vasconcelos, 2005b*: 3. Synonymy established by *Vasconcelos, 2005b*.

**Diagnosis.** *Mischonyx clavifemur* **comb. nov.** resembles *M. anomalus.* in the following: prolateral apophysis of coxa IV with apex directed posteriorly; prolateral apophysis of trochanter IV small when compared to other species; retrolateral row of femur IV with median apophysis larger than the other armatures of this row; ventral plate of penis with MS A forming a baso-apical, reduced MS B, MS E slightly medial when compared to the MS C, ventral side entirely covered with microsetae, lateral lobes basal. It differs from *M. anomalus* by: its size (5–6 mm of dorsal scutum) (4–4.5 mm in *M. anomalus*); mesotergal area III with small median tubercles (more developed in *M. anomalus*); retrolateral side of trochanter IV with two tubercles, with the apical more developed than the other (a row of small tubercles in *M. anomalus*); ventral plate of the penis as wide as long (longer than wider in *M. anomalus*) dorsal row of femur IV with three large tubercles after DBA (small tubercles only after DBA in *M. anomalus*), apical groove reaching deeper than the line of the last MS C (reaching the line of the second MS C in *M. anomalus*).

***Mischonyx fidelis*** **(Mello-Leitão, 1931)** (Figs. 2B, 2D, 10A–10C)

*Eduardoius fidelis Mello-Leitão, 1931a*: 95; *1932*: 344 (2 syntypes; Brazil, Rio de Janeiro, Piraí; MNRJ 1408).

*Ilhaia fidelis*: *Soares, 1943*: 56 [by implication]; *1945c*: 358; *Soares & Soares, 1946*: 76; *1949*: 186.

*Mischonyx fidelis*: *Kury, 2003*: 133; *Pinto-da-Rocha et al., 2012*: 52.

**Diagnosis.** *M. fidelis* resembles *M. parvus* **comb. nov.** in the following: pair of tubercles on frontal hump and lateral margins of dorsal scutum whitish (in ethanol); median tubercles on mesotergal area III large and elliptic; prolateral apophysis of trochanter IV large when compared to other species (*e.g., M. spinifrons* **comb. nov.**); DBA conic and the tallest of the genus (almost as tall as the whole body), with a tubercle on anterior side of apophysis; prolateral row of femur IV with median tubercles more developed than the others on this row; retrolateral row of femur IV with the largest tubercle on distal third; apex of penis truncus not globose in lateral view; ventral plate with microsetae only on basal half; apical groove shallow, reaching the line of the most apical MS C; lateral projections basal; MS A forming a dorso-ventral line; MS E basal when compared to the

MS C; flabellum with median large projection. It differs from *M. parvus* **comb. nov.** in the following: prolateral apophysis on coxa IV with small ventral lobe (ventral lobe as developed as the main projection in *M. parvus* **comb. nov.**); retrolateral side of trochanter IV with three small tubercles (two large tubercles in *M. parvus* **comb. nov.**); dorsal row of femur IV with an elevation basal to the DBA (absence of an elevation basal to the DBA in *M. parvus* **comb. nov.**); dorsal row of femur IV with small tubercles only after DBA (one large tubercle after DBA in *M. parvus* **comb. nov.**); retrolateral row of femur IV with three large tubercles on the basal half (without large tubercles tubercles on the basal half in *M. parvus* **comb. nov.**); ventral plate of the penis as large as wide (larger than wider in *M. parvus* **comb. nov.**); lateral lobes projected (not projected in *M. parvus* **comb. nov.**); MS B ventral to MS A (MS B apical to the MS A in *M. parvus* **comb. nov.**); MS C more distal than in *M. parvus* **comb. nov.**

***Mischonyx insulanus*** (*Soares, 1972*) (Figs. 3A, 3C, 10D–10F)
  *Ilhaia insulana Soares, 1972*: 65, figs. 1–4 (Male holotype, 1female paratype; Brazil, São Paulo, São Sebastião; HSPC 361).
  *Mischonyx insulanus*: *Kury, 2003*: 133; *Pinto-da-Rocha et al., 2012*: 52.

**Diagnosis.** *M. insulanus* resembles *M. processigerus* in the following combinations of characters: median tubercles on ocularium smaller than height of ocularium; ocularium with small tubercles on anterior and posterior sides; mesotergal area III with small median tubercles when compared to other species (*e.g., M. fidelis*); Free tergites II and III with median apophysis; prolateral row of femur IV with median tubercles larger than the others in this row; dorsal row of femur IV with small tubercles after DBA; retrolateral row of femur IV with the largest apophysis on the distal third; ventral side of the ventral plate of the penis with microsetae only on the laterals; lateral lobes well-developed; apical groove of the ventral plate reaching the line of the second MS C; MS A forming a dorso-ventral line; reduced MS B. It differs from *M. processigerus* by: prolateral apophysis of coxa IV with ventral lobe as large as the main projection and close to each other (ventral lobe smaller and more separated from the main projection of the apophysis in *M. processigerus*); retrolateral apophysis of coxa IV not visible in dorsal view; (visible in *M. processigerus*); DBA not branched (branched in *M. processigerus*); retrolateral row of femur IV with two large apophysis (one in *M. processigerus*); retrolateral row of femur IV with small tubercles besides the two apophysis (several large tubercles in *M. processigerus*); *flabellum* with smooth apex (serrated in *M. processigerus*); stylus without microsetae (stylus with microsetae in *M. processigerus*); MS B closer to MS E when compared to *M. processigerus*.

***Mischonyx intermedius*** (Mello-Leitão, 1935) (Figs. 3B, 3D, 10G–10I)
  *Ilhaia intermedia Mello-Leitão, 1935c*: 401, fig. 25; *1935b*: 107 (Male holotype; Brazil, Minas Gerais, Viçosa; IBSP 46).
  *Penygorna infuscata Mello-Leitão, 1936*: 31, fig. 26 (1 Male 2 female syntypes; Brazil, Minas Gerais; Viçosa; MNRJ 42695). Synonymy established by *Soares, 1944*.
  *Mischonyx intermedius*: *Kury, 2003*: 133; *Pinto-da-Rocha et al., 2012*: 52.

**Diagnosis.** *M. intermedius* resembles *M. arlei* **comb. nov.** in the following: lateral margin of dorsal scutum with several small tubercles; mesotergal area III with median tubercles that are not elliptic; prolateral apophysis of coxa IV smaller than trochanter IV, blunt and oblique to the body axis; femur IV thin and long; retrolateral row of femur IV with an apical sharp tubercle; MS B reduced; MS E in the same dorso-basal line of the MS C; *flabellum* with serrated ends. It differs from *M. arlei* **comb. nov.** by: median tubercles on mesotergal area I smaller than the median tubercle of the other mesotergal areas and darker than the rest of the body color (in ethanol) (bigger and whitish in *M. arlei* **comb. nov.**); Free tergite II with small tubercles only (large median apophysis in *M. arlei* **comb. nov.**); retrolateral apophysis of coxa IV not visible in dorsal view (visible in *M. arlei* **comb. nov.**) prolateral apophysis of trochanter IV large (reduced in *M. arlei* **comb. nov.**); retrolateral side of trochanter IV with a line of three tubercles (two n *M. arlei* **comb. nov.**); DBA large in relation to the other armature on the dorsal row and with its apex directed anteriorly (DBA almost same size as other tubercles on the row and with its apex directed dorsally in *M. arlei* **comb. nov.**); prolateral ros of femur IV with a large number of tubercles when compared to other species (*e.g., M. spinifrons* **comb. nov.** and *M. arlei* **comb. nov.**); retrolateral row of femur IV with tubercles increasing in size apically (retrolateral row with minute armature in *M. arlei* **comb. nov.**); ventral side of the ventral plate of the penis with microsetae on the basal half (ventral side entirely covered with microsetae in *M. arlei* **comb. nov.**); apical groove of the ventral plate of the penis reaches the line of the most basal MS C (apical groove reaches the line of the median MS C in *M. arlei* **comb. nov.**); MS A forming a parable (MS A forming a diagonal baso-apical line in *M. arlei* **comb. nov.**); MS D more apical, when compared to *M. arlei* **comb. nov.**, that has the MS D medial on the ventral plate.

***Mischonyx kaisara*** Vasconcelos, 2004 (Figs. 4B, 4D, 11A–11C)
   *Mischonyx kaisara* Vasconcelos, 2004: 130, figs. 1–9. (Male holotype; 5 Male paratypes; Brazil, São Paulo, Ilha Bela; MNRJ 17437 and MZSP 23147, respectively).
   As *M. kaisara* was recently described and there is no new combination for the species, Vasconcelos (2004) diagnosis for the species remains unaltered and with no necessity to add information.

***Mischonyx parvus*** (*Roewer, 1917*) **comb. nov.** (Figs. 5B, 5D, 11D–11F)
   *Weyhia parva* Roewer, 1917: 133 (Male holotype, Brazil, São Paulo. Santos; SMF 1331).
   *Geraeocormobius parva*: Mello-Leitão, 1940: 22.
   *Geraeocormobius parvus*: Soares, 1945c: 355; Soares & Soares, 1949: 171.
   *Ilhaia parva*: Soares & Soares, 1987: 6.
   *Cryptomeloleptes spinosus Mello-Leitão, 1931b*: 138 (holotype; Brazil, Rio de Janeiro, Rio de Janeiro; MNRJ 11392). Synonymy established by Soares & Soares, 1987.
   *Arleius incisus Mello-Leitão, 1935a*: 22 (holotype; Brazil, Rio de Janeiro, Rio de Janeiro; MNRJ 41759). Synonymy established by Soares & Soares, 1987.
   *Ilhaia incisa*: Soares & Soares, 1946: 76; Soares, 1974: 354, fig. 2. [= *Bunoleptes armatus* Mello-Leitão, 1935e; = *Geraecormobius cervicornis* Mello-Leitão, 1940b].

*Bunoleptes armatus* *Mello-Leitão, 1935a*: 398 (Male holotype, 2 Male paratypes; Brazil, Rio de Janeiro, Rio de Janeiro; MNRJ 42477 and MZSP 2328) Synonymy established by *Soares & Soares, 1987*.

*Geraecormobius cervicornis* Mello-Leitão, 1940: 17 (Male holotype lost; Brazil, Rio de Janeiro, Mangaritiba; MNRJ 53924). Synonymy established by *Soares & Soares, 1987*.

**Diagnosis.** *M. parvus* **comb. nov.** resembles *M. fidelis* in the following: pair of tubercles on the frontal hump and lateral margins of the dorsal scutum whitish (in ethanol); median tubercles on mesotergal area III large and elliptic; prolateral apophysis of trochanter IV large, when compared to other species (*e.g.*, *M. spinifrons* **comb. nov.**); DBA conic and the tallest of the genus (almost as tall as the whole body), with a tubercle on the anterior side of the apophysis; prolateral row of femur IV with median tubercles more developed than the others on this row; retrolateral row of femur IV with the largest tubercle on distal third; penis not globose in lateral view; ventral plate with microsetae only on basal half; apical groove shallow, reaching the line of the most apical MS C; lateral projections basal; MS A forming a dorso-ventral line; MS E basal when compared to MS C; flabellum with median projection large. It differs from *M. fidelis* by: prolateral apophysis on coxa IV with ventral lobe as developed as the main projection (ventral lobe reduced in *M. fidelis*); retrolateral side of trochanter IV with two large tubercles (small in *M. fidelis*); dorsal row of femur IV without an elevation basal to the DBA (presence of an elevation basal to the DBA in *M. fidelis*); dorsal row of femur IV with a large tubercle after DBA (small tubercles only after DBA in *M. fidelis*); retrolateral row of femur IV without large tubercles on the basal half (three large tubercles on the basal half in *M. fidelis*); ventral plate of the penis larger than wider (as large as wide in *M. fidelis*); lateral lobes not very projected, with the MS A and MS B close to the penis base (projected in *M. fidelis*); MS B apical to MS A (MS B ventral to the MS A in *M. fidelis*); MS C more median than in *M. fidelis*.

Taxonomic remarks: *Kury (2003)* synonymized this species with *M. squalidus*. However, the distribution of *M. parvus* does not match with the original location of the described individual in *Bertkau (1880)*. In the latter work, the location of the specimen is "Copacabana, Rio de Janeiro". By the distribution map in the Figs. 15, S1 and S2, the registers from this species are from Mangaratiba and Angra dos Reis, which are to the south of Rio de Janeiro state. For this reason, we removed this species from the synonymy created by *Kury (2003)*.

***Mischonyx poeta*** **Vasconcelos, 2005** (Figs. 6A, 6C, 11G–11I)

*Mischonyx poeta* *Vasconcelos, 2005a*: 229, figs. 1–9. (Male holotype; Brazil, Rio de Janeiro, Casimiro de Abreu; MNRJ 17460).

As *M. poeta* was recently described and there is no new combination for the species, *Vasconcelos (2005a)* diagnosis for the species remains unaltered and with no necessity to add information.

***Mischonyx processigerus*** (*Soares & Soares, 1970*) (Figs. 6B, 6D, 12A–12C)

*Ilhaia processigera* *Soares & Soares, 1970*: 340, figs. 1–3 (Male holotype, 1 female paratype; Brazil, Rio de Janeiro, Itatiaia; MZUSP 4501).

*Mischonyx processigerus*: *Kury, 2003*: 134; *Pinto-da-Rocha et al., 2012*: 52.

**Diagnosis.** *M. processigerus* resembles *M. insulanus* in the following: median tubercles on ocularium smaller than the ocularium height; ocularium with small tubercles on the anterior and posterior sides; mesotergal area III with small median tubercles when compared to other species (*e.g., M. fidelis*); Free Tergites II and III with median apophysis; prolateral row of femur IV with median tubercles larger than the others in this row; dorsal row of femur IV with small tubercles after DBA; retrolateral row of femur IV with the largest apophysis on the distal third; ventral side of the ventral plate of the penis with microsetae only on the laterals; lateral lobes well-developed; apical groove of the ventral plate reaching the line of the second MS C; MS A forming a dorso-ventral line; reduced MS B. It differs from *M. insulanus* by: prolateral apophysis of coxa IV with ventral lobe small and separated from the main projection (ventral lobe as large as the main projection and close to each other in *M. insulanus*); retrolateral apophysis of coxa IV visible in dorsal view; (not visible in *M. insulanus*); DBA branched (not branched in *M. insulanus*); retrolateral row of femur IV with one large apophysis (two in *M. insulanus*); retrolateral row of femur IV with large tubercles besides the apophysis (small tubercles in *M. insulanus*); *flabellum* with serrated apex (smooth in *M. insulanus*); stylus with microsetae (stylus without microsetae in *M. insulanus*); MS B distant from MS E when compared to *M. insulanus*.

### *Mischonyx reitzi* (Vasconcelos, 2005) comb. nov. (Figs. 7A, 7C, 22A–22C)
*Geraeocormobius reitzi Vasconcelos, 2005b*: 6, figs. 10–19. (Male holotype; Brazil, Santa Catarina, Ilhota; MNRJ 6949).

**Diagnosis.** *M. reitzi* **comb. nov.** resembles *M. tinguaensis* **sp. nov.** in the following: Median armature on mesotergal area III small when compared to other species (*e.g., M. spinifrons* **comb. nov.**) and elliptic; no median armature on Free Tergites I–III; prolateral apophysis on coxa IV with its apex directed laterally, as large as the trochanter IV and with ventral lobe; a small tubercles basal to DBA on the dorsal row; DBA branched; dorsal row of femur IV with small tubercles only; prolateral row with tubercles of the same size; apical groove on ventral plate of the penis reaching the line of the most basal MS C; MS A forming a baso-apical line; stylus with microsetae. It differs from *M. tinguaensis* **sp. nov.** by: lateral margin of dorsal scutum with small tubercles which have the same color of the rest of the body (whitish than the rest of the body in *M. tinguaensis* **sp. nov.**); median armature on ocularium smaller than the ocularium height (bigger in *M. tinguaensis* **sp. nov.**); trochanter IV with two retrolateral tubercles (three in *M. tinguaensis* **sp. nov.**); median apophysis on retrolateral row of femur IV is the largest on this row (biggest apophysis is on the apical third in *M. tinguaensis* **sp. nov.**); MS B reduced (as large as MS A in *M. tinguaensis* **sp. nov.**).

### *Mischonyx scaber* (Kirby, 1817) (Figs. 7B, 7D)
*Gonyleptes scaber Kirby, 1819*: 453 (3 males and 1 female syntypes; Brazil; NHM 1863.41).

*Eugonyleptes scaber*: *Roewer, 1913*: 219; *Kury, 2003*: 123.

*Xundarava holacantha Mello-Leitão, 1927b*: 20 (female holotype; Brazil, Rio de Janeiro, Niteroi; MNRJ 1469). Synonymy established by *Pinto-da-Rocha et al., 2012*.

*Weyhia vellardi* Mello-Leitão *in litteris*: Soares & Soares, 1987a: 7.

*Ilhaia holacantha*: *Soares & Soares, 1987*: 7, figs. 27–28.

*Weyhia absconsa Mello-Leitão, 1932*: 284, fig. 175; *Soares & Soares, 1987*: 7 [= *Xundarava holacantha* Mello-Leitão, 1927]. (Male holotype; Brazil, Rio de Janeiro, Niteroi; MNRJ 1501). Synonymy established by implication in *Pinto-da-Rocha et al., 2012*.

*Geraeocormobius absconsa*: Mello-Leitão, 1940: 22.

*Geraeocormobius absconsus*: *Soares, 1945c*: 354; *Soares & Soares, 1949*: 167.

*Geraeocormobius carioca* Mello-Leitão, 1940: 18, fig. 22; *Soares & Soares, 1949*: 168; *Soares & Soares, 1987*: 7 [= *Xundarava holacantha* Mello-Leitão, 1927]. (Male and female syntypes; Brazil, Rio de Janeiro, Rio de Janeiro; MNRJ 53927, lost). Synonymy established by implication in *Pinto-da-Rocha et al., 2012*.

*Mischonyx holacanthus*: *Kury, 2003*: 133.

**Diagnosis.** *M. scaber* resembles *M. fidelis* in the following: median tubercles on frontal hump whitish when compared to the rest of the body (in ethanol); lateral margin of dorsal scutum with whitish tubercles when compared to the rest of the body (in ethanol); dorsal row of tubercles with an elevation before DBA; DBA with its apex directed anteriorly; no apophysis after DBA on the dorsal row of femur IV; prolateral row with median tubercles larger than the others in this row; retrolateral row with the largest apophysis on the apical third. It differs from *M. fidelis* by: lateral margin of dorsal scutum with smaller tubercles when compared to *M. fidelis*; prolateral apophysis on coxa IV with its apex directed dorsally (Fig. 22D) (prolateral apophysis with apex directed posteriorly in *M. fidelis*); retrolateral apophysis on coxa IV visible in dorsal view (not visible in *M. fidelis*); prolateral apophysis on trochanter IV small when compared to *M. fidelis*; retrolateral side of trochanter IV with three large tubercles (small tubercles in *M. fidelis*); DBA small, much smaller than the body height (almost as large as the body height in *M. fidelis*); retrolateral row with tubercles increasing in size from the base to the middle of the row (small tubercles only in *M. fidelis*); after the apophysis on the retrolateral row, there is no large tubercles (two large tubercles in *M. fidelis*).

**Mischonyx spinifrons (*Mello-Leitão, 1923*) comb. nov.** (Figs. 8A, 8C, 12D–12F)

*Weyhia spinifrons Mello-Leitão, 1923*: 137; *Roewer, 1930*: 355; *Mello-Leitão, 1932*: 283, fig. 173 (Female holotype, Brazil, Rio de Janeiro, Petrópolis; MNRJ, lost)

*Geraeocormobius spinifrons*: *Mello-Leitão, 1940*: 21; *Soares & Soares, 1949*: 172; *Soares & Soares, 1987*: 7, figs. 23–26.

*Weyhia bresslaui Roewer, 1927*: 344; *1930*: 356, pl. 6, fig. 1; *Mello-Leitão, 1931b*: 127; *1932*: 285, fig. 178; 1933b: 143 (Male and female syntypes; Brazil, Rio de Janeiro, Teresópolis; SMF 1420). Synonymy established by *Soares & Soares, 1987*.

*Geraeocormobius bresslaui*: *Mello-Leitão, 1940*: 21; *Soares & Soares, 1949*: 168.

*Geraecormobiella convexa Mello-Leitão, 1931b*: 128, fig. 16 (Male lectotype, 5 paralectotypes; Brazil, Rio de Janeiro; Rio de Janeiro; MNRJ 18203). **Syn. nov.**

*Geraeocormobius convexus*: *Soares & Soares, 1949*: 169

*Weyhia montis Mello-Leitão, 1935c*: 389, fig. 15; *1935b*: 106 (Male holotype, Brazil, Rio de Janeiro, Petrópolis, Independência; MNRJ 42461). Synonymy established by *Soares & Soares, 1987*.

*Geraeocormobius montis*: *Mello-Leitão, 1940*: 21; *Soares, 1945c*: 355; *Soares & Soares, 1949*: 170.

*Geraeocormobius cheloides* Mello-Leitão, 1940: 19, fig. 23; *Soares & Soares, 1987*: 4 [= *Geraecormobiella convexa Mello-Leitão, 1931b*] (Male holotype; Brazil, Rio de Janeiro, Rio de Janeiro; MNRJ 58236). **Syn. nov.**

**Diagnosis.** *Mischonyx spinifrons* **comb. nov.** resembles *Mischonyx tinguaensis* **sp. nov.** in the following: anterior margin of dorsal scutum with two tubercles on each side; tubercles on mesotergal area III, besides the median ones, elliptic; lateral margin of dorsal scutum with the most posterior lateral tubercles fused (forming larger tubercles); all free tergites with small tubercles; retrolateral apophysis on coxa IV apparent in dorsal view; dorsal row on leg IV with a tubercle anterior to the DBA; retrolateral row on leg IV with a large median apophysis; ventral plate with three pairs of apical MS C. It differs from *Mischonyx tinguaensis* **sp. nov.** by: median tubercles on mesotergal area III strongly compressed (elliptic but not strongly compressed laterally in *Mischonyx tinguaensis* **sp. nov**); lateral margin of dorsal scutum with small tubercles (large in *Mischonyx tinguaensis* **sp. nov**); prolateral apophysis on coxa IV smaller than trochanter IV (approximately with the same length in *Mischonyx tinguaensis* **sp. nov**); DBA not branched (branched in *Mischonyx tinguaensis* **sp. nov**); dorsal row of tubercles of leg IV with three large tubercles after DBA (without large tubercles after DBA in *Mischonyx tinguaensis* **sp. nov**); retrolateral row of leg IV with large tubercles (small in *Mischonyx tinguaensis* **sp. nov**); MS B reduced much smaller than MS A (as large as the MS A in *Mischonyx tinguaensis* **sp. nov**); MS A forming a triangle and hidden behind ventral process (forming a dorso-ventral line and apparent in *Mischonyx tinguaensis* **sp. nov.**); *flabelum* with smooth ends (serrated in *Mischonyx tinguaensis* **sp. nov**).

***Mischonyx squalidus Bertkau, 1880*** (Figs. 8B, 8D, 12G–12I)

*Mischonyx squalidus Bertkau, 1880*: 107, pl.2, fig. 38; *Roewer, 1913*: 468; *1923*: 584; *Soares & Soares, 1949*: 221; *Pinto-da-Rocha et al., 2012*: 52 (Female holotype; Brazil, Rio de Janeiro, Copacabana; ISNB).

*Ilhaia cuspidata Roewer, 1913*: 221 (Male holotype; Brazil, Rio de Janeiro, Ilha Grande, SMF 900). **Syn.nov.**

*Jlhaia cuspidata*: *Roewer, 1930*: 363 (misspelling).

*Mischonyx cuspidatus*: *Kury, 2003*: 133; *Pinto-da-Rocha et al., 2012*: 53; *Pinto-da-Rocha et al., 2014*: 4, 16–18.

*Ilhaia fluminensis Mello-Leitão, 1922*: 334; *Soares, 1943*: 56 [= *Ilhaia cuspidata Roewer, 1913*] (13 syntypes; Brazil, Rio de Janeiro, Piraí; MZSP 503). **Syn.nov.**

*Jlhaia fluminensis*: *Roewer, 1930*: 363, fig. 4 (*lapsus calami*).

*Gonazula gibbosa Roewer, 1930*: 418, fig. 32; *Pinto-da-Rocha et al., 2012*: 53 [= *Ilhaia cuspidata Roewer, 1913*] (Male holotype, Brazil, Santa Catarina, Serra Azul. SMF 1328). **Syn.nov.**

*Eduardoius granulosus Mello-Leitão, 1931a*: 95; *Soares, 1944*: 171 [= *Ilhaia cuspidata Roewer, 1913*] (male holotype; Brazil, Rio de Janeiro, Piraí; MNRJ 1479). **Syn.nov.**

*Ilhaia granulosa*: *Soares, 1943*: 56.

*Giltaya solitaria Mello-Leitão, 1932*: 467; *Kury, 2003*: 133 [= *Ilhaia cuspidata Roewer, 1913*] (Male holotype; Brazil, Rio de Janeiro, Rio de Janeiro. MNRJ 1473). **Syn.nov.**

*Eduardoius lutescens Roewer, 1943*: 44; *Soares & Soares, 1970*: 340 [= *Ilhaia cuspidata Roewer, 1913*] (Male and female syntypes; Brazil, Rio de Janeiro, Mendes. SMF 5392/58). **Syn.nov.**

*Ilhaia lutescens*: *Soares, 1943*: 56.

Taxonomic remarks: *Vasconcelos (2003)*, proposed this new combination in his dissertation. We analyzed Bertkau's original drawing (*Bertkau, 1880*, fig. 38) and the original description of *M. squalidus*, but were not able to lay hands on the holotype because it is lost. It was deposited at the Institut Royal des Sciences Naturelles de Belgique. Part of the description translated from German is presented below:

"… The first abdominal dorsal segment is almost fused with the thorax, and in general the articulation skin between each segment is not very flexible. **The first three [abdominal] segments have in their superior part a line of "dots", of which the median ones stand out in height, like little spines**." (*Bertkau, 1880*, pp. 107)

The only species that has one median armature on each free tergite in females and juveniles in the region Bertkau collected the specimen (Copacabana, Rio de Janeiro) is the traditionally called *M. cuspidatus*. Therefore, we propose that *Ilhaia cuspidata* is a junior synonym of *M. squalidus*. We know the holotype is a juvenile, based on the image in *Bertkau (1880)*, and *Roewer's (1923)* and *Kury's (2003)* statements.

**Diagnosis.** *M. squalidus* resembles *M. spinifrons* **comb. nov.** in the following: lateral margin of dorsal scutum with whitish tubercles (in ethanol); posterior tubercles on lateral margin of dorsal scutum fused; retrolateral apophysis of coxa IV visible in dorsal view; DBA with apex directed anteriorly; dorsal row on femur IV with three tubercles after DBA, on distal half; retrolateral row on femur IV with median apophysis more developed than the others in this row; ventral side of ventral plate without microsetae on distal half; lateral projections of ventral plate projected dorsally and behind ventral projection of glans; MS A forming a triangle; MS B reduced; apical groove of ventral plate reaching the line of the most basal MS C. It differs from *M. spinifrons* **comb. nov.** in the following: median tubercles on mesotergal area III strongly compressed and large (small and elliptic but not strongly compressed laterally in *M. spinifrons* **comb. nov.**); prolateral apophysis on coxa IV approximately same length as trochanter IV (smaller in *M. spinifrons* **comb. nov.**); Free Tergites I–III with median apophysis (without median apophysis in *M. spinifrons* **comb. nov.**); prolateral row with median tubercles larger

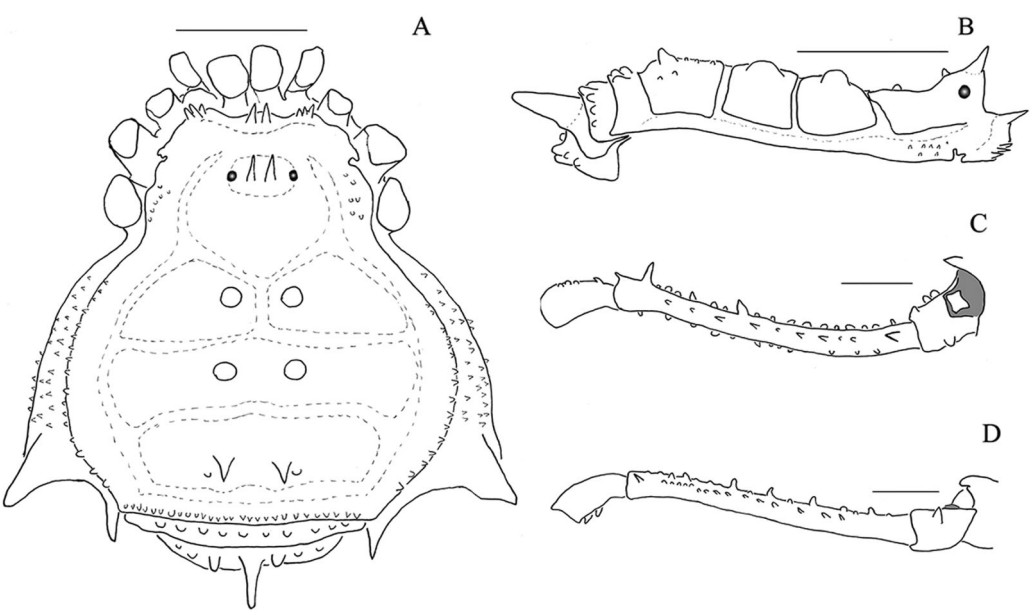

**Figure 19** *Mischonyx minimus* **sp. nov. male holotype drawings.** (A) Dorsal view; (B) lateral view; (C) dorsal view of the right leg; (D) retrolateral view of the right leg. The tubercles painted in gray are whitish in ethanol. Scale bars = 1 mm.

than the others in this row (all tubercles subequal in size in *M. spinifrons* **comb. nov.**); retrolateral row on femur IV with several (7–8) large tubercles basal to median apophysis (three tubercles basal, followed by a gap and one tubercle after this gap in *M. spinifrons* **comb. nov.**).

## New species description

***Mischonyx minimus* sp. nov.**
(Figs. 19, 14A–14C, 5A and 5C)

**Type material.** BRAZIL. Rio de Janeiro: Teresópolis (Parque Nacional da Serra dos Órgãos, Barragem Beija-flor, 22°26′16.4″S 43°36′35.4″W), C. Gueratto & M. Abrão leg., 29.VII.2017, male holotype (MZSP76524); same data, males and females paratypes, (IBSP); same data, A. Benedetti et al. leg., 30.IV.2014.

**Etymology.** From the Latin adjective *minimus, a, um* meaning small, little. This is due to its reduced size when compared to other *Mischonyx* species, specially *Mischonyx arlei* **comb. nov.**, sister species of *M. minimus* **sp. nov.**.

**Diagnosis.** *Mischonyx minimus* **sp. nov.** resembles *M. arlei* **comb. nov.** in the following: mesotergal area I with pair of well-developed median tubercles, paler (whitish) than rest of body (dark brown); median armatures on mesotergal area III are spines; lateral margin of dorsal scutum with several small tubercles; free tergite II with a well-developed median apophysis; prolateral apophysis on coxa IV small and pointing posteriorly; retrolateral side of trochanter IV with two tubercles; femur IV with several small apophyses on dorsal and retrolateral row of tubercles; femur IV with a well-developed apical

tubercle on prolateral and retrolateral rows of tubercles; ventral plate of penis with three subdistal MS C on each side; MS B smaller than MS A; *flabellum* with serrated ends. It differs from *M. arlei* **comb. nov.** in the following: reduced size (3–3.5 mm) (7–8 mm in *M. arlei* **comb. nov.**); mesotergal area II with median tubercles whitish and as large as the median tubercles on mesotergal area I (dark brown and smaller than the ones on mesotergal area I in *M. arlei* **comb. nov.**); basitarsus II with four segments (seven in *M. arlei* **comb. nov.**); leg IV not curved (straight) in dorsal view (curved in *M. arlei* **comb. nov.**); MS D well-developed (reduced in *M. arlei* **comb. nov.**).

**Description.** Male holotype: *Dorsum* (Figs. 19, 5A, 5C): Measurements: Dorsal scutum: L: 3.2; W:2.9; Prosoma: L:1.3; W: 1.6. Femur IV: 4.4. Scutum outline γP, widest at mesotergal area II. Anterior margin of carapace with three tubercles on each side, approximately the same size. Frontal hump high, with two spines the same color as rest of body (in ethanol), curved towards one another. Anterior region of ocularium smooth, ocularium with one pair of median tubercles (as tall as the ocularium height). Posterior region of the ocularium with one pair of small tubercles, right behind median tubercles. Lateral margin of prosoma with numerous small tubercles. Posterior portion of prosoma with a pair of tubercles. Besides these tubercles, prosoma has a low density of granules. Dorsal scutum divided into three mesotergal areas, with low density of granules (*DaSilva & Pinto-da-Rocha, 2010*). Areas: Area I divided by a median longitudinal groove, with a pair of whitish large median tubercles and no granules; area II with a pair of large whitish median tubercles, same size as the tubercles on Area I without granules; Area III with a pair of dark median sharp spines, smaller than the other armatures on other mesotergal areas, a pair of tubercles posterior to median spines. Lateral margins of dorsal scutum with a row of small tubercles, approximately the same size, extending from the middle of area I until the posterior margin of Area III; no fusion of tubercles. Posterior margin of dorsal scutum with a line of small tubercles. Free tergite I with a line of small tubercles approximately the same size. Free tergite II with a large sharp median apophysis and two large tubercles, lateral to the median apophysis; free tergite III with a line of small tubercles. Dorsal anal operculum with small sparse tubercles. *Venter.* Coxa I with several sparse tubercles, larger than the ones on other coxa. Coxa II with sparse numerous granules. Coxa III with an anterior and a posterior basal-apical row of tubercles; coxa IV with sparse numerous granules. Ventral anal operculum with granules. *Chelicerae.* Segment II with several setae, mainly apical. Fix and movable fingers with seven teeth each. *Pedipalps.* Venter of trochanter with few sparse tubercles; tibia setation: prolateral IIi, retrolateral IiIi. Tarsal setation: prolateral IiI, retrolateral III, ventral side with two baso-apical lines of setae. *Legs.* Leg I: trochanter with several ventral tubercles, femur, patella and tibia with granules. Leg II: Trochanter II with several ventral tubercles; femur, patella and tibia with granules. Leg III: trochanter with several ventral tubercles; femur, patella and tibia with granules; Leg IV: Coxa IV: robust apical oblique prolateral apophysis, smaller than the trochanter size; large retrolateral apophysis, visible in dorsal view. Trochanter IV: prolateral small blunt apophysis; retrolateral side with a line of three large tubercles, two slightly more ventral. Femur IV: long, thin and straight; all tubercles on prolateral row

approximately the same size; DBA small, unbranched, conic, sharp, pointing upwards; dorsal row with several small tubercles after DBA; retrolateral row of with several small tubercles and two more developed tubercles on the apical half; all tubercles on the ventral row small. Tarsal formula: 6(3)-6(3)-4-5. *Male genitalia* (Figs. 14A–14C). Ventral plate: Ventral surface covered with microsetae; pronounced apical groove (reaching the line of the first basal MS C); lateral lobes basal when compared to other species (*e.g., Mischonyx intervalensis* **sp. nov.**); three sub-apical helicoidal MS C on each side; two MS E, ventral and in the same baso-apical orientation of MS C; long MS D when compared to other species (*e.g., Mischonyx intervalensis* **sp. nov.**), basal relative to MS C and in the same dorso-ventral orientation of MS C; three spatular MS A, forming a diagonal baso-apical line; one reduced MS B, much smaller than MS A. Glans: Small dorsal process; flabelum triangular, with serrated apex; stylus with subapical microsetae, with the apex inclined relative to the penis axis and keeled. *Color.* Dark brown; pedipalps and trochanters I–III yellow.

*Female.* Unknown.

### *Mischonyx intervalensis* sp. nov.
(Figs. 4A and 4C, 14D–14F, 20)

**Type material.** BRAZIL. São Paulo: Ribeirão Grande (Parque Estadual Intervales, 24°15′27.1″S 48°16′23.0″W), C. Gueratto et al. leg., 25.III.2017, male hololtype (MZSP76525); same data, males and females paratypes (IBSP); ditto males and females paratypes (MNRJ); same data, Ribeirão Grande (Parque Estadual Intervales, 24°15′27.1″S 48°16′23.0″W), F. Carbayo et al. Leg., 12–14.XII.2008, males and females paratyes (SMF).

**Etymology.** Species name derives from the type locality, Parque Estadual Intervales. "Intervales" + the suffix -ēnsis, -ēnse, to form an adjective.

**Diagnosis.** It resembles *Mischonyx anomalus* in the following: Anterior margin of dorsal scutum with two tubercles on each side; Areas I and II with small median tubercles; area III with well-developed and elliptic median tubercles; other tubercles on area III rounded; all free tergites with small tubercles; retrolateral row of leg IV with large median apophysis; retrolateral row of leg IV with several well-developed tubercles. It differs from *M. anomalus* in the following: prolateral apophysis of coxa IV with ventral process and basal tubercle (not present in *M. anomalus*); retrolateral side of trochanter IV with three tubercles (one in *M. anomalus*); DBA of leg IV branched and dorsal branch is the largest (not branched in *M. anomalus*); one apophysis on the dorsal row of tubercles of leg IV after DBA (three in *M. anomalus*); tubercles on prolateral row of tubercles on leg IV small and subequal in size (median tubercles larger in *M. anomalus*); ventral plate with the same approximate height and width (square-shaped) (higher than wider in *M. anomalus*); lateral processes of the ventral plate medial (basal in *M. anomalus*).

**Description.** Male holotype: *Dorsum* (Figs. 4A and 4C, 20): Measurements: Dorsal scutum: L: 4.5; W:4.6; Prosoma: L:1.8; W: 2.4. Femur IV: 3.9. Scutum outline γP, widest at area II. Anterior margin of carapace with two tubercles on each side, approximately the

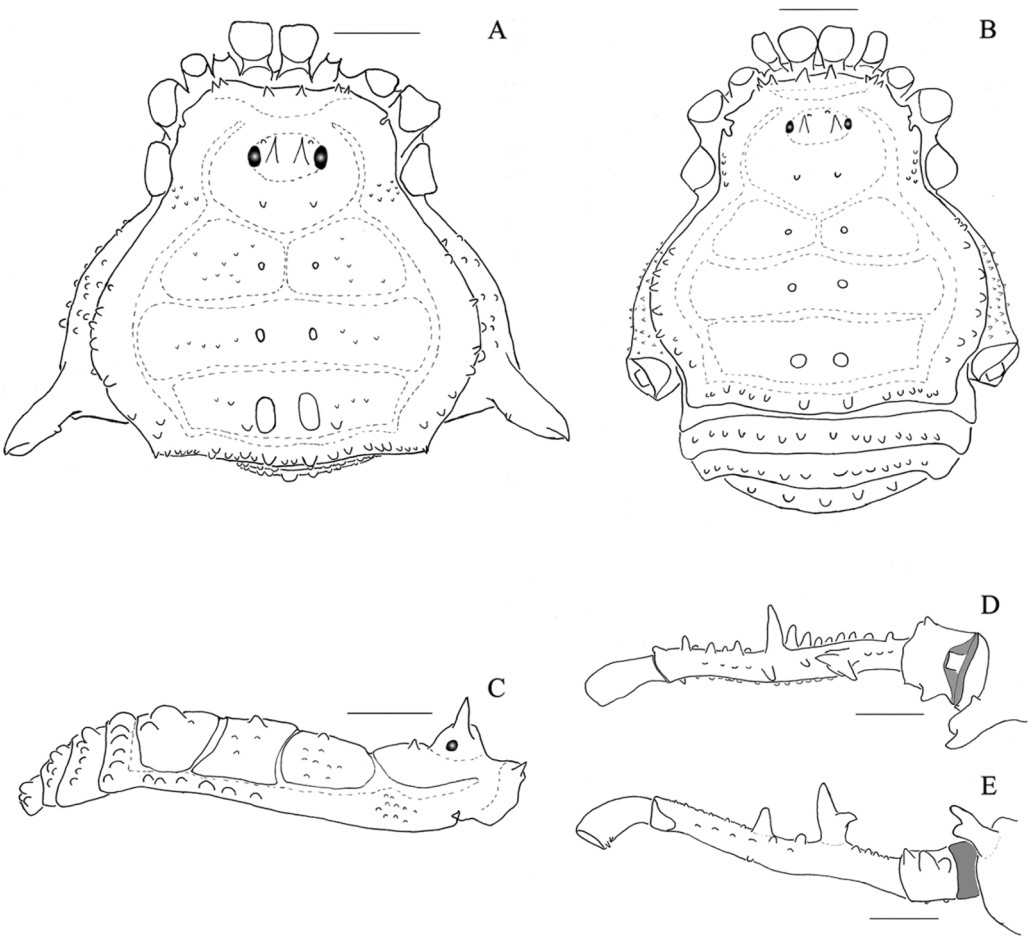

**Figure 20** *Mischonyx intervalensis* **sp. nov. male holotype and female paratype drawings.** (A & C), Male holotype, dorsal and lateral view, respectively; (B), Female paratype, dorsal view; (D & E) Right leg of the male holotype right, dorsal and retrolateral view, respectively. Scale bars = 1 mm.

same size. Frontal hump high, with two tubercles the same color as rest of body (in ethanol). Anterior surface of the ocularium with one pair of tubercles, one pair of median tubercles/spines (taller than the ocularium height). Anterior surface of ocularium with one pair of small tubercles, right before the eyes. Lateral margin of prosoma with numerous small tubercles. Posterior part of prosoma with a pair of tubercles. Besides these tubercles, prosoma with low density of granules. Dorsal scutum; Area I divided by median longitudinal groove, with a pair of dark median tubercles and few sparse granules; Area II with a pair of dark median tubercles slightly larger than the tubercles on Area I and few sparse granules; Area III with a pair of dark median elliptic tubercles, larger than the ones on the other mesotergal areas, a pair of rounded tubercles posterior to the median elliptic ones and few sparse granules. Lateral margins of dorsal scutum with a row of small tubercles, increasing in size posteriorly and from sulcus I to the posterior margin of area III; no fusion of tubercles. Posterior margin of dorsal scutum with a line of small tubercles, with the median ones slightly larger than the rest. Dorsal scutum with average

density of granules. Free tergites I–II with a line of small tubercles of the same approximate size. Free tergite III with a row of tubercles larger than the ones on the other free tergites and central tubercle slightly larger than the others. Dorsal anal operculum with small sparse tubercles. *Venter*. Coxa I with several sparse tubercles, larger than the ones on other coxae. Coxae II–IV with sparse numerous granules. Ventral anal operculum with granules. *Chelicerae*. segment II with several setae, mainly apical. Fixed finger with eight and movable finger with 12 teeth. *Pedipalps*. Ventral side of trochanter with few sparse tubercles; tibia setation: prolateral IiIi, retrolateral IiI. Tarsal setation: prolateral IiI, retrolateral II, ventral side with two baso-apical lines of setae. *Legs*. Leg I: trochanter, femur, patellae and tibia with granules. Leg II: Trochanter II with two retrolateral tubercles; femur, patella and tibia with granules. Leg III: trochanter, femur, patella and tibia with granules. Leg IV: coxa IV: robust apical prolateral apophysis, slightly inclined relative to the axis of the base of coxa IV, with ventral process and basal tubercle, with the approximate trochanter size; retrolateral apophysis small, not visible in dorsal view. Trochanter IV: prolateral small blunt apophysis; retrolateral side with a line of three large tubercles, two slightly more ventral. Femur IV: short and robust; all tubercles on prolateral row with approximately the same size; dorsal row of tubercles with a large tubercle before the DBA, DBA branched with the largest branch pointing upwards, one large tubercle after DBA; retrolateral row of with a large median apophysis, eight large tubercles before, three large (yet smaller than the ones anterior to the median apophysis) and three small tubercles posterior to the median apophysis, intercalated; all tubercles on the ventral row small. Tarsal formula: 3(3)-7(3)-4-5. *Male genitalia* (Figs. 14D–14F). Ventral plate: Ventral surface with microsetae on the whole extension; pronounced apical groove (reaching the line of the most basal MS C); lateral process median when compared to other species (*e.g.*, *Mischonyx tinguaensis* **sp. nov.**); three apical helicoidal MS C on each side; two MS E, ventral and in the same baso-apical orientation of MS C; one small MS D, basal relative to MS C and in the same dorso-ventral orientation of MS C; three spatular MS A, forming a parable line; one spatular MS B, smaller than MS A. Glans: Small dorsal process; flabellum triangular, with serrated margin; stylus with subapical microsetae, with the apex inclined relative to the penis axis and keeled. *Color*. Brown; dorsal scutum with yellowish tones; pedipalps and trochanters I–III yellow.

*Female*. (paratype; MZSP): Measurements: Dorsal scutum: L: 4.2; W: 4.0. Prosoma: L: 1.3; W: 2.0; Femur IV: L: 3.9. Dorsal scutum outline α, with a constriction at the area III and evident *coda*; small median tubercles on each area; median tubercles on area III rounded; lateral tubercles of the dorsal scutum small and the most posterior are not fused; absence of prolateral and retrolateral apophysis on coxa IV; trochanter and femur IV unarmed.

### *Mischonyx tinguaensis* sp. nov.
(Figs. 9A and 9B, 14G–14I, 21)

**Type material.** BRAZIL. Rio de Janeiro: Nova Iguaçu, (Reserva Biológica Tinguá/ RPPN CEC/Tinguá, 22°35′23.9″S 43°26′25.7″W), C. Sampaio, F. Uemori & C. T. Olivares leg., 04–06.IV.2012, male holotype (MZSP76526).

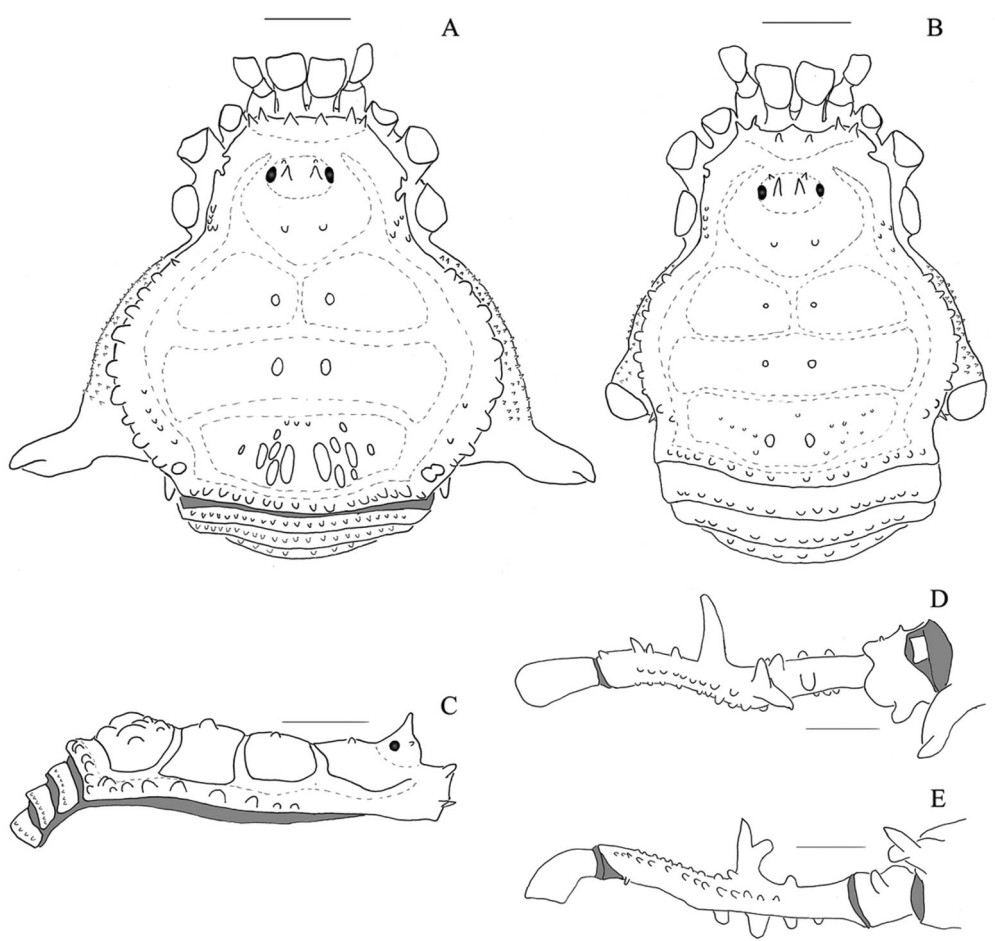

**Figure 21 *Mischonyx tinguaensis* sp. nov. male holotype and female paratype drawings.** (A & C) Male holotype, dorsal and lateral view, respectively; (B) female paratype, dorsal view; (D & E) right leg of the male holotype right, dorsal and retrolateral view, respectively. Scale bars = 1 mm.

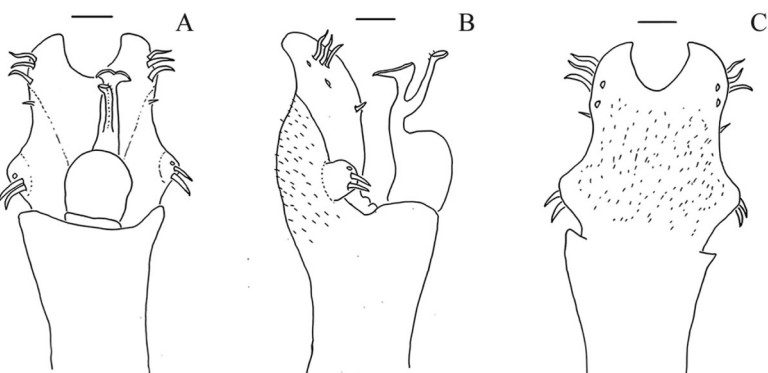

**Figure 22  Penis of *Mischonyx reitzi*.** (A–C) Dorsal, right lateral and ventral views, respectively, of the penis of Mischonyx reitzi Scale bars = 1 μm.

**Etymology.** Species name derives from "Tinguá", due to its first collecting locality, Reserva Biológica Tinguá, type and only locality registered for this species + the suffix *-ēnsis*, *-ēnse*, in order to form an adjective.

**Diagnosis.** It resembles *Mischonyx spinifrons* **comb. nov.** in the following: anterior margin of dorsal scutum with two tubercles on each side; several tubercles on area III elliptical; lateral margin of dorsal scutum with the most posterior lateral tubercles fused (forming larger tubercles); all free tergites with small tubercles; retrolateral apophysis on coxa IV apparent in dorsal view; dorsal row on leg IV with a tubercle anterior to the DBA; retrolateral row on leg IV with a large median apophysis; ventral plate with three pairs of apical MS C. It differs from *M. spinifrons* **comb. nov.** by: median tubercles on area III elliptic but not strongly compressed laterally (strongly compressed in *M. spinifrons* **comb. nov.**); large tubercles on lateral margin of dorsal scutum (small in *M. spinifrons* **comb. nov.**); prolateral apophysis on coxa IV approximately same length as trochanter IV (smaller in *M. spinifrons* **comb. nov.**); DBA branched (not branched in *M. spinifrons* **comb. nov.**); dorsal row of tubercles of leg IV without large tubercles after DBA (three large tubercles after DBA in *M. spinifrons* **comb. nov.**); tubercles on the basal half of the retrolateral row of leg IV small (some are large in *M. spinifrons* **comb. nov.**); MS B as large as the MS A (reduced in *M. spinifrons* **comb. nov.**); MS A forming a dorso-ventral line and apparent (forming a triangle and hidden behind the ventral process); *flabellum* with serrated on margin (smooth in *M. spinifrons* **comb. nov.**).

**Description.** Male holotype: *Dorsum* (Figs. 9A and 9B, 21): Measurements: Dorsal scutum: L: 4.1; W:4.2; Prosoma: L:1.6; W: 2.1. Femur IV: 4.0. Scutum outline γP, widest at mesotergal area II. Anterior margin of carapace with two tubercles on each side, with approximately the same size. Frontal hump high, with two whitish tubercles (in ethanol). Anterior surface of the ocularium with one pair of tubercles, one pair of median tubercles (as tall as the ocularium height). Lateral margin of prosoma with numerous small tubercles. Posterior part of prosoma with a pair of tubercles. Besides these tubercles, prosoma has a low density of granules (*DaSilva & Pinto-da-Rocha, 2010*). Dorsal scutum: area I divided by a median longitudinal groove, with a pair of dark median tubercles; area II with a pair of dark median tubercles slightly larger than the tubercles on area I; area III with a pair of dark median elliptic tubercles, larger than the ones on the other areas, and some sparse elliptic tubercles. Lateral margins of dorsal scutum with a row of whitish (in ethanol) large tubercles, reaching the posterior margin of area III; most posterior tubercles fused, forming large tubercles. Posterior margin of dorsal scutum with a line of white (in ethanol) small tubercles of similar size. Dorsal scutum with low density of granules. All free tergites with a line of small tubercles of the same approximate size. Dorsal anal operculum with small sparse tubercles. *Venter*. Coxa I with several sparse tubercles, larger than the one in other coxa. Coxa II with sparse tubercles; the apical are larger. Coxae III and IV with granules. Ventral anal operculum with granules. *Chelicerae*. Middle segment with several setae, mainly in the apical. Fixed and movable fingers with nine teeth each. *Pedipalps*. Tibia setation: prolateral IiIi, retrolateral IiI. Tarsal setation: prolateral II,

retrolateral II, ventral side with two baso-apical lines of setae. *Legs.* Leg I: trochanter, femur, patella and tibia with granules. Leg II: Trochanter II with two retrolateral tubercles; femur, patella and tibia with granules. Leg III: trochanter, femur, patella and tibia with granules. Leg IV: Coxa IV: robust apical transversal prolateral apophysis, with ventral process, with the approximate trochanter size; retrolateral apophysis visible in dorsal view. Trochanter IV: prolateral small blunt apophysis; retrolateral side with small tubercles. Femur IV: short and robust; all tubercles on prolateral row with approximately the same size; dorsal row of tubercles with a large tubercle before the DBA, DBA branched with the largest branch pointing upwards, small tubercles after DBA; retrolateral row of with a large median apophysis, four large tubercles before and three large tubercles posterior to the median apophysis; all tubercles on the ventral row small. Tarsal formula: 4(3)-8(3)-8-5. *Male genitalia* (Fig. 14G–14I). Ventral plate: Ventral surface with microsetae on basal 2/3; pronounced apical groove (reaching the line of MS B); lateral process basal when compared to other species (*e.g., Mischonyx intervalensis* **sp. nov.**); three apical helicoidal MS C on each side; two MS E, ventral and slightly basal relative to MS C; small MS D, basal relative to MS C and between MS E and MS C; four spatular MS A, forming a diagonal baso-apical line; one spatular MS B, same size as MS A. Glans: Small dorsal process; flabellum triangular with serrated margin; no information regarding stylus (broken in the analyzed specimen). *Color.* Brown; dorsal scutum with tones of yellow; pedipalps and trochanters I–III yellow. *Female.* (paratype; MZSP): Measurements: Dorsal scutum: L: 3.9; W: 3.4. Prosoma: L: 1.5; W: 2.0; Femur IV: L: 3.8. Dorsal scutum outline α, with a constriction at the chelicerae, area III and evident *coda*; small median tubercles on each area; median tubercles on Area III rounded; lateral tubercles of the dorsal scutum small and the most posterior are not fused; absence of prolateral apophysis on coxa IV, but with a small retrolateral apophysis; trochanter and femur IV unarmed.

### *Gonyleptes* Kirby, 1818

*Gonyleptes* Kirby, 1818: 450 (type species *Gonyleptes horridus* Kirby, 1818, by subsequent designation, *Roewer, 1913*)

   *Anoploleptes* *Piza, 1940*: 56; *Soares, 1943*: 53; *Kury, 2003*: 133 [= *Mischonyx* Bertkau, 1818] (type species *Anoploleptes dubium* *Piza, 1940*, by original designation).

REMARKS: We reestablished *Anoploleptes* as a subjective junior synonym of *Gonyleptes* as first established by *Soares (1943)*.

### *Gonyleptes antiquus* Mello-Leitão, 1934

*Gonyleptes antiquus* *Mello-Leitão, 1934*: 415. fig. 6; *1935b*: 106. *Soares, 1943*: 53 (Male holotype; Brazil, São Paulo; IBSP 11).
*Paragonyleptes antiquus*: *Soares, 1945a*: 11, fig. 1.
*Mischonyx antiquus*: *Kury, 2003*: 133.
*Anoploleptes dubium* *Piza, 1940*: 56, fig. 4. (Male holotype; Brazil, São Paulo, Juquiá; MZSP 401).

REMARKS: *Gonyleptes antiquus* returns to its former genus, so the original combination is reestablished (see discussion below).

## Identification key for Mischonyx males

1. Median armature on area I larger and paler (in ethanol) than those on area III (paler than the general body color) (Fig. 5A) ................................................................. 2

    Median armature on area I smaller and the same color (in ethanol) as those on area III (paler than the general body color) (Fig. 5B) ........................................................ 3

2. Small individuals (3–3.5 mm of dorsal scutum length); median armature on area II (in ethanol) and I the same color (paler than the body) (Fig. 5A) ........................................ .......................................................................................*Mischonyx minimus* **sp. nov.**

    Large individuals (7–8 mm of dorsal scutum length); median armature on area II (in ethanol) and III the same color (darker than the body) (Fig. 1B).............. *Mischonyx arlei*

3. Posterior lateral mid-bulge tubercles fused, forming larger tubercles, paler than rest of the body (Fig. 8A) ..................................................................................... 4

    Lateral mid-bulge tubercles not fused (Fig. 5B). . . . . . . . . . . . . . . . . . . . . . . . . . . . . . . . . . . . . . . . . . . 6

4. Ellipsed tubercles on mesotergal area III strongly compressed laterally; one clearly more developed apophysis on leg IV, with retrolateral row of tubercles (Fig. 8A) ........................ ............................................................................................... *Mischonyx spinifrons*

    Ellipsed tubercles on area III not strongly compressed laterally; more than one developed apophysis on leg IV, with retrolateral row of tubercles...................................... 5

5. DBA digitiform and uniramous (Fig. 6A).................................................. *Mischonyx poeta*

    DBA birramous (Fig. 9) ...................................................... *Mischonyx tinguaensis* **sp. nov.**

6. At least one mesotergal area with well-developed median armature (*e.g.*, Fig. 6B)..... 7

    Mesotergal areas with small tubercles subequal in size (*e.g.*, Fig. 4A) ........................... 9

7. All mesotergal areas and posterior part of dorsal scutum with well-developed median armature (Fig. 8B) ......................................................................... *Mischonyx squalidus*

    Mesotergal areas II –III only with well-developed median armature (*e.g.*, Fig. 6B).... 8

8. DBA branched, retrolateral branch the largest; prolateral row of tubercles on leg IV with medial tubercles more developed (Fig. 6B) ....................................... *Mischonyx processigerus*

    DBA falciform, not branched; prolateral row of tubercles on leg IV with tubercles of the same size (Fig. 3A) ......................................................................... *Mischonyx insulanus*

9. Median tubercles on mesotergal area III small (*e.g.*, Fig. 7A) ....................................... 10

Median tubercles on mesotergal area III well-developed (*e.g.*, Fig. 3B)............................. 11

10. Leg IV robust, with well-developed armature; DBA well-developed; dorsal row of tubercles on leg IV with four well-developed tubercles after DBA;........................................ ................................................................................................... *Mischonyx clavifemur*

    Leg IV long and thin, with few well-developed armatures located terminally; DBA small and sharp; without dorsal row of tubercles after DBA (Fig. 3B) ................................. .......................................................................................................*Mischonyx intermedius*

11. DBA branched (*e.g.*, Fig. 4C) ...................................................................................12

    DBA not branched (*e.g.*, Fig. 1C)........................................................................ 13

12. Retrolateral branch of DBA evidently larger than other branch; two apophysis on the leg IV dorsal row of tubercles, after DBA; prolateral apophysis of coxa IV with a prominent ventral process (Fig. 4A) ........................................................ *Mischonyx intervalensis* **sp. nov.**

Both branches of DBA of the same size; two well-developed apophyses on leg IV retrolateral row of tubercles (Fig. 7A) ........................................................ *Mischonyx reitzi*

13. DBA robust and sharp, with a tubercle emerging from median part and almost as high as the entire body (*e.g.*, Fig. 2D) ................................................................. 14

DBA smaller than the body height.................................................................. 15

14. DBA pointing upwards; after DBA, only one well-developed tubercle on the dorsal row (Fig. 5B) .................................................................... *Mischonyx parvus*

DBA pointing anteriorly; no well-developed tubercles on dorsal row, after DBA; lateral mid-bulge tubercles clearer than the general body color (in ethanol) (Fig. 2B).................... ................................................................................ *Mischonyx fidelis*

15. DBA the same approximate size as other tubercles on dorsal row (Fig. 4B) .................. ................................................................................ *Mischonyx kaisara*

DBA more developed than tubercles on dorsal row (Fig. 1A) ..... *Mischonyx anomalus*

One extra row of tubercles between dorsal and prolateral rows; median tubercles prolateral row of tubercles of Leg IV more developed; one apophysis on the leg IV terminal third of the retrolateral row of tubercles (Fig. 7B) ......................................*Mischonyx scaber*

## DISCUSSION

### Biogeographical remarks

In general, harvestmen in the Atlantic Forest have a high degree of endemism (*Pinto-da-Rocha, DaSilva & Bragagnolo, 2005*). Throughout the order, species distributions are restricted to specific areas of few thousands of square kilometers, with a few exceptions (*e.g.*, *Pinto-da-Rocha, DaSilva & Bragagnolo, 2005*). The distribution of most species of *Mischonyx* are consistent with this pattern. One exception is *M. squalidus*. There are records of this species from the southeastern state of Espirito Santo to the southern state of Rio Grande do Sul. It occurs not only in Atlantic Rainforest but also in cerrado areas (Figs. 15, S1 and S2), where the climate is drier (*Resende, Pinto-da-Rocha & Bragagnolo, 2012*). *Mestre & Pinto-da-Rocha (2004)* demonstrated that this species is synanthropic. It is able to thrive in environments like residential areas and agricultural areas. This characteristic may explain its wide distribution, since it helps these hasvestmen to disperse and colonize new areas more efficiently than most other species.

The distribution area of most *Mischonyx* species is restricted to only one or few records that are in close proximity to each other. This is consistent with the hypothesis that harvestmen have a high degree of endemism (*DaSilva, Pinto-da-Rocha & Morrone, 2017*). Serra do Órgãos, Mantiqueira, south coast of Rio de Janeiro and Serra do Mar areas of endemism hold 11 from the 16 species of the genus. According to *Pinto-da-Rocha, DaSilva & Bragagnolo (2005)* and *DaSilva, Pinto-da-Rocha & Morrone (2017)*, the southern coast of Rio de Janeiro and Serra dos Órgãos areas are the most species rich. This is

supported by our findings and is an important piece of information for conservation, since the few remaining harvestmen habitats are under the impact of anthropic changes (*Morellato & Haddad, 2000*). To maintain the diversity of the entire group, these endemic areas need to be better protected (*DaSilva, Pinto-da-Rocha & Morrone, 2017*; *Nogueira et al., 2019a*, *2019b*).

## Divergence time of Mischonyx clade

We are going to work with the Bayesian hypothesis to discuss divergence time and biogeography. BM is the preferred optimality criteria for estimating divergence time and there were no significant differences in the relationships among the internal branches of the topologies recovered using BM and TE (MP3 and ML3).

Two previous publications on two gonyleptid genera of the Atlantic Forest dated the divergence time of clades: *Bragagnolo et al. (2015)*, using *Promitobates*, and *Peres et al. (2019)*, using *Sodreana*. The divergence time of *Mischonyx* (~50 Mya) is consistent with the estimates obtained for *Promitobates*. *Sodreana* diverged more recently (~35.5 Mya) and occurs in a more restricted area than the other two genera (from the southern state of Paraná to the southern limit of Serra do Mar in the state of São Paulo). *Promitobates* occurs from the state of Santa Catarina to the northern edge of the state of São Paulo and *Mischonyx* occurs from Santa Catarina to the northern portion of the state of Rio de Janeiro (excluding *M. squalidus*, which is more widely distributed). The wider distribution of the last two genera may be a function of their older diversification times.

As stated by *DaSilva, Pinto-da-Rocha & Morrone (2017)*, "The main geographical barriers associated with the general historical patterns are the Valleys of the Doce, Paraíba do Sul, and Ribeira do Iguape rivers and the Todos os Santos Bay". Within *Mischonyx*, the split between the two major lineages occurred at ~45 Mya, which is consistent with the formation of Valley of Ribeira do Iguape River, 50–56 Mya (*Almeida & Carneiro, 1998*; *Pinto-da-Rocha, DaSilva & Bragagnolo, 2005*; *DaSilva, Pinto-da-Rocha & Morrone, 2017*).

In one of the lineages (Fig. 16), the split dividing species from SMSP from the species from SSP, PR and SC occurred at ~48 Mya. This could be the result of the rise of Serra do Mar (65–50 Mya) (*Almeida & Carneiro, 1998*; *Pinto-da-Rocha, DaSilva & Bragagnolo, 2005*). Still inside this lineage, the split between *M. intervalensis* **sp. nov.**, a species occurring at the northern portion of Ribeira do Iguape River (SSP AoE), from the species from the southern portion of this river (PR and SC AoE) occurred at ~28 Mya. The timing of this split is consistent with the results of *Peres et al. (2019)* on the split of *Sodreana* species from the north and south of this valley. After the valley was formed, it passed went through uplift and denudation events persisting from the Upper Cretaceous to the Paleogene/ Neogene (*Franco-Magalhães, Hackspacher & Saad, 2010*; *Franco-Magalhães et al., 2010*), a period consistent with the split mentioned above.

Inside the other lineage (Fig. 16), the first split occurred at ~45 Mya, when *M. intermedius* diverged from the remaining species. This species is the only one from Esp AoE. It is very likely that the distensive tectonic activity from the tertiary period, which separated the Rio Doce, Paraíba do Sul and São Francisco basins (*Cherem et al., 2012*; *Morais et al., 2005*), isolated it from the sister species from Org, LSRJ and Mnt AoE. Many

other studies with different taxa corroborate the relevance of the Doce River disjunction in shaping biogeographical patterns (*Müller, 1973*; *Prance, 1982*; *Amorim & Pires (1996)*; *Pellegrino et al., 2005*; *Sigrist & Carvalho, 2009*; *Brunes et al., 2010*; *Thomé et al., 2010*; *Silva et al., 2012*; *Cabanne et al., 2014*; *DaSilva, Pinto-da-Rocha & Morrone, 2017*). The split of *M. processigerus* (Mnt AoE) from species from LSRJ and Org occurred at ~29 Mya, agreeing with the formation of the Paraíba do Sul Valley and its river change of course, during the Oligocene-Miocene (*Almeida & Carneiro, 1998*; *Pinto-da-Rocha, DaSilva & Bragagnolo, 2005*; *Cherem et al., 2012*)

In general, the divergence times of *Mischonyx* species are older than 5 Mya (except for *M. clavifemur* **comb. nov.** diverging from *M. reitzi* **comb. nov.** and *M. parvus* **comb. nov.** diverging from *M. squalidus*). This is consistent with the speciation events in *Promitobates* (*Bragagnolo et al., 2015*). Authors who support the Pleistocene refugia hypothesis have proposed that it happened beginning ~5 Mya (*Ravelo et al., 2004*, *Carnaval & Moritz, 2008*; *Carnaval et al., 2009*; *Holbourn et al., 2014*). Therefore, the ancient cooling of the Miocene/Pliocene probably shaped most of the divergences between species inside the genus and the Pleistocene refugia contributed to the most recent speciation events to shape the extant population diversity.

Finally, it is important to stress that *M. squalidus* appears in all analyses using molecular and TE as sister to *M. parvus* **comb. nov.**, inside the clade with species from LSRJ. Based on that we conclude that it probably diverged at this AoE in the past and, later, spread all over the Atlantic Forest and Cerrado areas, as discussed in the biogeographical session. Therefore, from now on, in discussions regarding the AoE and the relationship among clades, we will consider *M. squalidus* as belonging to LSRJ AoE.

## The hypothesis of TE under maximum likelihood as the optimality criteria (ML3)

We choose ML3 grounded in the following arguments.

In the results of MP3, *M. tinguaensis* **sp. nov.** has more than 30 autapomorphies. This long branch encompasses almost one third of all morphological characters coded in the analysis. Additionally, comparing this situation with the number of morphological changes in other harvestmen phylogenies (*Bragagnolo & Pinto-da-Rocha, 2012*; *DaSilva & Gnaspini, 2010*; *DaSilva & Pinto-da-Rocha, 2010*; *Pinto-da-Rocha & Bragagnolo, 2010*), we believe that it it is unlikely that this single species has accumulated so many changes and that the results of ML3 are more likely.

Another reason to choose ML3 is the position of *M. tinguaensis* **sp. nov.** in MP3 (Fig. 18A), inside the clade formed strictly by *M. spinifrons* **comb. nov.**. It separates the seven sequenced specimens into two polyphyletic lineages. The polyphyly of *M. spinifrons* **comb. nov.** seems odd, since the individuals analyzed by us are morphologically identical and there are few site changes in their sequences. In contrast, ML3 (Fig. 17) places *M. tinguaensis* **sp. nov**. as the lineage diverging after *M. processigerus*. This odd placement of *M. tinguaensis* **sp. nov.** inside the clade formed by another species' clade in MP3 seems to contribute to the fact that this species has 30 autapomorphies, as discussed in the last paragraph. Similarly, MP3 and B3 (Fig. 18B) place *M. insulanus* inside the clade formed by

*M. kaisara*, splitting this last species into two polyphyletic lineages. To match this hypothesis, the character changes in this clade containing *M. insulanus* and *M. kaisara*, in B3, has several homoplastic changes, as in *M. tinguaensis* **sp. nov.** In ML3, instead of this split, *M. kaisara* is monophyletic and sister to *M. insulanus*. Therefore, we prefer ML3, since it does not separate exemplars from the same species into polyphyletic lineages.

Finally, in B3 (Fig. 18B), the clade formed by species from LSRJ AoE has two species from Org AoE, *M. scaber* and *M. tinguaensis* **sp. nov.**. When comparing our results with the biogeographic (*e.g.*, *DaSilva, Pinto-da-Rocha & Morrone, 2017*), phylogeographic (*e.g.*, *Bragagnolo et al., 2015*; *Peres et al., 2019*) and phylogenetic (*e.g.*, *DaSilva & Gnaspini, 2010*) works on Atlantic Forest harvestmen, we conclude that species from one AoE are rarely clustered with species from another AoE, given the high degree of endemism discussed in the biogeographic session (*Pinto-da-Rocha, DaSilva & Bragagnolo, 2005*; *DaSilva, Pinto-da-Rocha & Morrone, 2017*). In conclusion, we prefer ML3 to B3, since the clades recovered by it reflect biogeographical hypothesis from other researches.

## Diagnosis of previews authors

Although *Vasconcelos (2005a)* described some characteristics of *Mischonyx*, and noted two possibly diagnostic characters (yellowish-reddish tubercles on lateral margin of mid-bulge and large median tubercles on area III), *Pinto-da-Rocha et al. (2012)* were the first to propose a diagnosis for the genus, which includes the presence of well-developed median tubercles on mesotergal areas (and add their elliptic form) and the lateral tubercles of mid-bulge paler than the rest of body, in addition to robust spines on the anterior border of dorsal scutum. *Pinto-da-Rocha et al. (2012)* also suggested that *Mischonyx* is closely-related to Hernandariinae.

In view of our results, we agree that the elliptic median tubercles on area III are diagnostic for *Mischonyx*. The shape of the tubercle differs in the clade containing *M. arlei* **comb. nov.**, *M. intermedius* and *M. minimus* **sp. nov.**, but is elliptic in all other species of the genus. Along with that, our character "Lateral tubercles on anterior margin of dorsal scutum subequal in size" (#7-0) is roughly equivalent to "robust spines on the anterior border of dorsal scutum" proposed by *Pinto-da-Rocha et al. (2012)*.

In the results of our analyses, *Mischonyx* is not close to the Hernandariinae species (*Piassagera brieni* and *Pseudotrogulus telluris*), even when only morphological characters are considered (Figs. S3–S5). This is in agreement with *Pinto-da-Rocha et al. (2014)*, who considered *Mischonyx squalidus* (*Mischonyx cuspidatus* in the article) to lie outside of Hernandariinae.

## Other taxonomical and topological remarks

Recent publications on the taxonomy and systematics of harvestmen considered *G. antiquus* as *a* member of *Mischonyx* (*Kury, 2003*; *Vasconcelos, 2005a* and *Pinto-da-Rocha et al., 2012*). Our morphological analysis also places this species inside the genus. However, these results are not consistent with molecular and TE analyses (Figs. 16–22 and S6–S8). In ML3, it is sister to *Ampheres leucopheus*, a Caelopyginae. This indicates that the morphological similarities are convergences.

On the other hand, MP2, which does not include morphological characters, places a clade with *Multumbo* and *Deltaspidium* species inside *Mischonyx*, as sister to the clade with species from SMSP, SSP, PR and SC AoE. This group makes no morphological or biogeographical sense, since these species are from Org and LSRJ AoE. However, when we include morphological characters, MP3 does not recover the same clade and excludes *Multumbo* and *Deltaspidium* from *Mischonyx* genus.

The arguments discussed in the last two paragraphs hightlight the importance of combining morphological and molecular data to solve conflicting topologies. *Wiens (2004)* and *Baker & Gatesy (2002)* supported the hypothesis that morphological data is important especially when the results from molecular analysis seem problematic. For example, in the research of *De Sá et al. (2014)*, questionable relationships among frog speces became elucidated when morphological and behavioral characters from both larvae and adults were added. Here, we conclude that morphological characters also helped to strengthen the hypotheses and solve some problematic relationships in MP2, consistent with *Wipfler et al. (2016)*, *Lee & Palci (2015)* and *Giribet (2015)* who consider morphological characters fundamental even in the phylogenomics era, since the combination of morphological and molecular data provide independent sources of evidence.

## CONCLUSIONS

The total evidence analyses in this research shows that *Mischonyx sensu Pinto-da-Rocha et al. (2012)* is not monophyletic. The new definition based on our data includes *Michonyx arlei* **comb. nov** and excludes *Mischonyx antiquus*, which is placed back in *Gonyleptes*. In addition, *Geraecormobiella Mello-Leitão, 1931b*, *Ariaeus Sørensen, 1932* and *Urodiabunus* Mello-Leitão, 1935 are junior synonyms of *Mischonyx* Bertkau, 1880. *Geraecormobiella convexa Mello-Leitão, 1931b* and *Geraeocormobius cheloides Mello-Leitão, 1940* are junior synonym of *Weyhia spinifrons Mello-Leitão, 1923*; *Ilhaia cuspidata Roewer, 1913*, *Ilhaia fluminensis Mello-Leitão, 1922*, *Gonazula gibbosa Roewer, 1930*, *Eduardoius granulosus Mello-Leitão, 1931a*, *Giltaya solitaria Mello-Leitão, 1932* and *Eduardoius lutescens Roewer, 1943* are junior synonym of *Mischonyx squalidus* Bertkau, 1880; *Ilhaia sulina Soares & Soares, 1947* is a junior synonym of *Xundarava anomala Mello-Leitão, 1936*. We describe three new species for the genus: *Mischonyx minimus* **sp. nov.**, *Mischonyx intervalensis* **sp. nov.** and *Mischonyx tinguaensis* **sp. nov.**. *Geraeocormobius reitzi* Vasconcelos, 2005, *Weyhia clavifemur* Mello-Leitão, 1927 and *Weyhia spinifrons Mello-Leitão, 1923* were transferred to *Mischonyx*. *Weyhia parva Roewer, 1917* was removed from the synonym with *Mischonyx squalidus*, Bertkau, 1880 (see *Kury, 2003*: 134), considered as a valid species and transferred to *Mischonyx*.

The new composition of the genus after all synonyms, combinations and new species description is as follows: *Mischonyx. anomalus* (*Mello-Leitão, 1936*); *Mischonyx arlei* (*Mello-Leitão, 1935b*) **comb. nov.**, *Mischonyx clavifemur*, (*Mello-Leitão, 1927a*) **comb. nov.**; *Mischonyx fidelis* (*Mello-Leitão, 1931a*); *Mischonyx insulanus* (*Soares, 1972*); *Mischonyx intermedius* (*Mello-Leitão, 1935b*); *Mischonyx intervalensis* **sp. nov.**; *Mischonyx kaisara* Vasconcelos, 2004; *Mischonyx minimus* **sp. nov.**; *Mischonyx parvus* (*Roewer, 1917*) **comb. nov.**; *Mischonyx poeta* Vasconcelos, 2005a; *Mischonyx processigerus* (*Soares &*

*Soares, 1970*); *Mischonyx reitzi* (Vasconcelos, 2005) **comb. nov.**; *Mischonyx scaber* (*Kirby, 1819*); *Mischonyx spinifrons* (*Mello-Leitão, 1923*) **comb. nov.**; *Mischonyx squalidus* Bertkau, 1880; *Mischonyx tinguaensis* **sp. nov.**

We believe that the most plausible phylogenetic hypothesis was recovered using Total Evidence and Maximum Likelihood. Unlike the rival hypotheses, it does not require an unusual number of character changes (apomorphies) leading to *M. tinguaensis* **sp. nov.**, it has high bootstrap support for *Mischonyx* and is well supported by morphological synapomorphies. The *Mischonyx* clade is supported by the following morphological characters: lateral tubercles on anterior margin of dorsal scutum with the same size, elliptic tubercles on area III, absence of prolateral apophysis on females, femur prolaterally curved, three to six apophysis on the apical half of retrolateral row on femur IV and brown as the general body color. There are two major clades inside *Mischonyx*: one with species from LSRJ, Mnt, Org and Esp AoE, and the other with species from SMSP, SSP, PR and SC AoE. The divergence time of these clades are in agreement with geological events. We estimate that *Mischonyx* clade diverged 50.53 Mya, and inside the genus there are two major clades. One of them cointains species from Paraná, Santa Catarina, South of São Paulo and Serra do Mar Areas of Endemism and the other has species from Espinhaço, Bocaina, South coast of Rio de Janeiro and Serra dos Órgãos Areas of Endemism. The first split inside these two clades occurred at 48.94 and 44.80 Mya, respectively.

## ACKNOWLEDGEMENTS

We thank Marcio B. da Silva, Cristina A. Rheims for their advice and suggestions. We are grateful to Gonzalo Giribet and an anonymous reviewer for suggestions on an early draft of the manuscript. We thank Jimmy Cabra-Garcia, Brittany Damron, Daniel Chirivi, Marília Pessoa Silva, Daniel Castro and André Nogueira for their help in field trips, opinions and advices during the whole process of writing this manuscript. We also thank Jairo Moreno-González for his huge help in our phylogenies and Adriano Kury for sharing pictures and information We thank Manuel Antunes Junior, Beatriz Vieira Freire, Phillip Lenktaitis, Ênio Mattos for their help in DNA sequencing and SEM operation.

### Funding

This study was funded by CAPES, CNPq (306722/2018-6), FAPESP (BIOTA, 2013/50297-0), NSF (DOB 1343578), and NASA to Ricardo Pinto-da-Rocha. The funders had no role in study design, data collection and analysis, decision to publish, or preparation of the manuscript.

### Grant Disclosures

The following grant information was disclosed by the authors:
CAPES, CNPq: 306722/2018-6.
FAPESP (BIOTA): 2013/50297-0; fapesp 2021/08430-0.

NSF (DOB): 1343578.
NASA to Ricardo Pinto-da-Rocha.

## Competing Interests

The authors declare that they have no competing interests.

## Author Contributions

- Caio Gueratto conceived and designed the experiments, performed the experiments, analyzed the data, prepared figures and/or tables, authored or reviewed drafts of the paper, and approved the final draft.
- Alípio Benedetti performed the experiments, analyzed the data, authored or reviewed drafts of the paper, and approved the final draft.
- Ricardo Pinto da Rocha analyzed the data, authored or reviewed drafts of the paper, and approved the final draft.

## Field Study Permissions

The following information was supplied relating to field study approvals (*i.e.*, approving body and any reference numbers):

Field expeditions and collections were approved by Ministério do Meio Ambiente (MMA), Instituto Chico Mendes de Conservação da Biodiversidade (ICMBio), Sistema de Autorização e Informação em Biodiversidade (SISBIO) (project number: 57281-2).

## DNA Deposition

The following information was supplied regarding the deposition of DNA sequences:

The morphological matrix and all the molecular data are available as Supplemental Files.

The internal transcribed spacer subunit 2 sequences are available at GenBank: MT957095 to MT957139.

The 28S ribosomal gene sequences are available at GenBank: MT990776 to MT990827.

The 12S ribosomal gene sequences are available at GenBank: MW000789 to MW000836.

The 16S ribosomal gene sequences are available at GenBank: MW000837 to MW000887.

The cytochrome oxidase subunit I gene sequences are available at GenBank: MT992257 to MT992308.

The carbamoyl-phosphate synthetase gene sequences are available at GenBank: MW017366 to MW017404.

The histone H3 gene sequences are available at GenBank: MW017405 to MW017455.

## Data Availability

Morphological matrix and sequence data are available in the Supplemental Files and at Morphobank: DOI: 10.7934/P3599.

## New Species Registration

The following information was supplied regarding the registration of a newly described species:

Publication LSID: urn:lsid:zoobank.org:pub:157DB276-ABB3-413A-B642-73F40FA28926

*Mischonyx minimus* sp. nov. LSID:
urn:lsid:zoobank.org:act:A6F34641-1AF1-4BE2-A16A-4A4497ECA1FC;
*Mischonyx intervalensis* sp. nov. LSID:
urn:lsid:zoobank.org:act:3DDE0A87-E9F6-4504-9C54-6DC37D202A0E;
*Mischonyx tinguaensis* sp. nov. LSID:
urn:lsid:zoobank.org:act:5FA4CC13-EC27-4E3A-AB19-81A97FE74177.

## Supplemental Information

Supplemental information for this article can be found online at http://dx.doi.org/10.7717/peerj.11682#supplemental-information.

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
