# Peer review of "Phylogenetic relationships of the genus Mischonyx Bertkau, 1880, with taxonomic changes and three new species description (Opiliones: Gonyleptidae)"

_PeerJ, doi:10.7717/peerj.11682_

## Round 0.1 · original submission · Major Revisions

Both reviewers have made important points, that should be addressed before sending back the revised version. Please, also check the English carefully.

·

Basic reporting

The English needs revision. I have made a few edits/suggestions in the pdf, but it needs to be read more carefully.

Also, there is room for improvement for the figures. The trees are not very readable, colors are difficult to see and aligning the names on the right doesn’t seem to work. Fonts need editing. Why not using the same font and size for all the trees? Also, there is no need to show every tree, many of which are very similar. It would be better to select some of the relevant tree, edit them properly for readability, and mention the others in the discussion (or add them as supplementary information).

The three maps should be combined into a single figure. I didn’t get the idea of why should they be separate, with huge overlap between them.

Experimental design

no comment

Validity of the findings

no comment

Additional comments

This is a thorough taxonomic revision of a genus of Gonyleptidae, Mischonyx, using a combination of multiple Sanger-based molecular markers and a new morphological data matrix. The data are analyzed using standard phylogenetic methods and the taxonomic results are translated into a classification, new generic assignments, and resulting in new species descriptions. The taxonomic part is well developed. There is also a dating analysis used to discuss some biogeographical hypotheses. Unfortunately, the fossil record for this part of the Laniatores tree is nonexistent, so the authors opt to us a mutation rate for their dating analyses. To me this is the weakest part of the study.

This will make another nice contribution to Laniatores systematics and biogeography after some edits and minor corrections (see below and in attached pdf):

I have made some grammar suggestions (in the pdf) but the paper needs to be thoroughly reviewed for language.

Gene names should be spelled out, and only abbreviated after they have been properly named (see my suggested corrections in the methods section).

Reviewer 2 ·

Basic reporting

Afterwards an extensive revision of the ms #52148 submitted to PeerJ: “Phylogenetic relationships of the genus Mischonyx Bertkau, 1880, with taxonomic changes and three new species description (Opiliones: Gonyleptidae)”, I have some recommendations for the authors.

The ms is clear and concise on its objectives, Introduction and Myschonyx background sections offer to the reader all basis to understand the main issue. Literature revised is appropriate and well referenced along the text.
Personally, I am pleased with the idea of the approach of morphological, molecular and total evidence analyzed separately by MP, ML and Bayes, currently there are few phylogenetics works on Opiliones including those kind of evidence data.

Experimental design

But, my major criticisms for the work, is about on methods as follows:

First. The authors on the Phylogenetic inferences on methods section emphasized the use of two optimal criteria, Maximum Parsimony (MP) and Maximum Likelihhod (ML), analyzing Morphology (Morpho), Molecular (Mol) and Total Evidence (TE). For Morpho, they analyzed through ML, the authors only mentioned: “…using the best model found by the program, which uses BIC …”, later for Mol and TE, they stated: “The program selected the best substitution model for each gene partition under the BIC … using the program ModelFinder …” Here I considered is relevant to know which are these selected models and why they were selected, in nowhere else is anything mentioned again about the models used. To analyze morphology separately or in a concatenated matrix concatenated through ML is recommended to select Mk model accounting for ascertainment bias allowing gamma-distribution rate variation across characters (Klopfstein & Spasojevic, 2019). But this information is not specified in the text, and should be clarify.

Second. Is not clear for me, why the authors only emphasize the use of MP and ML for the phylogenetic reconstruction, wherein these two methods have some similarities in the way how they estimate the phylogeny. In this regard, I recommend the inclusion of Bayesian Inference through Mr. Bayes to analyze the same three data sets. In this way, the third topology obtained surely would be different to those under MP and ML, and the authors would have more results for discussion and not only focusing on taxonomic decision.

Third. In the section Phylogenetic inferences, the authors say: “We used a parsimony method to analyze character change because, as pointed by Chen & Kuntner (2014), the aim is to understand the evolutionary changes of characters rather than the probability of particular ancestral states on the phylogeny”. Later they say: “To analyze character change throughout the phylogeny, we used Winclada 1.6.1.”, and almost the end of this section, they say: …we analyze the character changes through the phylogeny using parsimony on Winclada 1.6.1”.
In the first place, there is no specified method to estimate character change, which mapping criteria was chosen in this study? ACCTRAN, DELTRAN?
Also, there are many evidences that parsimony methods for inferred character-state transformation find the obvious intuitively answer, ignoring two important sources of error: mapping uncertainty and phylogenetic uncertainty (Felsenstein, 1988; Frumhoff & Reeve, 1994; Ronquist, 2004).
With this last I do not mean that the parsimony methods for reconstruction are not adequate, however, I do not consider necessary use the same criterion for mapping on both MP and ML topologies, because there are not significant differences between these topologies. Instead, I recommended compare character change using different approaches (e.g. Parsimony vs. Likelihood or Bayesian) mapping on either MP, ML or Bayes tree.

Fourth. On the figures. Currently, there are many software to do more visually attractive and understandable trees. In the present work, I have the impression that the trees obtained were edited as little as possible, the names of species are hard to see, and in the mapped trees, the number of character/state is not possible to see in detail. On the Figures 4, 7 and 12 the labels say: “The values near the nodes are the Bootstrap values of each node”, but, near the nodes are two values: E.g. 97.4/100, 86.1/50, 99.9/69 … ¿Which is the correct Bootstrap value? What does it mean this strange notation? On Figure 5, Most parsimonious trees, How many trees? It looks like as the strict consensus. On Figure 8, the node values are not the 95% HDP, these values are the node ages.

REFERENCES
Felsenstein J. 1988. Phylogenies and quantitative characters. Annual Revision of Ecology and Systematics. 19: 445-471.
Frumhoff PC, Reeve HK. 1994. Using phylogenies to test hypotheses of adaptation: A critique of some current proposals. Evolution. 48: 172-180.
Klopfstein S., Spasojevic T. 2019. Illustrating phylogenetic placement of fossils using RoguePlots: An example from ichneumonid parasitoid wasps (Hymenoptera, Ichneumonidae) and an extensive morphological matrix. PLoS ONE: 425090.

Validity of the findings

No comment

Additional comments

Dear Editor,

Afterwards an extensive revision of the ms #52148 submitted to PeerJ: “Phylogenetic relationships of the genus Mischonyx Bertkau, 1880, with taxonomic changes and three new species description (Opiliones: Gonyleptidae)”, I have some recommendations for the authors.

The ms is clear and concise on its objectives, Introduction and Myschonyx background sections offer to the reader all basis to understand the main issue. Literature revised is appropriate and well referenced along the text.
Personally, I am pleased with the idea of the approach of morphological, molecular and total evidence analyzed separately by MP, ML and Bayes, currently there are few phylogenetics works on Opiliones including those kind of evidence data.

But, my major criticisms for the work, is about on methods as follows:

First. The authors on the Phylogenetic inferences on methods section emphasized the use of two optimal criteria, Maximum Parsimony (MP) and Maximum Likelihhod (ML), analyzing Morphology (Morpho), Molecular (Mol) and Total Evidence (TE). For Morpho, they analyzed through ML, the authors only mentioned: “…using the best model found by the program, which uses BIC …”, later for Mol and TE, they stated: “The program selected the best substitution model for each gene partition under the BIC … using the program ModelFinder …” Here I considered is relevant to know which are these selected models and why they were selected, in nowhere else is anything mentioned again about the models used. To analyze morphology separately or in a concatenated matrix concatenated through ML is recommended to select Mk model accounting for ascertainment bias allowing gamma-distribution rate variation across characters (Klopfstein & Spasojevic, 2019). But this information is not specified in the text, and should be clarify.

Second. Is not clear for me, why the authors only emphasize the use of MP and ML for the phylogenetic reconstruction, wherein these two methods have some similarities in the way how they estimate the phylogeny. In this regard, I recommend the inclusion of Bayesian Inference through Mr. Bayes to analyze the same three data sets. In this way, the third topology obtained surely would be different to those under MP and ML, and the authors would have more results for discussion and not only focusing on taxonomic decision.

Third. In the section Phylogenetic inferences, the authors say: “We used a parsimony method to analyze character change because, as pointed by Chen & Kuntner (2014), the aim is to understand the evolutionary changes of characters rather than the probability of particular ancestral states on the phylogeny”. Later they say: “To analyze character change throughout the phylogeny, we used Winclada 1.6.1.”, and almost the end of this section, they say: …we analyze the character changes through the phylogeny using parsimony on Winclada 1.6.1”.
In the first place, there is no specified method to estimate character change, which mapping criteria was chosen in this study? ACCTRAN, DELTRAN?
Also, there are many evidences that parsimony methods for inferred character-state transformation find the obvious intuitively answer, ignoring two important sources of error: mapping uncertainty and phylogenetic uncertainty (Felsenstein, 1988; Frumhoff & Reeve, 1994; Ronquist, 2004).
With this last I do not mean that the parsimony methods for reconstruction are not adequate, however, I do not consider necessary use the same criterion for mapping on both MP and ML topologies, because there are not significant differences between these topologies. Instead, I recommended compare character change using different approaches (e.g. Parsimony vs. Likelihood or Bayesian) mapping on either MP, ML or Bayes tree.

Fourth. On the figures. Currently, there are many software to do more visually attractive and understandable trees. In the present work, I have the impression that the trees obtained were edited as little as possible, the names of species are hard to see, and in the mapped trees, the number of character/state is not possible to see in detail. On the Figures 4, 7 and 12 the labels say: “The values near the nodes are the Bootstrap values of each node”, but, near the nodes are two values: E.g. 97.4/100, 86.1/50, 99.9/69 … ¿Which is the correct Bootstrap value? What does it mean this strange notation? On Figure 5, Most parsimonious trees, How many trees? It looks like as the strict consensus. On Figure 8, the node values are not the 95% HDP, these values are the node ages.

REFERENCES
Felsenstein J. 1988. Phylogenies and quantitative characters. Annual Revision of Ecology and Systematics. 19: 445-471.
Frumhoff PC, Reeve HK. 1994. Using phylogenies to test hypotheses of adaptation: A critique of some current proposals. Evolution. 48: 172-180.
Klopfstein S., Spasojevic T. 2019. Illustrating phylogenetic placement of fossils using RoguePlots: An example from ichneumonid parasitoid wasps (Hymenoptera, Ichneumonidae) and an extensive morphological matrix. PLoS ONE: 425090.

---

## Round 0.2 · Minor Revisions

Please address comments by the reviewer.

Reviewer 2 ·

Basic reporting

To Editor and authors,

After a second revision of the manuscript entitled: “Phylogenetic relationships of the genus Mischonyx Bertkau, 1880, with taxonomic changes and three new species description (Opiliones: Gonyleptidae)”, I have the following comments.

The manuscript has improved too much, with the language revision, all the paragraphs in the document are clearer. However, reviewing the document with tracked changes, there are still small errors in the writing, e.g. two continuous end points, errors in some italics, etc.
On the phylogenetics methods, with the new additions, it is now understandable how the authors obtained the different topologies.
On discussion section, changes on the divergence time and optimality criteria analysis choosing only a tree ML3 tree, have made this part of the manuscript comprehensible to the reader.

However, despite the best of the manuscript, there are still small details to be considered accepted, which are:
1.- On abstract and conclusions, the authors do not mention almost anything about divergence and biogeographic inferences, placing greater emphasis on the phylogenetic and taxonomic part only. I recommend a larger paragraph mentioning more about time divergence, which is a very important part of the manuscript.
2.- Trees visualization. On the previous version of the manuscript, I made the suggestion to change the current trees for others with a better visual quality, this in order to better convey the idea that each tree represents. In this sense, the overall design of the trees did not change at all, or fuse all of them in only one or two trees. There are many packages in R to make very illustrative and nice trees, for example, the dated tree could be accompanied with an ancient distribution map, some picture of one species, etc. Trees with character changes have the number of character and character state very small on the branch, and also the design is “wide”, spanning a lot of white space unnecessarily. Figure 23 Total Evidence Parsimony hypothesis, values near the nodes are the bootstrap values, but, on each node there are two values, the bootstrap one and another after the slash, please clarify what values are those.

Experimental design

No comment.

Validity of the findings

No comment.

Additional comments

To Editor and authors,

After a second revision of the manuscript entitled: “Phylogenetic relationships of the genus Mischonyx Bertkau, 1880, with taxonomic changes and three new species description (Opiliones: Gonyleptidae)”, I have the following comments.

The manuscript has improved too much, with the language revision, all the paragraphs in the document are clearer. However, reviewing the document with tracked changes, there are still small errors in the writing, e.g. two continuous end points, errors in some italics, etc.
On the phylogenetics methods, with the new additions, it is now understandable how the authors obtained the different topologies.
On discussion section, changes on the divergence time and optimality criteria analysis choosing only a tree ML3 tree, have made this part of the manuscript comprehensible to the reader.

However, despite the best of the manuscript, there are still small details to be considered accepted, which are:
1.- On abstract and conclusions, the authors do not mention almost anything about divergence and biogeographic inferences, placing greater emphasis on the phylogenetic and taxonomic part only. I recommend a larger paragraph mentioning more about time divergence, which is a very important part of the manuscript.
2.- Trees visualization. On the previous version of the manuscript, I made the suggestion to change the current trees for others with a better visual quality, this in order to better convey the idea that each tree represents. In this sense, the overall design of the trees did not change at all, or fuse all of them in only one or two trees. There are many packages in R to make very illustrative and nice trees, for example, the dated tree could be accompanied with an ancient distribution map, some picture of one species, etc. Trees with character changes have the number of character and character state very small on the branch, and also the design is “wide”, spanning a lot of white space unnecessarily. Figure 23 Total Evidence Parsimony hypothesis, values near the nodes are the bootstrap values, but, on each node there are two values, the bootstrap one and another after the slash, please clarify what values are those.

---

## Round 0.3 · accepted · Accept

Thanks for correcting the previous version.